# A Non-Asymptotic Convergent Analysis for Scored-Based Graph Generative Model via a System of Stochastic Differential Equations

**Junwei Su** [1]  **Chuan Wu** [1]

## Abstract

Score-based graph generative models (SGGMs) have proven effective in critical applications such as drug discovery and protein synthesis. However, their theoretical behavior, particularly regarding convergence, remains underexplored. Unlike common score-based generative models (SGMs), which are governed by a single stochastic differential equation (SDE), SGGMs involve a system of coupled SDEs. In SGGMs, the graph structure and node features are governed by separate but interdependent SDEs. This distinction makes existing convergence analyses from SGMs inapplicable for SGGMs. In this work, we present the first non-asymptotic convergence analysis for SGGMs, focusing on the convergence bound (the risk of generative error) across three key graph generation paradigms: (1) feature generation with a fixed graph structure, (2) graph structure generation with fixed node features, and (3) joint generation of both graph structure and node features. Our analysis reveals several unique factors specific to SGGMs (e.g., the topological properties of the graph structure) which affect the convergence bound. Additionally, we offer theoretical insights into the selection of hyperparameters (e.g., sampling steps and diffusion length) and advocate for techniques like normalization to improve convergence. To validate our theoretical findings, we conduct a controlled empirical study using synthetic graph models, and the results align with our theoretical predictions. This work deepens the theoretical understanding of SGGMs, demonstrates their applicability in critical domains, and provides practical guidance for designing effective models.

[1]School of Computing and Data Science, University of Hong Kong. Correspondence to: Junwei Su <junweisu@connect.hku.hk>, Chuan Wu <cwu@cs.hku.hk>.

*Proceedings of the 42nd International Conference on Machine Learning*, Vancouver, Canada. PMLR 267, 2025. Copyright 2025 by the author(s).

## 1. Introduction

Graph-structured data is ubiquitous across a wide range of domains, including social networks, biological systems, recommendation engines, and knowledge graphs (Newman, 2018). The graph generation problem involves creating new graphs that closely resemble real-world data, a task critical to many graph-based applications. In recent years, **score-based graph generative models (SGGMs)** (Niu et al., 2020; Jo et al., 2022; Vignac et al., 2022; Chen et al., 2023c) have emerged as a flexible and powerful approach, delivering state-of-the-art empirical performance in graph generation tasks. These models have shown significant impact in critical areas such as molecular generation (Gnaneshwar et al., 2022), protein design (Lee et al., 2023c), graph-based recommendation systems (Liu et al., 2024), and automated program generation (Zhu et al., 2022).

SGGMs are a subclass of the broader family of **score-based generative models (SGMs)**, also referred to as diffusion probabilistic models (Song et al., 2020a; Ho et al., 2020; Nichol & Dhariwal, 2021; Sohl-Dickstein et al., 2015; Song & Ermon, 2019; Song et al., 2020b; Li et al., 2024; Yang et al., 2024). SGMs generate data by learning a **score function** (see Sec. 3 for more details), which represents the gradient of the log-probability of the data distribution. These models conceptualize the data generation process as a diffusion process, where data is progressively corrupted by noise and then reconstructed through a reverse process. The forward process involves gradually adding noise over several steps, transforming the data into a simple distribution (e.g., Gaussian noise), and is governed by a **stochastic differential equation (SDE)** (Song et al., 2020b). In the reverse process, the model uses the learned score function to iteratively denoise the data, starting from random noise and guiding it step-by-step toward the original data distribution. To efficiently approximate this reverse process, a discrete sampling approach is employed, using numerical methods such as the Euler-Maruyama scheme or the exponential integrator (Chen et al., 2023a).

Understanding the convergence behavior of SGGMs is both theoretically and practically crucial (Suh & Cheng, 2024; Huang et al., 2024; Chen et al., 2022a; 2024a; Lee et al., 2023b). Convergence bounds quantify the discrepancy be-

tween the generated graphs and the true data distribution, indicating whether the diffusion process will eventually produce samples that align with the desired target distribution. *This is particularly important for SGGMs, as convergence guarantees the reliability and validity of the generated graphs, which are used in high-stakes applications like drug discovery*. Furthermore, understanding convergence provides valuable insights into practical considerations, such as the selection of hyperparameters (e.g., sampling steps and diffusion length). Convergence analysis also helps identify sources of error and guides the development of more effective models.

**Gap in Existing Research.** Despite the empirical success of SGGMs in high-stakes applications, their convergence behavior remains underexplored. Most existing research has primarily focused on SGMs derived from the image domain (Yeğin & Amasyalı, 2024; Li et al., 2024; Wang et al., 2024; Chen et al., 2024b; Benton et al., 2023). There are two key differences between SGGMs and SGMs. *First*, while the generative process of SGMs is governed by a single SDE, SGGMs, due to the nature of graph data (which includes both graph structure and node features), involve a system of coupled SDEs (Jo et al., 2022; Niu et al., 2020). In SGGMs, the graph structure and node features are governed by separate but interdependent SDEs. *Second*, SGM convergence analyses often assume independence between data elements (e.g., pixels in images), while in SGGMs, the graph structure and node features are inherently coupled and interdependent (Deshpande et al., 2018). Thus, the formulation and convergence analysis of SGGMs must account for these relationships. As a result, existing convergence analyses for SGMs cannot be directly applied to SGGMs, highlighting the need for new theoretical formulations and analyses to understand and ensure their convergence.

**Problem Studied and Challenges.** In this paper, we address this gap by extending the convergence analysis to SGGMs. Several key challenges arise in this extension: *First*, the interdependency between graph structure and node features requires a careful and nuanced formulation. Unlike the independent elements typically found in conventional SGMs, the graph structure and node features in SGGMs are deeply interconnected. Accurately modeling this interdependency is crucial for capturing the true generative process of SGGMs and deriving meaningful insights. *Second*, this interdependency creates entangled dynamics in the generative process. Changes in the graph structure directly affect the node features, and vice versa. This reciprocal influence complicates the convergence analysis, as conventional methods that assume independence are no longer applicable. The simultaneous evolution of both the graph structure and node features calls for the development of new tools and methodologies to analyze the convergence of the entire system effectively. *Finally*, it is essential to connect the

factors and insights from this analysis with the practical SGGM framework. This includes not only understanding the theoretical results in the context of graph data but also leveraging these insights to guide model design decisions, such as hyperparameter tuning (e.g., sampling steps, diffusion length) and implementing regularization techniques. These challenges make the convergence analysis of SGGMs both novel and non-trivial.

**Our Contributions and Results.** We present a comprehensive formulation for SGGMs that captures the complex interdependencies between graph structure, node features, and the use of graph neural networks (GNNs). Based on this formulation, we present a detailed non-asymptotic convergence analysis for three common graph generation paradigms: 1) node feature generation with a fixed graph structure (Theorem 4.1), 2) graph structure generation with fixed node features (Theorem 4.2), and 3) joint generation of both graph structure and node features (Theorem 4.3). Our analysis provides a detailed account of the factors influencing convergence bound of SGGMs. In addition, our results reveal several key insights and implications (Sec. 4) regarding the convergence behavior of SGGMs:

1. **Graph size vs. feature dimensionality**: There is a non-isotropic effect between the graph size (number of nodes) and feature dimensionality. Specifically, increasing the size of the graph leads to a greater risk of generative error (i.e., larger convergence bounds) than increasing the dimensionality of node features. This finding helps explain why SGGMs are empirically effective for smaller graphs, even when the node features are complex (Jo et al., 2022).

2. **Topological properties of the graph structure**: The topological properties of the graph significantly influence the convergence bounds of SGGMs. In particular, graphs with heterogeneous degree distributions—where some nodes have significantly more connections than others—tend to result in a larger generative error. Our findings suggest that SGGMs perform more reliably when generating graphs with more uniform degree distributions, as the risk of large error is lower in these graphs. This insight is critical for applications where the generated graphs need to closely match real-world graph structures.

3. **Impact of norms on feature matrices**: Smaller norms in the feature matrices reduce the risk of generative error in the generated graphs, providing theoretical support for employing normalization techniques in SGGMs. By controlling the scale of the feature matrices, these techniques help tighten the convergence bounds, resulting in more stable and accurate graph generation.

The empirical results from our controlled experiments using synthetic graphs are consistent with our theoretical predictions, thereby validating our analysis and conclusions.

These results enhance the theoretical understanding of SG-GMs, confirm their applicability in high-stakes applications, and provide practical insights for designing and deploying more effective SGGMs.

## 2. Related Work

**Graph Generation Methods.** The graph generation problem has a long history of study, with rule-based random graph models traditionally dominating the field (Barabási & Albert, 1999; Holland et al., 1983; Erdős et al., 1960; Newman et al., 2002). A prime example of such models is the Stochastic Block Model (SBM) (Holland et al., 1983), which is based on the observation that real-life graphs often consist of densely connected blocks of vertices exhibiting similar behaviors (Abbe, 2018; Newman et al., 2002; Newman & Girvan, 2004; Newman, 2006; Karrer & Newman, 2011; Cherifi et al., 2019; Su & Marbach, 2022). However, such rule-based models fail to capture the complex and nuanced distribution of graph-structured data observed in real-world problems (Russell & Norvig, 2016). As a result, the focus has shifted towards deep learning-based methods that can model more intricate graph properties (Zhu et al., 2022; You et al., 2018; Xie et al., 2021; Liao et al., 2019; Li et al., 2018; Jensen, 2019; Fu et al., 2021; De Cao & Kipf, 2018; Zang & Wang, 2020; Simonovsky & Komodakis, 2018). Among these, SGGMs have emerged as a promising approach, showing impressive empirical performance in graph generation (Niu et al., 2020; Jo et al., 2022; Vignac et al., 2022; Chen et al., 2023c). Due to the nature of graph data, the standard formulation of SGGMs involves a system of stochastic processes, either manifested as a Markov chain (discrete) (Chen et al., 2023c; Vignac et al., 2022) or SDE (continuous) (Jo et al., 2022; Niu et al., 2020). In this paper, we focus on the formulation with SDE, and point out our analysis can be extended to the discrete version by replacement of suitable theoretical tools (Sec. 6).

**Convergent Analysis of SGMs.** Theoretical studies on SGMs have garnered significant attention in recent years (De Bortoli et al., 2021; Zhang et al., 2024; Lee et al., 2022; 2023a; Wang et al., 2024; Chen et al., 2024b; Li et al., 2023; Benton et al., 2023; Chen et al., 2022a; 2023a;b). A central focus of these studies is examining the convergence behavior of these models, specifically how well they approximate the true data distribution. Existing research has shown that, under suitable smoothness and regularity assumptions on the score functions, SGMs provide provable guarantees for convergence, meaning that the generated samples increasingly resemble the true data distribution as the number of iterations increases (Chen et al., 2022a; 2023a;b). However, much of the existing work has been motivated by tasks such as image generation, where the generative process is governed by a single stochastic process. In con-

trast, SGGMs involve a system of dependent SDEs, with separate equations governing the graph structure and the node features, respectively. It remains unclear how this interconnectedness nature of graph data is manifested in the convergent behavior of SGGMs. In this paper, we address this gap by providing a non-asymptotic convergence analysis for SGGMs, accounting for the distinctive challenges posed by graph data.

## 3. Preliminaries and Problem Formulation

In this section, we introduce the formulation SGGMs via a system of SDEs, and different graph generation paradigms.

**Notation.** We use bold upper and lower-case letters to denote vectors and matrices. For two functions $f(x) \geq 0$ and $g(x) \geq 0$, we write $f(x) \lesssim g(x)$ if $f(x) \leq c \cdot g(x)$ for some absolute constant $c > 0$. For a given SDE, we use $x$ to denote the forward continuous process, $\bar{x}$ to denote the backward continuous process and $\widehat{x}$ to denote the backward approximated process. A graph $\mathcal{G}$ can be formally defined as a two-tuple $\mathcal{G} = (\mathbf{X}, \mathbf{A})$, where $\mathbf{X} \in \mathbb{R}^{N \times F}$ is the node feature matrix and $\mathbf{A} \in \mathbb{R}^{N \times N}$ is the adjacency matrix representing the graph structure. Here, $N$ is the number of nodes in the graph and $F$ is the dimension of node features. In addition, we use $\mathbb{N}(a, b)$ to denote Gaussian distribution with mean $a$ and variance $b$.

### 3.1. SGGMs via System of SDEs.

SGGMs consist of three main components: 1) a forward process, 2) a reverse process, and 3) a sampling process that approximates the reverse process to facilitate data generation. In the following sections, we provide an introduction to each of these components.

#### 3.1.1. FORWARD PROCESS.

Formally, the forward diffusion process can be described by the trajectory of random variables $\{\mathcal{G}_t = (\mathbf{X}_t, \mathbf{A}_t)\}_{t \in [0,T]}$ in a fixed time horizon $[0, T]$, where $\mathcal{G}_0 = (\mathbf{X}_0, \mathbf{A}_0)$ is sampled from the data distribution $\mathbb{P}(\mathcal{G}_0)$ (what we want to learn). This process is modelled using the following system of dependent SDEs:

$$\begin{aligned} d\mathbf{X}_t &= f_{\mathbf{X}}(\mathcal{G}_t, t) \, dt + g_{\mathbf{X}}(\mathcal{G}_t, t) \, d\mathbf{W}_{\mathbf{X}}, \\ d\mathbf{A}_t &= f_{\mathbf{A}}(\mathcal{G}_t, t) \, dt + g_{\mathbf{A}}(\mathcal{G}_t, t) \, d\mathbf{W}_{\mathbf{A}}, \end{aligned} \quad (3.1)$$

where the first equation is the forward SDE for the node feature and the second equation is the forward SDE for the graph structure. $f_{\mathbf{X}}(.)$ and $f_{\mathbf{A}}(.)$ are drift coefficients, and $g_{\mathbf{X}}(.)$ and $g_{\mathbf{A}}(.)$ are scalar diffusion coefficients. $\mathbf{W}_{\mathbf{X}}, \mathbf{W}_{\mathbf{A}}$ are standard Wiener processes (also known as Brownian motion) acting in the node feature space and graph structure space, respectively. This diffusion process gradually adds

noise to the initial graph samples, and at the terminal time $T$, the sample $\mathcal{G}_T = (\mathbf{X}_T, \mathbf{A}_T)$ follows a simple convergent distribution (also referred to as the prior distribution) which we denote as $\mathbf{\Pi_A}$ and $\mathbf{\Pi_X}$ for the graph structure process and node feature process respectivelys. In this paper, we focus on the commonly used standard Gaussian distribution (with zero mean and unit variance) as the convergent distribution.

We focus our analysis on the following choices of the drift and diffusion coefficients:

$$
\begin{aligned}
f_{\mathbf{X}}(\mathcal{G}_t, t) &= -1/2 g_{\mathbf{X}}(\mathcal{G}_t, t)^2 \mathbf{X}_t, \\
f_{\mathbf{A}}(\mathcal{G}_t, t) &= -1/2 g_{\mathbf{A}}(\mathcal{G}_t, t)^2 \mathbf{A}_t, \\
g_{\mathbf{X}}(\mathcal{G}_t, t) &= g_{\mathbf{A}}(\mathcal{G}_t, t) \equiv 1.
\end{aligned} \tag{3.2}
$$

These choices match those in the original SGMs paper (Song et al., 2020a; Jo et al., 2022), and are commonly used for similar convergence analysis (Chen et al., 2023a). We emphasize that our analysis can be adapted for some other choices of linear drift terms and constant variance functions as well (we provide a further discussion on this regard in Appendix A).

Under these choices of coefficients, the forward process becomes the Ornstein-Uhlenbeck (OU) process (Jacobsen, 1996), which has an explicit conditional density:

$$
\begin{aligned}
\mathbf{X}_t | \mathbf{X}_0 &\sim \mathbb{N}(e^{-1/2t} \mathbf{X}_0, (1 - e^{-t}) \mathbf{I}_{N \times F}), \\
\mathbf{A}_t | \mathbf{A}_0 &\sim \mathbb{N}(e^{-1/2t} \mathbf{A}_0, (1 - e^{-t}) \mathbf{I}_{N \times N}).
\end{aligned} \tag{3.3}
$$

Moreover, the OU process converges exponentially to the standard Gaussian distribution:

$$
\begin{aligned}
\mathrm{KL}(\mathbb{P}(\mathbf{X}_t) \| \mathbf{\Pi_X}) &\leq e^{-t} \mathrm{KL}(\mathbb{P}(\mathbf{X}_0) \| \mathbf{\Pi_X}), \\
\mathrm{KL}(\mathbb{P}(\mathbf{A}_t) \| \mathbf{\Pi_A}) &\leq e^{-t} \mathrm{KL}(\mathbb{P}(\mathbf{A}_0) \| \mathbf{\Pi_A}),
\end{aligned}
$$

where KL denotes the Kullback–Leibler (KL) divergence used to quantify the discrepancy of two distributions (see Appendix D for a further discussion).

### 3.1.2. REVERSE PROCESS.

The reverses process aims to generate graph samples from the convergent distribution by reversing the forward process. This is also described by a system of SDEs driven by the score function:

$$
\begin{aligned}
\mathrm{d}\bar{\mathbf{X}}_t &= \left[1/2 \bar{\mathbf{X}}_t - \nabla_{\mathbf{X}} \log \mathbb{P}(\mathcal{G}_t)\right] \mathrm{d}t + \mathrm{d}\bar{\mathbf{W}}_{\mathbf{X}}, \\
\mathrm{d}\bar{\mathbf{A}}_t &= \left[1/2 \bar{\mathbf{A}}_t - \nabla_{\mathbf{A}} \log \mathbb{P}(\mathcal{G}_t)\right] \mathrm{d}t + \mathrm{d}\bar{\mathbf{W}}_{\mathbf{A}},
\end{aligned} \tag{3.4}
$$

where $\bar{\mathbf{X}}_t, \bar{\mathbf{A}}_t, \bar{\mathbf{W}}_{\mathbf{X}}, \bar{\mathbf{W}}_{\mathbf{A}}$ are the respective reverse processes in time. $\nabla_{\mathbf{X}} \log \mathbb{P}(\mathcal{G}_t)$ and $\nabla_{\mathbf{A}_t} \log \mathbb{P}(\mathcal{G}_t)$ are the partial score (partial gradient of log-probility of graph distribution) with respect to node feature and graph structure respectively.

**Training SGGMs.** The partial score functions can be estimated by training time-dependent neural networks (also referred to as the score networks) $s_{\boldsymbol{\theta}}(.)$ and $s_{\boldsymbol{\phi}}(.)$, so that

$$
s_{\boldsymbol{\theta}}(\mathcal{G}_t, t) \approx \nabla_{\mathbf{X}} \log \mathbb{P}(\mathcal{G}_t), \quad s_{\boldsymbol{\phi}}(\mathcal{G}_t, t) \approx \nabla_{\mathbf{A}_t} \log \mathbb{P}(\mathcal{G}_t),
$$

where $\theta$ and $\phi$ are learnable parameters. Since the drift coefficients of the forward diffusion process are linear, the transition distribution $\mathbb{P}(\mathcal{G}_t | \mathcal{G}_0)$ can be decomposed in terms of $\mathbf{X}_t$ and $\mathbf{A}_t$, as follows:

$$
\mathbb{P}(\mathcal{G}_t | \mathcal{G}_0) = \mathbb{P}(\mathbf{X}_t | \mathbf{X}_0) \mathbb{P}(\mathbf{A}_t | \mathbf{A}_0).
$$

Notably, it is easy to sample from the transition distributions of each component, $\mathbb{P}(\mathbf{X}_t | \mathbf{X}_0)$ and $\mathbb{P}(\mathbf{A}_t | \mathbf{A}_0)$, as they are Gaussian distributions with mean and variance determined by the coefficients of the forward diffusion process given by Eq. 3.3. Then, with the decomposition above, training objectives of SGGMs generalize the one from (Song et al., 2020b) to minimizing the estimation of partial scores on the given graph dataset:

$$
\min_{\boldsymbol{\theta}} \mathbb{E}_t \left[ \mathbb{E}_{\mathcal{G}_0} \mathbb{E}_{\mathcal{G}_t | \mathcal{G}_0} \big\| s_{\boldsymbol{\theta}}(\mathcal{G}_t, t) - \nabla_{\mathbf{X}} \log \mathbb{P}(\mathbf{X}_t | \mathbf{X}_0) \big\|_2^2 \right],
$$

$$
\min_{\boldsymbol{\phi}} \mathbb{E}_t \left[ \mathbb{E}_{\mathcal{G}_0} \mathbb{E}_{\mathcal{G}_t | \mathcal{G}_0} \big\| s_{\boldsymbol{\phi}}(\mathcal{G}_t, t) - \nabla_{\mathbf{A}} \log \mathbb{P}(\mathbf{A}_t | \mathbf{A}_0) \big\|_2^2 \right].
$$

The expectations above can be efficiently computed using Monte Carlo estimation with samples $(t, \mathcal{G}_0, \mathcal{G}_t)$. For completeness, we provide a more detailed derivation and discussion on the training objectives in Appendix A.

**Functional Form of Score and Score Networks.** To capture and model the dependencies between graph structure and node features, GNNs, such as Graph Convolutional Networks (GCNs) (Kipf & Welling, 2017) and Graph Transformers (Dwivedi & Bresson, 2020), are used as the score networks for SGGMs. A defining characteristic of GNNs is their usage of graph structure to combine and update the representation of each node or edge. From a functional perspective, GNNs can be expressed as a function $\mathscr{F}(\mathcal{T}(\mathbf{A})\mathcal{T}'(\mathbf{X}))$, where $\mathcal{T}$ and $\mathcal{T}'$ represent some transformations applied to the adjacency matrix $\mathbf{A}$ and the feature matrix $\mathbf{X}$, respectively. For simplicity and interpretability, in this paper, we focus on the case where $\mathcal{T}$ and $\mathcal{T}'$ are identity mappings, corresponding to a vanilla GCN without the normalization. Our analysis and results can be extended to other transformations by replacing $\mathbf{A}, \mathbf{X}$ with $\mathcal{T}(\mathbf{A}), \mathcal{T}'(\mathbf{X})$. In addition, we focus on a feasible setting where we assume certain regularity conditions on the data distribution, and that the score function aligns with the functional form of GNNs.

**Assumption 3.1.** The data distributions for node features and graph structure are twice differentiable and have bounded second moments, i.e.,

$$
H_{\mathbf{X}} := \mathbb{E}\|\mathbf{X}\|^2 \leq \infty, \quad H_{\mathbf{A}} := \mathbb{E}\|\mathbf{A}\|^2 \leq \infty.
$$

Furthermore, the score functions $\nabla \log \mathbb{P}_t$ are $L$-Lipschitz and can be written as a function of the form $\mathscr{F}(\mathbf{A}_t \mathbf{X}_t)$.

The regularity conditions such as continuity, boundedness and smoothness of data distributions and score function are commonly employed in other similar studies (Benton et al., 2023; Chen et al., 2022a; 2023a;b). We note that the smoothness assumption could be relaxed in recent analysis (Chen et al., 2023a); however, such a relaxation would complicate the results. In this paper, we target more user-friendly results for better interpretability and connection with practical settings, and hence retain these standard assumptions.

### 3.1.3. SAMPLING PROCESS.

A discrete-time approximation of the sampling dynamics in Eq. 3.4 is required for practical implementation. Let $0 = t_0 \leq t_1 \leq \cdots \leq t_M = T$ be the discretization points. For the $k$-th discretization step ($1 \leq k \leq M$), we denote $\Delta t_k := t_k - t_{k-1}$ as the step size for the $k$-th step. Let $t'_k = T - t_{M-k}$ be the corresponding discretization points in the reverse-process SDE. In addition, we make the following assumption on the learned score functions with respect to the discretization.

**Assumption 3.2.** For both score functions $s_{\boldsymbol{\theta}}(.)$ and $s_{\boldsymbol{\phi}}(.)$, we assume that there exist constant $\epsilon_{\mathbf{X}}, \epsilon_{\mathbf{A}}$ for any $1 \leq i \leq M$:

$$\sum_{i=1}^{M} \frac{\Delta t_i}{T} \mathbb{E} \left\| \nabla_{\mathbf{X}} \log \mathbb{P}(\mathcal{G}_{t_i}) - s_{\boldsymbol{\theta}}(\mathcal{G}_{t_i}, t_i) \right\|^2 \leq \epsilon_{\mathbf{X}}^2,$$

$$\sum_{i=1}^{M} \frac{\Delta t_i}{T} \mathbb{E} \left\| \nabla_{\mathbf{A}} \log \mathbb{P}(\mathcal{G}_{t_i}) - s_{\boldsymbol{\phi}}(\mathcal{G}_{t_i}, t_i) \right\|^2 \leq \epsilon_{\mathbf{A}}^2.$$

Assumption 3.2 is commonly used in convergence analyses of diffusion models with general distributions (Chen et al., 2022a; Zhang et al., 2024; Chen et al., 2023a). We will further discuss in Sec. 6 how we can extend the analysis and relax this assumption to obtain a more precise (but less general) result, with additional modelling assumptions on the underlying distribution.

We consider two types of discretization schemes that are widely used in existing works: the Euler-Maruyama scheme and the exponential integrator scheme.

**Euler-Maruyama Scheme** is a simple, first-order discretization method that approximates an SDE by applying a first-order Taylor approximation. For approximated trajectory $\widehat{\mathbf{X}}_t, \widehat{\mathbf{A}}_t$ with the discrete scheme, the progression rule at the $k$-th step is given by:

$$\widehat{\mathbf{X}}_{t'_{k+1}} = \widehat{\mathbf{X}}_{t'_k} + \left[ -1/2 \widehat{\mathbf{X}}_{t'_k} - s_{\boldsymbol{\theta}}(\widehat{\mathcal{G}}_{t'_k}, t'_k) \right] \Delta t_k + \Delta t_k \bar{\mathbf{W}}_{\mathbf{X}},$$

$$\widehat{\mathbf{A}}_{t'_{k+1}} = \widehat{\mathbf{A}}_{t'_k} + \left[ -1/2 \widehat{\mathbf{A}}_{t'_k} - s_{\boldsymbol{\phi}}(\widehat{\mathcal{G}}_{t'_k}, t'_k) \right] \Delta t_k + \Delta t_k \bar{\mathbf{W}}_{\mathbf{A}},$$
$$(3.5)$$

where $\Delta \bar{\mathbf{W}}_{\mathbf{A}}$ and $\Delta \bar{\mathbf{W}}_{\mathbf{A}}$ represent the increment in the Wiener processes within the interval.

**Exponential Integrator Scheme** provides a more accurate method for solving SDEs, especially when the system has a semi-linear structure. It discretizes the nonlinear terms while retaining the continuous dynamics arising from the linear terms. Specifically, the nonlinear term is discretized while the linear part evolves continuously according to:

$$\begin{aligned} \widehat{\mathbf{X}}_{t'_{k+1}} =& e^{1/2\Delta t_k} \widehat{\mathbf{X}}_{t'_k} + 2 \left( e^{1/2\Delta t_k} - 1 \right) s_{\boldsymbol{\theta}}(\widehat{\mathcal{G}}_{t'_k}, t'_k) \\ & + \sqrt{e^{\Delta t_k} - 1} \boldsymbol{\xi}_k, \\ \widehat{\mathbf{A}}_{t'_{k+1}} =& e^{1/2\Delta t_k} \widehat{\mathbf{A}}_{t'_k} + 2 \left( e^{1/2\Delta t_k} - 1 \right) s_{\boldsymbol{\phi}}(\widehat{\mathcal{G}}_{t'_k}, t'_k) \\ & + \sqrt{e^{\Delta t_k} - 1} \boldsymbol{\xi}'_k. \end{aligned} \quad (3.6)$$

where $\boldsymbol{\xi}_k$ and $\boldsymbol{\xi}'_k$ are sampled from standard Gaussian.

## 3.2. Graph Generation Paradigms

For graph generation tasks, there are three distinct paradigms, each with its unique significance.

**Joint Generation of Graph Structure and Node Features.** This paradigm aims to simultaneously generate both the graph structure and node features. It is particularly significant in biological applications, such as protein design (Vignac et al., 2022; Jo et al., 2022), where the graph structure represents the interactions between proteins (nodes), and the node features describe attributes like protein function, structure, or expression levels. Jointly generating both components ensures that the resulting graph is biologically plausible, accurately reflecting both the interaction patterns and functional properties of the proteins. The corresponding forward process is given by Eq. 3.1.

**Node Feature Generation with Fixed Graph Structure.** In this paradigm, the goal is to generate node features while maintaining a fixed graph structure. This is particularly important in domains like molecular graph generation, where the connectivity of atoms is predefined, but their properties—such as chemical features—can vary (Sanchez-Lengeling et al., 2017; Simonovsky & Komodakis, 2018; Jin et al., 2018). By focusing on feature generation, this paradigm enables the exploration of diverse attribute combinations while maintaining a consistent structural backbone, facilitating the generation of realistic and varied instances for a given graph structure. In this setting, node features are generated based on a fixed graph structure. The corresponding forward process is:

$$\begin{aligned} \mathrm{d}\mathbf{X}_t &= 1/2 \mathbf{X}_t \, \mathrm{d}t + \mathrm{d}\mathbf{W}_{\mathbf{X}}, \\ \mathrm{d}\mathbf{A}_t &= 0, \quad \mathbf{A}_0 = \mathbf{A}^* \end{aligned} \quad (3.7)$$

where $\mathbf{A}^*$ is the fixed graph structure that remains constant during the feature generation process.

**Graph Structure Generation with Fixed Node Features.** This paradigm focuses on generating the graph structure while keeping the node features fixed. It is particularly useful when the node features are known or predefined, but the relationships between the nodes (i.e., the graph structure) need to be learned or generated (Martínez et al., 2016; Zhang & Chen, 2018; Benson et al., 2016; Niu et al., 2020). For example, in social network analysis, the characteristics of individuals (e.g., age, location, interests) are known, but the interactions or connections between them need to be inferred. In this setting, the generating process focuses solely on the graph structure, as modeled by:

$$d\mathbf{X}_t = 0, \quad \mathbf{X}_0 = \mathbf{X}^*$$
$$d\mathbf{A}_t = 1/2\mathbf{A}_t\,dt + d\mathbf{W_A}, \tag{3.8}$$

where $\mathbf{X}^*$ is the node feature matrix that remains constant during the feature generation process.

# 4. Main Results

We next present our main results on the convergence behaviour of SGGMs across the three graph generation paradigms. We first present the convergence results for each paradigm, deferring the detailed discussion of these results until after all paradigms have been presented.

## 4.1. Convergence Results

**Theorem 4.1** (Convergence Bound of Node Feature Generation with Fixed Structure). *Consider the graph generation paradigm in Eq. 3.7. Under Assumptions 3.1 and 3.2 and supposing that the fixed graph structure $\mathbf{A}^*$ has a bounded norm, i.e., $\|\mathbf{A}^*\|^2 \leq \sigma_{\mathbf{A}}^2$, we have the following results:*

● *For exponential integration scheme,*

$$\mathrm{KL}(\mathbb{P}(\mathbf{X}_0)\|\mathbb{P}(\widehat{\mathbf{X}}_0)) \lesssim (H_{\mathbf{X}} + NF)e^{-T} + T\epsilon_{\mathbf{X}}^2$$
$$+ NFL\left(\sigma_{\mathbf{A}}^2 L\sum_{i=1}^M \Delta t_i^2 + \sum_{i=1}^M \Delta t_i^3\right). \tag{4.1}$$

● *For Euler-Maruyama scheme,*

$$\mathrm{KL}(\mathbb{P}(\mathbf{X}_0)\|\mathbb{P}(\widehat{\mathbf{X}}_0)) \lesssim (H_{\mathbf{X}} + NF)e^{-T} + T\epsilon_{\mathbf{X}}^2$$
$$+ (NFL^2\sigma_{\mathbf{A}}^2 + NF)\sum_{i=1}^M \Delta t_i^2 + (NFL + H_{\mathbf{X}})\sum_{i=1}^M \Delta t_i^3. \tag{4.2}$$

**Theorem 4.2** (Convergence Bound of Graph Structure Generation with Fixed Node Features). *Consider the graph generation paradigm in Eq. 3.8. Under Assumptions 3.1*

and *3.2 and supposing the fixed feature matrix $\mathbf{X}^*$ has a bounded norm, i.e., $\|\mathbf{X}^*\|^2 \leq \sigma_{\mathbf{X}}^2$, we have the following results:*

● *For exponential integration scheme,*

$$\mathrm{KL}(\mathbb{P}(\mathbf{A}_0)\|\mathbb{P}(\widehat{\mathbf{A}}_0)) \lesssim (H_{\mathbf{A}} + N^2)e^{-T} + T\epsilon_{\mathbf{A}}^2$$
$$+ N^2 L\left(\sigma_{\mathbf{X}}^2 L\sum_{i=1}^M \Delta t_i^2 + \sum_{i=1}^M \Delta t_i^3\right). \tag{4.3}$$

● *For Euler-Maruyama scheme,*

$$\mathrm{KL}(\mathbb{P}(\mathbf{A}_0)\|\mathbb{P}(\widehat{\mathbf{A}}_0)) \lesssim (H_{\mathbf{A}} + N^2)e^{-T} + T\epsilon_{\mathbf{A}}^2$$
$$+ (N^2 L^2\sigma_{\mathbf{X}}^2 + N^2)\sum_{i=1}^M \Delta t_i^2 + (N^2 L + H_{\mathbf{X}})\sum_{i=1}^M \Delta t_i^3. \tag{4.4}$$

**Theorem 4.3** (Convergence Bound of Joint Generation of Graph Structure and Node Features). *Consider the graph generation paradigm in Eq. 3.1. Under Assumptions 3.1 and 3.2, we have the following results:*

● *For exponential integration scheme,*

$$\mathrm{KL}(\mathbb{P}(\mathbf{X}_0)\|\mathbb{P}(\widehat{\mathbf{X}}_0)) \lesssim (H_{\mathbf{X}} + NF)e^{-T} + T\epsilon_{\mathbf{X}}^2$$
$$+ NFL\left(L\sum_{i=1}^M \Delta t_i^2 + \sum_{i=1}^M \Delta t_i^3\right),$$
$$\mathrm{KL}(\mathbb{P}(\mathbf{A}_0)\|\mathbb{P}(\widehat{\mathbf{A}}_0)) \lesssim (H_{\mathbf{A}} + N^2)e^{-T} + T\epsilon_{\mathbf{A}}^2$$
$$+ N^2 L\left(L\sum_{i=1}^M \Delta t_i^2 + \sum_{i=1}^M \Delta t_i^3\right). \tag{4.5}$$

● *For Euler-Maruyama scheme,*

$$\mathrm{KL}(\mathbb{P}(\mathbf{X}_0)\|\mathbb{P}(\widehat{\mathbf{X}}_0)) \lesssim (H_{\mathbf{X}} + N^2)e^{-T} + T\epsilon_{\mathbf{X}}^2$$
$$+ (NFL + NF)\sum_{i=1}^M \Delta t_i^2 + (NFL + H_{\mathbf{X}})\sum_{i=1}^M \Delta t_i^3,$$
$$\mathrm{KL}(\mathbb{P}(\mathbf{A}_0)\|\mathbb{P}(\widehat{\mathbf{A}}_0)) \lesssim (H_{\mathbf{A}} + N^2)e^{-T} + T\epsilon_{\mathbf{A}}^2$$
$$+ (N^2 L + N^2)\sum_{i=1}^M \Delta t_i^2 + (N^2 L + H_{\mathbf{A}})\sum_{i=1}^M \Delta t_i^3. \tag{4.6}$$

The proofs of Theorem 4.1, Theorem 4.2 and Theorem 4.3 can be found in the Appendix.

## 4.2. Discussion and Insights.

In this section, we provide a detailed discussion of the convergence results, highlighting several interesting insights and their implications.

**Source of Error.** From Theorems 4.1, 4.2, and 4.3, we identify three primary sources of generative errors in SG-GMs: 1) **Distance to convergent distribution**: represented by the first term, $(H_{\mathbf{A}} + N^2)e^{-T}$ and $(H_{\mathbf{X}} + NF)e^{-T}$, in the convergence bounds, this error quantifies how far the forward process has transformed the initial data distribution towards the convergent distribution. Interestingly, it is independent of the initial data distribution and is influenced by the dimensionality, second moments of the data distribution, and the length of the diffusion process; 2) **Score estimation error**: captured by the second term, $T\epsilon_{\mathbf{X}}^2$ and $T\epsilon_{\mathbf{A}}^2$ in the convergence bounds, this error arises from inaccuracies in estimating the underlying score functions; 3) **Discretization error**: represented by the remaining terms in the convergence bounds, this error is induced by the discretization of sampling scheme.

**Connection with Graph Data.** The convergence bounds reveal several interesting connections between the properties of the graph data and the convergence behavior of the SGGM. The scale of graph data is determined by two factors: the size of the graph $N$ and the dimension of the features $F$. Our convergence results show that these two factors affect convergence behavior in a non-isotropic manner. Specifically, we have the following remark:

*Remark* 4.4. The performance of SGGMs deteriorates more significantly as the graph size increases compared to an increase in the feature dimensionality.

This is evident from the convergence bounds, where the bounds grow quadratically with respect to graph size but only linearly with respect to feature dimensionality. This also explains why SGGMs perform well for smaller graphs, even when the features are complex (Jo et al., 2022).

Additionally, the terms $\sigma_{\mathbf{A}}$ and $\sigma_{\mathbf{X}}$, which represent bounds on the norms of the graph structure and feature matrices, are significant in the feature generation and structure generation paradigms respectively. Larger $\sigma_{\mathbf{A}}$ and $\sigma_{\mathbf{X}}$ lead to a larger risk of generative errors (i.e., larger convergence bound). Based on spectral graph theory, $\sigma_{\mathbf{A}}$ has an intrinsic connection with the graph structure. For example, $\sigma_{\mathbf{A}}$ is closely related to the maximum degree of the graph (Spielman, 2012):

$$\sigma_{\mathbf{A}} \leq \max \text{degree}(\mathcal{G}).$$

Based on this connection, we make the following remark:

*Remark* 4.5. SGGMs are better at learning and generating graphs with uniform degree distributions (more regular-like) than graphs with heterogeneous degree distributions (e.g., power-law graphs).

On the other hand, $\sigma_{\mathbf{X}}$, can be reduced with the commonly used normalization techniques, leading to a smaller convergence bound. This insight results in the following remark:

*Remark* 4.6. Applying the normalization technique to $\mathbf{X}$ in the feature generation paradigm can improve the convergence bound of SGGMs.

**Hyperparameters.** Next, we discuss the implications of the convergence results for the hyperparameters involved in SGGM learning, particularly the length of the diffusion process $T$ and the sampling step $M$. For clarity, we assume uniform discretization steps and focus on the joint generation paradigm with the exponential integrator scheme. The implications also extend to other paradigms and the Euler-Maruyama scheme (see Appendix D).

**Corollary 4.7.** *Suppose the discretization step is uniform* $\Delta t_i = T/M \leq 1, \forall i \in 1, ..., M$. *Then the convergence results of SGGMs under the exponential integrator scheme are given by*

$$\text{KL}(\mathbb{P}(\mathbf{X}_0)\|\mathbb{P}(\widehat{\mathbf{X}}_T)) \lesssim$$
$$(H_{\mathbf{X}} + NF)e^{-T} + T\epsilon_{\mathbf{X}}^2 + \frac{NFL^2T^2}{M},$$
$$\text{KL}(\mathbb{P}(\mathbf{A}_0)\|\mathbb{P}(\widehat{\mathbf{A}}_T)) \lesssim$$
$$(H_{\mathbf{A}} + N^2)e^{-T} + T\epsilon_{\mathbf{A}}^2 + \frac{N^2LT^2}{M}.$$

*Furthermore, taking*

$$T = \max\left\{\log\left(\frac{H_{\mathbf{X}} + NF}{\epsilon_{\mathbf{X}}^2}\right), \log\left(\frac{H_{\mathbf{X}} + N^2}{\epsilon_{\mathbf{A}}^2}\right)\right\},$$
$$M = \max\left\{\frac{NFL^2T^2}{\epsilon_{\mathbf{X}}^2}, \frac{N^2L^2T^2}{\epsilon_{\mathbf{A}}^2}\right\},$$

*we have that the overall generative error of SGGMs is bounded by the score estimation errors, i.e.,*

$$\text{KL}(\mathbb{P}(\mathbf{X}_0)\|\mathbb{P}(\widehat{\mathbf{X}}_T)) \lesssim \epsilon_{\mathbf{X}}^2, \quad \text{KL}(\mathbb{P}(\mathbf{A}_0)\|\mathbb{P}(\widehat{\mathbf{A}}_T)) \lesssim \epsilon_{\mathbf{A}}^2.$$

This corollary provides a clearer formulation for the convergence bounds and offers guidance for choosing the hyperparameters. It is evident that there is a trade-off in the length of the diffusion $T$: larger values of $T$ lead to smaller errors and better convergence to the target distribution, while a larger sampling step $M$ (which increases computational complexity) is required to control the sampling error.

**Sampling Scheme.** When comparing the convergence bounds under the Euler-Maruyama scheme and the exponential integrator scheme (e.g., Eq.4.1 vs. Eq.4.2), we observe that the Euler-Maruyama scheme introduces additional error terms due to higher-order discretization errors. As a result, the Euler-Maruyama scheme leads to a larger convergence bound than the exponential integrator scheme. Therefore, theoretically, the exponential integrator scheme is expected to outperform the Euler-Maruyama scheme in graph generation tasks. This finding aligns with the results in SGMs (Chen et al., 2023a).

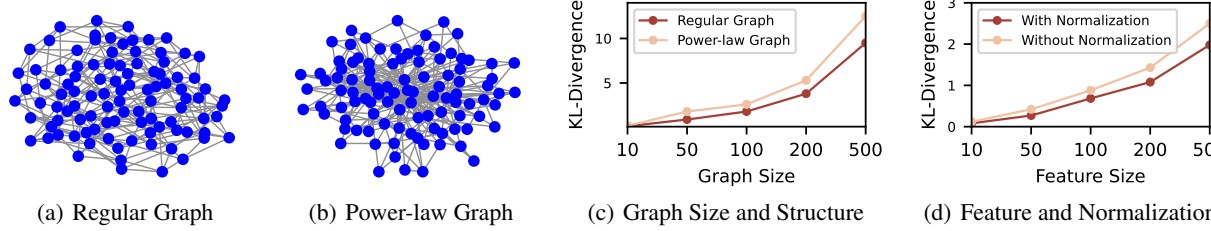

(a) Regular Graph     (b) Power-law Graph     (c) Graph Size and Structure     (d) Feature and Normalization

*Figure 1.* Performance of SGGMs. Fig. 1(a) and Fig. 1(b) are examples of regular and power-law graphs generated from our synthetic graph model. Fig. 1(c) plots the performance of SGGMs with respect to increasing graph size for regular and power-law graphs. The feature size in this experiment is fixed to 50. Fig. 1(d) plots the performance of SGGMs with respect to increasing feature size w./w.o. normalization. The graph size is fixed to 50.

## 5. Empirical Study

We next present an empirical study to validate our theoretical results. Specifically, we answer the following questions:

Q1: Is SGGM more effective at learning and generating regular graphs compared to power-law graphs in the feature generation paradigm?

Q2: Does applying normalization techniques improve the convergence of SGGM in the feature generation paradigm?

Q3: Does increasing graph size result in more significant performance degradation than increasing feature size?

**Experimental Setup.** We conduct controlled experiments using synthetic graph models, consistent with other theoretical studies, to examine how the performance of SGGMs varies with changes in graph size, graph structure (regular vs. power-law), feature size, and the application of normalization techniques. To generate power-law graphs, we employ the well-known Barabási-Albert model (Pósfai & Barabási, 2016). Node features for all experiments are drawn from a Gaussian distribution with a mean of 1 and a variance of 2 (to distinguish them from the convergent distribution). The SGGM is implemented using the hyperparameters specified in the theoretical analysis, with a diffusion length $T = 100$ and sampling steps $M = 500$, and a simple one-layer GCN as the score network. For simplicity, we use uniform discretization and the exponential integrator scheme to facilitate the sampling process. Each experiment generates 200 independent samples, which are then split into training, validation, and test sets in a 6:2:2 ratio. Each experiment consists of five independent trials, with the results averaged across these trials to ensure statistical robustness. Further technical details of the experiments and implementation can be found in Appendix E.

**Results.** The experimental results, summarized in Fig.1, provide affirmative answers to the questions (Q1-Q3) and validate our theoretical predictions. Fig.1(c) shows the performance of SGGMs in the graph structure generation

paradigm as the graph size increases. As predicted, SGGMs perform better on regular graphs compared to power-law graphs. Fig.1(d) illustrates the performance of SGGMs as the feature size increases. As predicted, applying normalization improves SGGMs' performance. Comparing the trends in Fig.1(c) and Fig. 1(d), we observe that SGGMs' performance degrades more rapidly with increasing graph size than with increasing feature size.

## 6. Concluding Discussions

This paper presents a novel convergence analysis for SGGMs across three common graph generation paradigms. Our analysis identifies the primary sources of generative error in SGGMs and provides valuable insights into the factors unique to graph data that influence their convergence behavior. Specifically, we examine how graph size, structure, and feature dimensionality affect the convergence bound. Additionally, we offer practical recommendations for improving model performance, including the use of normalization and the selection of key hyperparameters, such as diffusion length and sampling step size. Our empirical study using synthetic graph data validates the theoretical predictions. Overall, this work advances the theoretical foundation of score-based generative models, confirms the applicability of SGGMs in critical applications, and provides actionable insights for their effective use in graph generation tasks.

### 6.1. Future Works

**Learning Process.** Our current analysis assumes a static outcome of the learning process, as specified in Assumption 3.2. However, in practice, the estimation (or learning) of score functions is subject to errors that accumulate over time as the score network learns from data. These errors are influenced by the specific behavior of the learning algorithm. Future work could incorporate the dynamics of the learning process in SGGMs, as explored in studies such as (Chen et al., 2022a;b; Benton et al., 2024; Zhu et al., 2023), to gain a more nuanced understanding of SGGMs' behavior. This

would be especially valuable for addressing challenges like sample complexity in SGGMs and for providing insights into their empirical performance.

**More Precise Analysis with Synthetic Graph Model.** The analysis presented in this paper does not assume any specific structure for the data distribution, making it applicable to general smooth and bounded distributions. However, to make the analysis more precise, one could impose additional structural assumptions on the underlying data distribution, similar to case studies with Gaussian mixture models in SGM research (Chen et al., 2024b; Shah et al., 2023). A natural extension for graph data would be to assume that the graph is generated from the contextual stochastic block model (Deshpande et al., 2018). Integrating this modeling choice could lead to more accurate convergence bounds and provide valuable insights that would require a more detailed, fine-grained analysis.

## Impact Statement

This paper presents work whose goal is to advance the understanding of score-based graph generation methods. There are many potential societal consequences of our work, none which we feel must be specifically highlighted here.

## Acknowledgement

We would like to thank the anonymous reviewers and area chairs for their helpful comments. This work was supported in part by grants from Hong Kong RGC under the contracts 17207621, 17203522, and C7004-22G (CRF).

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

# A. Problem Formulation Discussion

In this appendix, we provide a further discussion on our problem formulation, including the choice of hyper-parameter and the derivation of the training objectives.

## A.1. Choice of Hyper-parameter

For our analysis, we have chosen the following set of hyper-parameter:

$$f_{\mathbf{X}}(\mathcal{G}_t, t) = -1/2 g_{\mathbf{X}}(\mathcal{G}_t, t)^2 \mathbf{X}_t,$$
$$f_{\mathbf{A}}(\mathcal{G}_t, t) = -1/2 g_{\mathbf{A}}(\mathcal{G}_t, t)^2 \mathbf{A}_t,$$
$$g_{\mathbf{X}}(\mathcal{G}_t, t) = g_{\mathbf{A}}(\mathcal{G}_t, t) \equiv 1. \tag{A.1}$$

As mentioned in the main paper, our analysis can be generalize further to any linear drift function $f_{\mathbf{X}}(\mathcal{G}_t, t)$ and $f_{\mathbf{A}}(\mathcal{G}_t, t)$ and other constant variance functions so long as the underlying process remain a OU process and would converge to the prior distribution (i.e., the convergent distribution)

Without loss of generality, we can express the set of linear function and constant variance function as,

$$f_{\mathbf{X}}(\mathcal{G}_t, t) = \xi g_{\mathbf{X}}(\mathcal{G}_t, t)^2 \mathbf{X}_t,$$
$$f_{\mathbf{A}}(\mathcal{G}_t, t) = \xi g_{\mathbf{A}}(\mathcal{G}_t, t)^2 \mathbf{A}_t,$$
$$g_{\mathbf{X}}(\mathcal{G}_t, t) = g_{\mathbf{A}}(\mathcal{G}_t, t) \equiv \kappa, \tag{A.2}$$

where $\xi \in (-1, 0)$ and $\kappa$ is some positive constant.

The selection of a constant variance function does not result in any loss of generality. This is because altering the variance function can be viewed as a re-scaling of time, provided that the drift function does not explicitly depend on time. In other words, changing the variance function only affects the rate at which the diffusion process evolves, but this transformation is effectively equivalent to adjusting the temporal scale of the process. Consequently, the analysis remains valid even if the variance function is modified, as long as the time re-scaling is appropriately accounted for.

By changing the coefficient in the linear drift function, this amounts to a different conditional density for the forward process and score:

$$\mathbf{X}_t | \mathbf{X}_0 \sim \mathbb{N}(e^{\xi t} \mathbf{X}_0, (1 - e^{-2\xi t}) \mathbf{I}_{N \times F}),$$
$$\mathbf{A}_t | \mathbf{A}_0 \sim \mathbb{N}(e^{\xi t} \mathbf{A}_0, (1 - e^{-2\xi t}) \mathbf{I}_{N \times N}). \tag{A.3}$$

Since all the theoretical tools we use for the analysis are grounded in the OU process and independent of the scale of the hyper-parameters, all the results and analysis can be immediately applied with the general formulation above. However, in the paper, we strike for cleaner results for better interpretability and focus on the set of hyper-parameters specified in the paper.

## A.2. Training Objective Derivation

In this appendix, we present a derivation for the Equations used in the problem formulation for completeness. A similar derivation can be found in (Jo et al., 2022).

The partial score functions can be estimated by training the time-dependent score-based models $s_\theta(.)$ and $s_\phi(.)$, so that

$$s_\theta(\mathcal{G}_t, t) \approx \nabla_{\mathbf{X}} \log \mathbb{P}(\mathcal{G}_t), s_\phi(\mathcal{G}_t, t) \approx \nabla_{\mathbf{A}} \log \mathbb{P}(\mathcal{G}_t).$$

However, the objectives introduced in SGM for estimating the score function are not directly applicable here, since the partial score functions are defined as the gradient of each component, rather than the gradient of the data as in the conventional score function. This interdependence between the two diffusion processes tied by the partial scores adds another layer of difficulty.

To address this issue, an new objective for estimating the partial scores is needed. Intuitively, the score-based models should be trained to minimize the distance to the corresponding ground-truth partial scores. The following new objectives

generalize score matching (Song et al., 2020b) to the estimation of partial scores for the given graph dataset, as follows:

$$\min_{\theta} \mathbb{E}_t \left[ \mathbb{E}_{\mathcal{G}_0} \mathbb{E}_{\mathcal{G}_t | \mathcal{G}_0} \big\| s_{\theta,t}(\mathcal{G}_t) - \nabla_{\mathbf{X}} \log \mathbb{P}(\mathcal{G}_t) \big\|_2^2 \right], \tag{A.4}$$

$$\min_{\phi} \mathbb{E}_t \left[ \mathbb{E}_{\mathcal{G}_0} \mathbb{E}_{\mathcal{G}_t | \mathcal{G}_0} \big\| s_{\phi,t}(\mathcal{G}_t) - \nabla_{\mathbf{A}} \log \mathbb{P}(\mathcal{G}_t) \big\|_2^2 \right], \tag{A.5}$$

where $t$ is uniformly sampled from $[0, T]$. The expectations are taken over samples $\mathcal{G}_0 \sim p_{\text{data}}$ and $\mathcal{G}_t \sim \mathbb{P}(\mathcal{G}_t | \mathcal{G}_0)$, where $\mathbb{P}(\mathcal{G}_t | \mathcal{G}_0)$ denotes the transition distribution from $0$ to $t$ induced by the forward diffusion process.

Unfortunately, the equations above are still not directly trainable since the ground-truth partial scores are not analytically accessible in general. This is why we need to underlying process to be an OU process, as we can leverage the known conditional density of OU process for training.

$$\min_{\theta} \mathbb{E}_t \left[ \mathbb{E}_{\mathcal{G}_0} \mathbb{E}_{\mathcal{G}_t | \mathcal{G}_0} \big\| s_{\theta,t}(\mathcal{G}_t, t) - \nabla_{\mathbf{X}} \log \mathbb{P}(\mathcal{G}_t | \mathcal{G}_0) \big\|_2^2 \right], \tag{A.6}$$

$$\min_{\phi} \mathbb{E}_t \left[ \mathbb{E}_{\mathcal{G}_0} \mathbb{E}_{\mathcal{G}_t | \mathcal{G}_0} \big\| s_{\phi}(\mathcal{G}_t, t) - \nabla_{\mathbf{A}} \log \mathbb{P}(\mathcal{G}_t | \mathcal{G}_0) \big\|_2^2 \right]. \tag{A.7}$$

Since the drift coefficient of the forward diffusion process is linear, the transition distribution $\mathbb{P}(\mathcal{G}_t | \mathcal{G}_0)$ can be separated in terms of $\mathbf{X}_t$ and $\mathbf{A}_t$ as follows:

$$\mathbb{P}(\mathcal{G}_t | \mathcal{G}_0) = \mathbb{P}(\mathbf{X}_t | \mathbf{X}_0) \mathbb{P}(\mathbf{A}_t | \mathbf{A}_0). \tag{A.8}$$

Notably, we can easily sample from the transition distributions of each component, $\mathbb{P}(\mathbf{X}_t | \mathbf{X}_0)$ and $\mathbb{P}(\mathbf{A}_t | \mathbf{A}_0)$, as they are Gaussian distributions with mean and variance determined by the coefficients of the forward diffusion process. This leads to the following training objective:

$$\min_{\theta} \mathbb{E}_t \left[ \mathbb{E}_{\mathcal{G}_0} \mathbb{E}_{\mathcal{G}_t | \mathcal{G}_0} \big\| s_{\boldsymbol{\theta}}(\mathcal{G}_t, t) - \nabla_{\mathbf{X}} \log \mathbb{P}(\mathbf{X}_t | \mathbf{X}_0) \big\|_2^2 \right], \tag{A.9}$$

$$\min_{\phi} \mathbb{E}_t \left[ \mathbb{E}_{\mathcal{G}_0} \mathbb{E}_{\mathcal{G}_t | \mathcal{G}_0} \big\| s_{\boldsymbol{\phi}}(\mathcal{G}_t, t) - \nabla_{\mathbf{A}} \log \mathbb{P}(\mathbf{A}_t | \mathbf{A}_0) \big\|_2^2 \right]. \tag{A.10}$$

The expectations in the equation above can be efficiently computed using the Monte Carlo estimate with the samples $(t, \mathcal{G}_0, \mathcal{G}_t)$. Note that estimating the partial scores is not equivalent to estimating $\nabla_{\mathbf{X}} \log \mathbb{P}(\mathbf{X}_t)$ or $\nabla_{\mathbf{A}} \log \mathbb{P}(\mathbf{A}_t)$, the main objective of previous score-based generative models, since estimating the partial scores requires capturing the dependency between $\mathbf{X}_t$ and $\mathbf{A}_t$ determined by the joint probability through time.

### A.2.1. DERIVATION OF TRAINING OBJECTIVE A.4

The original score matching objective can be written as follows:

$$\mathbb{E}_{\mathcal{G}_t} \left[ \| s_{\boldsymbol{\theta}}(\mathcal{G}_t, t) - \nabla_{\mathbf{X}} \log \mathbb{P}(\mathcal{G}_t) \|_2^2 \right] = \mathbb{E}_{\mathcal{G}_t} \left[ \| s_{\boldsymbol{\theta}}(\mathcal{G}_t, t) \|_2^2 \right] - 2 \mathbb{E}_{\mathcal{G}_t} \left[ \langle s_{\boldsymbol{\theta}}(\mathcal{G}_t, t), \nabla_{\mathbf{X}} \log \mathbb{P}(\mathcal{G}_t) \rangle \right] + C_1,$$

where $C_1$ is a constant that does not depend on $\mathbf{W}$. On the other hand, we have

$$\mathbb{E}_{\mathcal{G}_t} \mathbb{E}_{\mathcal{G}_t | \mathcal{G}_0} \left[ \| s_{\boldsymbol{\theta}}(\mathcal{G}_t, t) - \nabla_{\mathbf{X}} \log \mathbb{P}(\mathcal{G}_t | \mathcal{G}_0) \|_2^2 \right] = \mathbb{E}_{\mathcal{G}_t} \mathbb{E}_{\mathcal{G}_t | \mathcal{G}_0} \left[ \| s_{\boldsymbol{\theta}}(\mathcal{G}_t, t) \|_2^2 \right] - 2 \mathbb{E}_{\mathcal{G}_t} \mathbb{E}_{\mathcal{G}_t | \mathcal{G}_0} \left[ \langle s_{\boldsymbol{\theta}}(\mathcal{G}_t, t), \nabla_{\mathbf{X}} \log \mathbb{P}(\mathcal{G}_t | \mathcal{G}_0) \rangle \right] + C_2,$$

For the second term, from the derivation (Appendix A.1 from (Jo et al., 2022)), we know that it has the following equivalency:

$$\mathbb{E}_{\mathcal{G}_t} \left[ \langle s_{\boldsymbol{\theta}}(\mathcal{G}_t, t), \nabla_{\mathbf{X}} \log \mathbb{P}(\mathcal{G}_t) \rangle \right] = \mathbb{E}_{\mathcal{G}_t} \mathbb{E}_{\mathcal{G}_t | \mathcal{G}_0} \left[ \langle s_{\boldsymbol{\theta}}(\mathcal{G}_t, t), \nabla_{\mathbf{X}} \log \mathbb{P}(\mathcal{G}_t | \mathcal{G}_0) \rangle \right]$$

Since the constant $C_1$ and $C_2$ does not affect the optimization results, we can conclude that the following two objectives are equivalent with respect to $\boldsymbol{\theta}$

$$\mathbb{E}_{\mathcal{G}_t}\mathbb{E}_{\mathcal{G}_t|\mathcal{G}_0}\left[\|s_{\boldsymbol{\theta}}(\mathcal{G}_t, t) - \nabla_{\mathbf{X}}\log\mathbb{P}(\mathcal{G}_t|\mathcal{G}_0)\|_2^2\right]$$
$$\mathbb{E}_{\mathcal{G}_t}\left[\|s_{\boldsymbol{\theta}}(\mathcal{G}_t, t) - \nabla_{\mathbf{X}}\log\mathbb{P}(\mathcal{G}_t)\|_2^2\right]$$

Similarly, computing the gradient with respect to $\mathbf{A}$, we can show that the following two objectives are also equivalent with respect to $\boldsymbol{\phi}$:

$$\mathbb{E}_{\mathcal{G}_t}\mathbb{E}_{\mathcal{G}_t|\mathcal{G}_0}\left[\|s_{\boldsymbol{\phi}}(\mathcal{G}_t, t) - \nabla_{\mathbf{A}}\log\mathbb{P}(\mathcal{G}_t|\mathcal{G}_0)\|_2^2\right]$$
$$\mathbb{E}_{\mathcal{G}_t}\left[\|s_{\boldsymbol{\phi}}(\mathcal{G}_t, t) - \nabla_{\mathbf{A}}\log\mathbb{P}(\mathcal{G}_t)\|_2^2\right]$$

Now, it remains to show that $\nabla_{\mathbf{X}}\log\mathbb{P}(\mathcal{G}_t|\mathcal{G}_0)$ is equivalent to $\nabla_{\mathbf{X}}\log\mathbb{P}(\mathbf{X}_t|\mathbf{X}_0)$. Using the chain rule, we get that

$$\frac{\partial\log\mathbb{P}(\mathbf{A}_t|\mathbf{A}_0)}{\partial(\mathbf{X}_t)_{ij}} = \mathrm{Tr}\left[\nabla_{\mathbf{A}}\log\mathbb{P}(\mathbf{A}_t|\mathbf{A}_0)\frac{\partial\mathbf{A}_t}{\partial(\mathbf{X}_t)_{ij}}\right] = 0.$$

With this result, we have that,

$$\nabla_{\mathbf{X}}\log\mathbb{P}(\mathcal{G}_t|\mathcal{G}_0) = \nabla_{\mathbf{X}}\log\mathbb{P}(\mathbf{X}_t|\mathbf{X}_0) + \nabla_{\mathbf{X}}\log\mathbb{P}(\mathbf{A}_t|\mathbf{A}_0) = \nabla_{\mathbf{X}}\log\mathbb{P}(\mathbf{X}_t|\mathbf{X}_0).$$

Therefore, we can conclude that

$$\nabla_{\mathbf{X}}\log\mathbb{P}(\mathcal{G}_t|\mathcal{G}_0) = \nabla_{\mathbf{X}}\log\mathbb{P}(\mathbf{X}_t|\mathbf{X}_0)$$

With a similar computation for $\mathbf{A}_t$, we can also show that $\nabla_{\mathbf{A}}\log\mathbb{P}(\mathcal{G}_t|\mathcal{G}_0)$ is equal to $\nabla_{\mathbf{A}}\log\mathbb{P}(\mathbf{A}_t|\mathbf{A}_0)$.

### A.2.2. TRACTABLE TRAINING OBJECTIVE

In the previous section, we have proved the equivalence of the training objective used in our analysis and the common score-based generative model. It remains to show how we compute this training objective with tractable objects. To simplify the notation, we define the following for the rest of appendix, for any $0 \le t \le s \le T$

$$\sigma_t^2 := 1 - e^{-t},$$
$$\alpha_{t,s} := e^{-1/2(s-t)},$$
$$\alpha_t := \alpha_{0,t}.$$

Then, using the ideas in (Vincent, 2011), we get the following,

$$\mathbb{E}\|s_{\boldsymbol{\theta}}(\mathcal{G}_t, t) - \nabla_{\mathbf{X}}\mathbb{P}(\mathcal{G}_t)\|^2$$
$$= \mathbb{E}\|s_{\boldsymbol{\theta}}(\mathcal{G}_t, t)\|^2 + \mathbb{E}\|\nabla_{\mathbf{X}}\mathbb{P}(\mathcal{G}_t)\|^2 - 2\mathbb{E}\langle s_{\boldsymbol{\theta}}(\mathcal{G}_t, t), \nabla_{\mathbf{X}}\log\mathbb{P}(\mathcal{G}_t)\rangle$$
$$= \mathbb{E}\|s_{\boldsymbol{\theta}}(\mathcal{G}_t, t)\|^2 + \mathbb{E}\|\nabla_{\mathbf{X}}\mathbb{P}(\mathcal{G}_t)\|^2 - 2\mathbb{E}\nabla\cdot s_{\boldsymbol{\theta}}(\mathcal{G}_t, t)$$
$$= \mathbb{E}\|s_{\boldsymbol{\theta}}(\mathcal{G}_t, t)\|^2 + \mathbb{E}\|\nabla_{\mathbf{X}}\mathbb{P}(\mathcal{G}_t)\|^2 - 2\mathbb{E}_{\mathbb{P}(\mathcal{G}_0)}\mathbb{E}_{\mathbb{P}(\mathcal{G}_t|\mathcal{G}_0)}\nabla\cdot s_{\boldsymbol{\theta}}(\mathcal{G}_t, t)$$
$$= \mathbb{E}\|s_{\boldsymbol{\theta}}(\mathcal{G}_t, t)\|^2 + \mathbb{E}\|\nabla_{\mathbf{X}}\mathbb{P}(\mathcal{G}_t)\|^2 - 2\mathbb{E}_{\mathbb{P}(\mathcal{G}_0)}\mathbb{E}_{\mathbb{P}(\mathcal{G}_t|\mathcal{G}_0)}\langle\nabla_{\mathbf{X}}\log\mathbb{P}(\mathcal{G}_t|\mathcal{G}_0), s_{\boldsymbol{\theta}}(\mathcal{G}_t, t)\rangle$$
$$= \mathbb{E}\|s_{\boldsymbol{\theta}}(\mathcal{G}_t, t)\|^2 + \mathbb{E}\|\nabla_{\mathbf{X}}\mathbb{P}(\mathcal{G}_t)\|^2 - 2\mathbb{E}_{\mathbb{P}(\mathcal{G}_0)}\mathbb{E}_{\mathbb{P}(\mathcal{G}_t|\mathcal{G}_0)}\langle\nabla_{\mathbf{X}}\log\mathbb{P}(\mathbf{X}_t|\mathbf{X}_0), s_{\boldsymbol{\theta}}(\mathcal{G}_t, t)\rangle$$
$$= \mathbb{E}\|s_{\boldsymbol{\theta}}(\mathcal{G}_t, t)\|^2 + \mathbb{E}\|\nabla_{\mathbf{X}}\mathbb{P}(\mathcal{G}_t)\|^2 - 2\mathbb{E}_{\mathbb{P}(\mathcal{G}_0)}\mathbb{E}_{\mathbb{P}(\mathcal{G}_t|\mathcal{G}_0)}\left\langle\frac{\mathbf{X}_t - \alpha_t\mathbf{X}_0}{\sigma_t^2}, s_{\boldsymbol{\theta}}(\mathcal{G}_t, t)\right\rangle$$
$$= \mathbb{E}\left\|s_{\boldsymbol{\theta}}(\mathcal{G}_t, t) - \frac{\mathbf{X}_t - \alpha_t\mathbf{X}_0}{\sigma_t^2}\right\|^2 + \mathbb{E}\|\nabla_{\mathbf{X}}\mathbb{P}(\mathcal{G}_t)\|^2 - \frac{d}{\sigma_t^2}$$
$$= \mathbb{E}\left\|s_{\boldsymbol{\theta}}(\mathcal{G}_t, t) - \frac{\mathbf{X}_t - \alpha_t\mathbf{X}_0}{\sigma_t^2}\right\|^2 + C_3$$

where $C_3$ is some constant independent of $\boldsymbol{\theta}$. Therefore, we can use the last equation as the training objective. Through a similar computation, we get that,

$$\mathbb{E}\|s_{\boldsymbol{\phi}}(\mathcal{G}_t, t) - \nabla_{\mathbf{A}}\mathbb{P}(\mathcal{G}_t)\|^2 = \mathbb{E}\left\|s_{\boldsymbol{\phi}}(\mathcal{G}_t, t) - \frac{\mathbf{A}_t - \alpha_t \mathbf{A}_0}{\sigma_t^2}\right\|^2 + C_4$$

### A.2.3. DISCUSSION OF ASSUMPTION 3.2

In this section, we present an argument for why Assumption 3.2 is reasonable and hold in practice. We can observe that

$$\mathbb{E}\left\|\frac{\mathbf{X} - \alpha_t \mathbf{X}_0}{\sigma_t^2}\right\| = \frac{1}{\sigma_t^2}.$$

Therefore, it is natural to expect the error to scale as

$$\mathbb{E}\left\|s_{\boldsymbol{\theta}}(\mathcal{G}_t, t) - \nabla_{\mathbf{X}}\log\mathbb{P}(\mathcal{G}_t)\right\|^2 \lesssim \frac{\delta_{\mathbf{X}}^2}{\sigma_t^2}.$$

for some $\delta_{\mathbf{X}}$. Furthermore, notice that $\sigma_{t_k}^2 \simeq \min\{t_k, 1\}$, then we have,

$$\frac{1}{T}\sum_{i=1}^{T}\Delta_{t_i}\mathbb{E}\|s_{\boldsymbol{\theta}}(\mathcal{G}_t, t) - \nabla_{\mathbf{X}}\log\mathbb{P}(\mathcal{G}_t)\|^2 \lesssim \frac{1}{T}\int_{t_1}^{T}\frac{\delta_{\mathbf{X}}^2}{t \wedge 1}\mathrm{d}t \lesssim \delta_{\mathbf{X}}^2\log(1/t_1).$$

Through a symmetric argument, we can get a similar result for the structure process.

## B. Useful Lemmas

In this appendix, we present a set of useful results that are going to be used in the proof of the main theorem. The proof of some of the results can also be found in other diffusion convergent analysis (Chen et al., 2023a; 2022a; Zhu et al., 2023; Lee et al., 2023b; Gnaneshwar et al., 2022). We present the proof for these results for completeness.

To simplify the notation a bit, in the following, we use $\mathbb{P}_t$ and $\mathbb{P}(\mathbf{X}_t)$ interchangeably to represent the density of $\mathbf{X}$ at time $t$. We use $\mathbb{P}_{t|t'}$ and $\mathbb{P}(\mathbf{X}_t|\mathbf{X}_{t'})$ interchangeably to represent the condition density of $\mathbf{X}$ at time $t$ given $t'$. In addition, we adopt the Frobenius norm as the matrix norm.

**Lemma B.1.** *Given two Itô processes coupled by the same initial condition and random noise as follows,*

$$\mathrm{d}\mathcal{X}_t = f_1(\mathcal{X}_t, t)\mathrm{d}t + g(t)\mathrm{d}\mathbf{W}_t, \quad \mathcal{X}_0 = \boldsymbol{\gamma},$$
$$\mathrm{d}\mathcal{X}_t' = f_2(\mathcal{X}_t', t)\mathrm{d}t + g(t)\mathrm{d}\mathbf{W}_t, \quad \mathcal{X}_0' = \boldsymbol{\gamma},$$

*where $f_1, f_2, g$ are continuous function. Furthermore, suppose the two SDEs satisfy the following conditions,*

1. *the two SDEs have unique solutions,*

2. *$\mathcal{X}_t, \mathcal{X}_t'$ admit densities $\mathbb{P}_t, \mathbb{Q}_t$ that are twice continuously differentiable with respect to inputs $t > 0$.*

*Then, we denote the relative Fisher information between $\mathbb{P}_t$ and $\mathbb{Q}_t$ by*

$$J(\mathbb{P}_t\|\mathbb{Q}_t) = \int \mathbb{P}_t(\mathcal{X})\left\|\nabla\log\frac{\mathbb{P}_t}{\mathbb{Q}_t}\right\|^2 \mathrm{d}\mathbf{X}.$$

*Then for any $t > 0$, the time derivative of $\mathrm{KL}(\mathbb{P}_t\|\mathbb{Q}_t)$ is given by,*

$$\frac{\mathrm{d}}{\mathrm{d}t}\mathrm{KL}(\mathbb{P}_t\|\mathbb{Q}_t) = -g(t)^2 J(\mathbb{P}_t\|\mathbb{Q}_t) + \mathbb{E}\left[\left\langle f_1(\mathcal{X}_t, t) - f_2(\mathcal{X}_t, t), \nabla\log\frac{\mathbb{P}_t}{\mathbb{Q}_t}\right\rangle\right]$$

*Proof.* By definition of KL divergence, we have that

$$\mathrm{KL}(\mathbb{P}_t \| \mathbb{Q}_t) = \int \mathbb{P}_t(\mathcal{X}) \log\left(\frac{\mathbb{P}_t(\mathcal{X})}{\mathbb{Q}_t(\mathcal{X})}\right) d\mathcal{X}.$$

Taking the time derivative of the expression above, we obtain,

$$\frac{\partial}{\partial t}\mathrm{KL}(\mathbb{P}_t \| \mathbb{Q}_t) = \frac{\partial}{\partial t}\left[\int \mathbb{P}_t(\mathcal{X}) \log\left(\frac{\mathbb{P}_t(\mathcal{X})}{\mathbb{Q}_t(\mathcal{X})}\right) d\mathcal{X}\right]$$

$$= \int \frac{\partial \mathbb{P}_t(\mathcal{X})}{\partial t} \log\left(\frac{\mathbb{P}_t(\mathcal{X})}{\mathbb{Q}_t(\mathcal{X})}\right) d\mathcal{X} - \int \frac{\mathbb{P}_t(\mathcal{X})}{\mathbb{Q}_t(\mathcal{X})} \frac{\partial \mathbb{Q}_t(\mathcal{X})}{\partial t} d\mathcal{X}$$

Then, we can use Fokker-Plank equation to obtain the time derivatives of $\mathbb{P}_t$ and $\mathbb{Q}_t$ which are given by

$$\frac{\partial \mathbb{P}_t(\mathcal{X})}{\partial \mathcal{X}} = \nabla \cdot \left[-f_1(\mathcal{X}, t)\mathbb{P}_t(\mathcal{X}) + \frac{g(t)^2}{2}\nabla \mathbb{P}_t(\mathcal{X})\right],$$

$$\frac{\partial \mathbb{Q}_t(\mathcal{X})}{\partial \mathcal{X}} = \nabla \cdot \left[-f_2(\mathcal{X}, t)\mathbb{Q}_t(\mathcal{X}) + \frac{g(t)^2}{2}\nabla \mathbb{Q}_t(\mathcal{X})\right].$$

Substituting the time derivatives of $\mathbb{P}_t$ and $\mathbb{Q}_t$ into the corresponding terms of the time derivative of $\frac{\partial}{\partial t}\mathrm{KL}(\mathbb{P}_t \| \mathbb{Q}_t)$, we get that,

$$\int \frac{\partial \mathbb{P}_t(\mathcal{X})}{\partial t} \log\left(\frac{\mathbb{P}_t(\mathcal{X})}{\mathbb{Q}_t(\mathcal{X})}\right) d\mathcal{X} = \int \nabla \cdot \left[-f_1(\mathcal{X}, t)\mathbb{P}_t(\mathcal{X}) + \frac{g(t)^2}{2}\nabla \mathbb{P}_t(\mathcal{X})\right] \log\left(\frac{\mathbb{P}_t(\mathcal{X})}{\mathbb{Q}_t(\mathcal{X})}\right) d\mathcal{X}$$

$$= \int \left\langle \nabla \log\left(\frac{\mathbb{P}_t(\mathcal{X})}{\mathbb{Q}_t(\mathcal{X})}\right), \mathbb{P}_t(\mathcal{X})f_1(\mathcal{X}, t) - \frac{g(t)^2}{2}\nabla \mathbb{P}_t(\mathcal{X})\right\rangle d\mathcal{X},$$

$$= \int \mathbb{P}_t(\mathcal{X}) \left\langle f_1(\mathcal{X}, t), \nabla \log\left(\frac{\mathbb{P}_t(\mathcal{X})}{\mathbb{Q}_t(\mathcal{X})}\right)\right\rangle - \frac{g(t)^2}{2} \left\langle \nabla \log\left(\frac{\mathbb{P}_t(\mathcal{X})}{\mathbb{Q}_t(\mathcal{X})}\right), \nabla \mathbb{P}_t(\mathcal{X})\right\rangle d\mathcal{X}.$$

$$\int \frac{\mathbb{P}_t(\mathcal{X})}{\mathbb{Q}_t(\mathcal{X})} \frac{\partial \mathbb{Q}_t(\mathcal{X})}{\partial t} d\mathcal{X} = \int \frac{\mathbb{P}_t(\mathcal{X})}{\mathbb{Q}_t(\mathcal{X})} \nabla \cdot \left[-f_2(\mathcal{X}, t)\mathbb{Q}_t(\mathcal{X}) + \frac{g(t)^2}{2}\nabla \mathbb{Q}_t(\mathcal{X})\right] d\mathcal{X},$$

$$= \int \left\langle \nabla \frac{\mathbb{P}_t(\mathcal{X})}{\mathbb{Q}_t(\mathcal{X})}, f_2(\mathcal{X}, t)\mathbb{Q}_t(\mathcal{X}) - \frac{g(t)^2}{2}\nabla \mathbb{Q}_t(\mathcal{X})\right\rangle,$$

$$= \int \mathbb{Q}_t(\mathcal{X}) \left\langle \nabla \frac{\mathbb{P}_t}{\mathbb{Q}_t}, f_2(\mathcal{X}, t)\right\rangle - \frac{g(t)^2}{2}\left\langle \nabla \frac{\mathbb{P}_t(\mathcal{X})}{\mathbb{Q}_t(\mathcal{X})}, \nabla \mathbb{Q}_t(\mathcal{X})\right\rangle d\mathcal{X}.$$

Combining the results above, we obtain,

$$\frac{\partial}{\partial t}\mathrm{KL}(\mathbb{P}_t \| \mathbb{Q}_t) = \frac{\partial}{\partial t}\left[\int \mathbb{P}_t(\mathcal{X}) \log\left(\frac{\mathbb{P}_t(\mathcal{X})}{\mathbb{Q}_t(\mathcal{X})}\right) d\mathcal{X}\right],$$

$$= \int \frac{\partial \mathbb{P}_t(\mathcal{X})}{\partial t} \log\left(\frac{\mathbb{P}_t(\mathcal{X})}{\mathbb{Q}_t(\mathcal{X})}\right) d\mathcal{X} - \int \frac{\mathbb{P}_t(\mathcal{X})}{\mathbb{Q}_t(\mathcal{X})} \frac{\partial \mathbb{Q}_t(\mathcal{X})}{\partial t} d\mathcal{X},$$

$$= \int \mathbb{P}_t(\mathcal{X}) \left\langle f_1(\mathcal{X}, t), \nabla \log\left(\frac{\mathbb{P}_t(\mathcal{X})}{\mathbb{Q}_t(\mathcal{X})}\right)\right\rangle - \frac{g(t)^2}{2}\left\langle \nabla \log\left(\frac{\mathbb{P}_t(\mathcal{X})}{\mathbb{Q}_t(\mathcal{X})}\right), \nabla \mathbb{P}_t(\mathcal{X})\right\rangle d\mathcal{X} - ...$$

$$\int \mathbb{Q}_t(\mathcal{X}) \left\langle \nabla \frac{\mathbb{P}_t}{\mathbb{Q}_t}, f_2(\mathcal{X}, t)\right\rangle + \frac{g(t)^2}{2}\left\langle \nabla \frac{\mathbb{P}_t(\mathcal{X})}{\mathbb{Q}_t(\mathcal{X})}, \nabla \mathbb{Q}_t(\mathcal{X})\right\rangle d\mathcal{X},$$

$$= \frac{g(t)^2}{2}\left(\int \left\langle \nabla \frac{\mathbb{P}_t(\mathcal{X})}{\mathbb{Q}_t(\mathcal{X})}, \nabla \mathbb{Q}_t(\mathcal{X})\right\rangle - \left\langle \nabla \log\left(\frac{\mathbb{P}_t(\mathcal{X})}{\mathbb{Q}_t(\mathcal{X})}\right), \nabla \mathbb{P}_t(\mathcal{X})\right\rangle d\mathcal{X}\right) + ...$$

$$\int \mathbb{P}_t(\mathcal{X}) \left\langle f_1(\mathcal{X}, t), \nabla \log \frac{\mathbb{P}_t(\mathcal{X})}{\mathbb{Q}_t(\mathcal{X})}\right\rangle - \mathbb{Q}_t(\mathcal{X})\left\langle \nabla \frac{\mathbb{P}_t(\mathbf{X})}{\mathbb{Q}_t(\mathbf{X})}, f_2(\mathcal{X}, t)\right\rangle d\mathcal{X},$$

Notice that

$$\int \left\langle \nabla \frac{\mathbb{P}_t(\mathcal{X})}{\mathbb{Q}_t(\mathcal{X})}, \nabla \mathbb{Q}_t(\mathcal{X}) \right\rangle - \left\langle \nabla \log \left( \frac{\mathbb{P}_t(\mathcal{X})}{\mathbb{Q}_t(\mathcal{X})} \right), \nabla \mathbb{P}_t(\mathcal{X}) \right\rangle d\mathcal{X}$$

$$= \int \left\langle \frac{\mathbb{Q}_t(\mathcal{X}) \nabla \mathbb{P}_t(\mathcal{X}) - \mathbb{P}_t(\mathcal{X}) \nabla \mathbb{Q}_t(\mathcal{X})}{\mathbb{Q}_t(\mathcal{X})}, \nabla \log \mathbb{Q}_t(\mathcal{X}) \right\rangle - \mathbb{P}_t \left\langle \nabla \log \frac{\mathbb{P}_t(\mathcal{X})}{\mathbb{Q}_t(\mathcal{X})}, \nabla \log \mathbb{P}_t(\mathcal{X}) \right\rangle,$$

$$= \int \mathbb{P}_t(\mathcal{X}) \left\langle \nabla \log \frac{\mathbb{P}_t(\mathcal{X})}{\mathbb{Q}_t(\mathcal{X})}, \nabla \log \mathbb{Q}_t(\mathcal{X}) \right\rangle - \mathbb{P}_t \left\langle \nabla \log \frac{\mathbb{P}_t(\mathcal{X})}{\mathbb{Q}_t(\mathcal{X})}, \nabla \log \mathbb{P}_t(\mathcal{X}) \right\rangle d\mathcal{X},$$

$$= -\mathrm{J}(\mathbb{P}_t(\mathcal{X}) \| \mathbb{Q}_t(\mathcal{X})).$$

In addition, we have

$$\int \mathbb{P}_t(\mathcal{X}) \left\langle f_1(\mathcal{X}, t), \nabla \log \frac{\mathbb{P}_t(\mathcal{X})}{\mathbb{Q}_t(\mathcal{X})} \right\rangle - \mathbb{Q}_t(\mathcal{X}) \left\langle \nabla \frac{\mathbb{P}_t(\mathbf{X})}{\mathbb{Q}_t(\mathbf{X})}, f_2(\mathcal{X}, t) \right\rangle d\mathcal{X}$$

$$= \int \mathbb{P}_t(\mathcal{X}) \left\langle f_1(\mathcal{X}, t), \nabla \log \frac{\mathbb{P}_t(\mathcal{X})}{\mathbb{Q}_t(\mathcal{X})} \right\rangle - \mathbb{P}_t(\mathcal{X}) \left\langle \nabla \log \frac{\mathbb{P}_t(\mathbf{X})}{\mathbb{Q}_t(\mathbf{X})}, f_2(\mathcal{X}, t) \right\rangle d\mathcal{X}$$

$$= \int \mathbb{P}_t(\mathcal{X}) \left\langle \nabla \log \frac{\mathbb{Q}_t(\mathcal{X})}{\mathbb{P}_t(\mathcal{X})}, f_1(\mathcal{X}, t) - f_2(\mathcal{X}, t) \right\rangle$$

$$= \mathbb{E} \left[ \left\langle f_1(\mathcal{X}, t) - f_2(\mathcal{X}, t), \nabla \log \frac{\mathbb{Q}_t(\mathcal{X})}{\mathbb{P}_t(\mathcal{X})} \right\rangle \right]$$

Combining all the results above, we get that,

$$\frac{\partial}{\partial t} \mathrm{KL}(\mathbb{P}_t \| \mathbb{Q}_t) = -g(t)^2 \mathrm{J}(\mathbb{P}_t(\mathcal{X}) \| \mathbb{Q}_t(\mathcal{X})) + \mathbb{E} \left[ \left\langle f_1(\mathcal{X}, t) - f_2(\mathcal{X}, t), \nabla \log \frac{\mathbb{Q}_t(\mathcal{X})}{\mathbb{P}_t(\mathcal{X})} \right\rangle \right]$$

This completes the proof. $\qquad \square$

The next two lemmas capture the properties of Gaussian perturbation in the forward process.

**Lemma B.2.** *Let $\mathbb{P}$ be a probability measure on $\mathbb{R}^{N \times M}$. Consider the Gaussian perturbation of $\mathbb{P}$ that admits a density $\mathbb{P}_{\mu,\sigma}(\mathbf{X})$, where $\mathbf{X} \in \mathbb{R}^{N \times M}$. Specifically, we define*

$$\mathbb{P}_{\mu,\sigma}(\mathbf{X}) \propto \int_{\mathbb{R}^{N \times M}} \exp \left( -\frac{\|\mathbf{X} - \mu \mathbf{Y}\|_F^2}{2\sigma^2} \right) d\mathbb{P}(\mathbf{Y})$$

*where $\| \cdot \|_F$ is the Frobenius norm. Let $\mathbb{P}_{\mu,\sigma}(\mathbf{Y}|\mathbf{X})$ be the conditional probability measure given $\mathbf{X}$, defined as*

$$dP_{\mu,\sigma}(\mathbf{Y}|\mathbf{X}) \propto \exp \left( -\frac{\|\mathbf{X} - \mu \mathbf{Y}\|_F^2}{2\sigma^2} \right) d\mathbb{P}(\mathbf{Y})$$

*If $\mathbb{P}$ admits a density in $C^1(\mathbb{R}^{N \times M})$, we have*

$$\nabla_{\mathbf{X}} \log \mathbb{P}_{\mu,\sigma}(\mathbf{X}) = \frac{1}{\mu} \mathbb{E}_{\mathbb{P}_{\mu,\sigma}(\mathbf{Y}|\mathbf{X})} [\nabla_{\mathbf{Y}} \log \mathbb{P}(\mathbf{Y})]$$

*Proof.*

$$\nabla \log \mathbb{P}_{\mu,\sigma}(\mathbf{X}) = \frac{\int_{\mathbb{R}^{N \times M}} \mathbb{P}(\mathbf{Y}) \nabla_{\mathbf{X}} \left[ \exp \left( -\frac{\|\mathbf{X} - \mu \mathbf{Y}\|^2}{2\sigma^2} \right) \right] d\mathbf{Y}}{\int_{\mathbb{R}^{N \times M}} \mathbb{P}(\mathbf{Y}) \exp \left( -\frac{\|\mathbf{X} - \mu \mathbf{Y}\|}{2\sigma^2} \right)}$$

$$= \frac{-\int_{\mathbb{R}^{N \times M}} \mathbb{P}(\mathbf{Y}) \left[ \nabla_{\mathbf{X}} \exp \left( -\frac{\|\mathbf{X} - \mu \mathbf{Y}\|^2}{2\sigma^2} \right) \right] d\mathbf{Y}}{\mu \int_{\mathbb{R}^{N \times M}} \mathbb{P}(\mathbf{Y}) \exp \left( -\frac{\|\mathbf{X} - \mu \mathbf{Y}\|}{2\sigma^2} \right) d\mathbf{Y}}$$

$$= \frac{\int_{\mathbb{R}^{N \times M}} \mathbb{P}(\mathbf{Y}) \left[ \nabla_{\mathbf{X}} \exp \left( -\frac{\|\mathbf{X} - \mu \mathbf{Y}\|^2}{2\sigma^2} \right) \right] d\mathbf{Y}}{\alpha_{t,s} \int_{\mathbb{R}^{N \times M}} \mathbb{P}(\mathbf{Y}) \exp \left( -\frac{\|\mathbf{X} - \mu \mathbf{Y}\|}{2\sigma^2} \right) d\mathbf{Y}}$$

$$= \frac{1}{\mu} \mathbb{E}_{\mathbb{P}_{\mu,\sigma}(\mathbf{Y}|\mathbf{X})} \nabla_{\mathbf{Y}} \log \mathbb{P}(\mathbf{Y}).$$

$\square$

**Lemma B.3.** *For $0 \le k \le M-1$ and for time $t \in (t_k, t_{k+1}]$, consider the continuous and the discrete approximated reverse SDE starting from $\boldsymbol{\gamma}$,*

$$\mathrm{d}\bar{\mathbf{X}}_t = \left[ 1/2 \bar{\mathbf{X}}_t + \nabla_{\mathbf{X}} \log \mathbb{P}(\mathcal{G}_t) \right] \mathrm{d}t + \mathrm{d}\mathbf{W}_t, \qquad\qquad \bar{\mathbf{X}}_0 = \boldsymbol{\gamma},$$

$$\mathrm{d}\widehat{\mathbf{X}}_t = \left[ 1/2 \widehat{\mathbf{X}}_t + s_{\boldsymbol{\theta}}(\widehat{\mathcal{G}}_{t_k'}, t_k') \right] \mathrm{d}t + \mathrm{d}\mathbf{W}_t, \qquad\qquad \widehat{\mathbf{X}}_0 = \boldsymbol{\gamma},$$

*for time $t \in (t_k, t_{k+1}]$. Let $\bar{\mathbb{P}}_{t|t_k}$ be the density of $\bar{\mathbf{X}}_t$ given $\bar{\mathbf{X}}_{t_k}$ and $\widehat{\mathbb{P}}_{t|t_k}$ be the density of $\widehat{\mathbf{X}}_t$ given $\widehat{\mathbf{X}}_{t_k}$. Then we have that*

1. *For any $\boldsymbol{\gamma}$, the two processes above satisfy: 1) there is a unique solution; and 2) the density functions are two continuously differentiable for $t > 0$.*

2. *For a.e. $\boldsymbol{\gamma}$ (with respect to he Lebesgue measure), we have that*

$$\lim_{t \mapsto t_k^+} \mathrm{KL} \left( \bar{\mathbb{P}}_{t|t_k}(.|\boldsymbol{\gamma}) \| \widehat{\mathbb{P}}_{t|t_k}(.|\boldsymbol{\gamma}) \right) = 0$$

*Proof.* Let $\mathbb{P}_{[t_k',t]}$ and $\mathbb{Q}_{[t_k',t]}$ denote the path measure of $(\bar{\mathbf{X}}_s)_{t_k' \le s \le t}$ and $(\widehat{\mathbf{X}}_s)_{t_k' \le s \le t}$, respectively. For any $\mathbf{Y} \in \mathbb{R}^{N \times M}$, we have

$$\mathrm{KL}(\bar{\mathbb{P}}_{t|t_k'}(.|\mathbf{Y}) \| \widehat{\mathbb{P}}_{t|t_k'}(.|\mathbf{Y})) \le \mathrm{KL}(\mathbb{P}_{[t_k',t]}(.|\mathbf{Y}) \| \mathbb{Q}_{[t_k',t]}(.|\mathbf{Y}).$$

Thus, it suffices to show

$$\lim_{t \to t_k'+} \mathrm{KL}(\mathbb{P}_{[t_k',t]}(.|\mathbf{Y}) \| \mathbb{Q}_{[t_k',t]}(.|\mathbf{Y})) = 0,$$

for a.e. $\mathbf{Y} \in \mathbb{R}^{N \times M}$. It is easy to check the Novikov condition satisfied under Assumption 3.1 and 3.2. Therefore, we can apply Girsanov change of measure (Revuz & Yor, 2013) on $\mathbb{P}_{[t_k',t]}(.|\mathbf{Y})$ and $\mathbb{Q}_{[t_k',t]}(.|\mathbf{Y})$. Then, for the exponential integrator scheme, we have that

$$\mathrm{KL}(\mathbb{P}_{[t_k',t]}(.|\mathbf{Y}) \| \mathbb{Q}_{[t_k',t]}(.|\mathbf{Y})) = \mathbb{E} \left[ \int_{t_k}^t \|\nabla_{\mathbf{X}} \log \mathbb{P}(\mathcal{G}) - s_{\boldsymbol{\theta}}(\widehat{\mathcal{G}}_{t_i'}, t_i')\|^2 | \bar{\mathbf{X}}_{t_k} = \boldsymbol{\gamma} \right]$$

Again, by Assumption 3.2, we have that $\|\nabla_{\mathbf{X}} \log \mathbb{P}(\mathcal{G}) - s_{\boldsymbol{\theta}}(\widehat{\mathcal{G}}_{t_i'}, t_i')\|^2$ is bounded and therefore, we can apply the dominated convergence theorem (Trench, 2013) and move the limit inside the expectation, i.e.,

$$\lim_{t \to t_k'+} \mathrm{KL}(\mathbb{P}_{[t_k',t]}(.|\mathbf{Y}) \| \mathbb{Q}_{[t_k',t]}(.|\mathbf{Y})) = \lim_{t \to t_k'+} \mathbb{E} \left[ \int_{t_k}^t \|\nabla_{\mathbf{X}} \log \mathbb{P}(\mathcal{G}) - s_{\boldsymbol{\theta}}(\widehat{\mathcal{G}}_{t_i'}, t_i')\|^2 | \bar{\mathbf{X}}_{t_i} = \boldsymbol{\gamma} \right]$$

$$= \mathbb{E} \left[ \lim_{t \to t_k'+} \int_{t_k}^t \|\nabla_{\mathbf{X}} \log \mathbb{P}(\mathcal{G}) - s_{\boldsymbol{\theta}}(\widehat{\mathcal{G}}_{t_i'}, t_i')\|^2 | \bar{\mathbf{X}}_{t_i} = \boldsymbol{\gamma} \right]$$

$$= 0$$

This complete the proof for the exponential integrator scheme. Similarly for the Euler-Maruyama scheme, we have that

$$\mathrm{KL}(\mathbb{P}_{[t'_k,t]}(.|\mathbf{Y})\|\mathbb{Q}_{[t'_k,t]}(.|\mathbf{Y})) = \mathbb{E}\left[\int_{t_k}^{t}\|\nabla_{\mathbf{X}}\log\mathbb{P}(\mathcal{G}_t) - s_{\boldsymbol{\theta}}(\widehat{\mathcal{G}}_{t'_i}, t'_i) + 1/2(\mathcal{G}_t - \boldsymbol{\gamma})\|^2|\bar{\mathbf{X}}_{t_i} = \boldsymbol{\gamma}\right].$$

Again, by Assumption 3.1 and 3.2, we can conclude that $\|\nabla_{\mathbf{X}}\log\mathbb{P}(\mathcal{G}_t) - s_{\boldsymbol{\theta}}(\widehat{\mathcal{G}}_{t'_i}, t'_i) + 1/2(\mathcal{G}_t - \boldsymbol{\gamma})\|^2$ is bounded. Then, again, by the dominant convergence theorem, we get that

$$\lim_{t\to t'_k+}\mathrm{KL}(\mathbb{P}_{[t'_k,t]}(.|\mathbf{Y})\|\mathbb{Q}_{[t'_k,t]}(.|\mathbf{Y})) = \mathbb{E}\left[\int_{t_k}^{t}\|\nabla_{\mathbf{X}}\log\mathbb{P}(\mathcal{G}_t) - s_{\boldsymbol{\theta}}(\widehat{\mathcal{G}}_{t'_i}, t'_i) + 1/2(\mathcal{G}_t - \boldsymbol{\gamma})\|^2|\bar{\mathbf{X}}_{t_i} = \boldsymbol{\gamma}\right]$$

$$= \mathbb{E}\left[\lim_{t\to t'_k+}\int_{t_k}^{t}\|\nabla_{\mathbf{X}}\log\mathbb{P}(\mathcal{G}_t) - s_{\boldsymbol{\theta}}(\widehat{\mathcal{G}}_{t'_i}, t'_i) + 1/2(\mathcal{G}_t - \boldsymbol{\gamma})\|^2|\bar{\mathbf{X}}_{t_i} = \boldsymbol{\gamma}\right]$$

$$= 0$$

This completes the proof. $\square$

We have the following results for the decomposition of the convergence bound for the Euler-Maruyama scheme and the exponential integrator scheme. We emphasize that the result below is independent of the generation paradigms.

**Lemma B.4.** *For the exponential integrator scheme, we have that,*

$$\mathrm{KL}(\mathbb{P}(\mathbf{X})\|\mathbb{P}(\widehat{\mathbf{X}}_T)) \lesssim \mathrm{KL}(\mathbb{P}(\mathbf{X}_T)\|\boldsymbol{\Pi}_{\mathbf{X}}) + \sum_{i=1}^{M}\int_{t_{i-1}}^{t_i}\|s_{\boldsymbol{\theta}}(\mathcal{G}_t, t) - \nabla_{\mathbf{X}}\log\mathbb{P}(\mathcal{G}_{t_i})\|^2\mathrm{d}t \tag{B.1}$$

$$+ \sum_{i=1}^{M}\int_{t_{i-1}}^{t_i}\mathbb{E}\|\nabla_{\mathbf{X}}\log\mathbb{P}_t(\mathcal{G}_t) - \nabla_{\mathbf{X}}\log\mathbb{P}(\mathcal{G}_{t_i})\|^2\mathrm{d}t,$$

$$\mathrm{KL}(\mathbb{P}(\mathbf{A})\|\mathbb{P}(\widehat{\mathbf{A}}_T)) \lesssim \mathrm{KL}(\mathbb{P}(\mathbf{A}_T)\|\boldsymbol{\Pi}_{\mathbf{A}}) + \sum_{i=1}^{M}\int_{t_{i-1}}^{t_i}\|s_{\boldsymbol{\phi}}(\mathcal{G}_t, t) - \nabla_{\mathbf{A}}\log\mathbb{P}(\mathcal{G}_{t_i})\|^2\mathrm{d}t \tag{B.2}$$

$$+ \sum_{i=1}^{M}\int_{t_{i-1}}^{t_i}\mathbb{E}\|\nabla_{\mathbf{A}}\log\mathbb{P}_t(\mathcal{G}_t) - \nabla_{\mathbf{A}}\log\mathbb{P}(\mathcal{G}_{t_i})\|^2\mathrm{d}t.$$

*For the Euler-Maruyama scheme, we have that,*

$$\mathrm{KL}(\mathbb{P}(\mathbf{X})\|\mathbb{P}(\widehat{\mathbf{X}}_T)) \lesssim \mathrm{KL}(\mathbb{P}(\mathbf{X}_T)\|\boldsymbol{\Pi}_{\mathbf{X}}) + \sum_{i=1}^{M}\int_{t_{i-1}}^{t_i}\|s_{\boldsymbol{\theta}}(\mathcal{G}_t, t) - \nabla_{\mathbf{X}}\log\mathbb{P}(\mathcal{G}_{t_i})\|^2\mathrm{d}t \tag{B.3}$$

$$+ \sum_{i=1}^{M}\int_{t_{i-1}}^{t_i}\left(\mathbb{E}\|\nabla_{\mathbf{X}}\log\mathbb{P}_t(\mathcal{G}_t) - \nabla_{\mathbf{X}}\log\mathbb{P}(\mathcal{G}_{t_{i-1}})\|^2 + \mathbb{E}\|\mathbf{X}_t - \mathbf{X}_{t_i}\|^2\right)\mathrm{d}t,$$

$$\mathrm{KL}(\mathbb{P}(\mathbf{A})\|\mathbb{P}(\widehat{\mathbf{A}}_T)) \lesssim \mathrm{KL}(\mathbb{P}(\mathbf{A}_T)\|\boldsymbol{\Pi}_{\mathbf{A}}) + \sum_{i=1}^{M}\int_{t_{i-1}}^{t_i}\|s_{\boldsymbol{\phi}}(\mathcal{G}_t, t) - \nabla_{\mathbf{A}}\log\mathbb{P}(\mathcal{G}_{t_i})\|^2\mathrm{d}t \tag{B.4}$$

$$+ \sum_{i=1}^{M}\int_{t_{i-1}}^{t_i}\left(\mathbb{E}\|\nabla_{\mathbf{A}}\log\mathbb{P}(\mathcal{G}_t) - \nabla_{\mathbf{A}}\log\mathbb{P}(\mathcal{G}_{t_i})\|^2 + \mathbb{E}\|\mathbf{A}_t - \mathbf{A}_{t_i}\|^2\right)\mathrm{d}t.$$

*Proof.* We start with deriving the expression for the feature progression. Consider an arbitrary time interval $t_i < t \le t_{i+1}$, let $\bar{\mathbb{P}}_{t|t_i}$ denote the distribution of $\bar{\mathbf{X}}_t$ given $\bar{\mathbf{X}}_{t_i}$. Let $\widehat{\mathbb{P}}_{t|t_i}$ denote the distribution of the discrete approximation $\widehat{\mathbf{X}}_t$ given $\bar{\mathbf{X}}_{t_i}$.

Then for any $\boldsymbol{\gamma} \in \mathbb{R}^{N\times F}$, by the chain rule of KL-divergence, we can get the following progression relation,

$$\mathrm{KL}(\bar{\mathbb{P}}_{t'_{i+1}}\|\widehat{\mathbb{P}}_{t'_{i+1}}) \le \mathbb{E}_{\bar{\mathbb{P}}_{t'}}\mathrm{KL}(\bar{\mathbb{P}}_{t'_{i+1}|t'_i}(.|\boldsymbol{\gamma})\|\widehat{\mathbb{P}}_{t'_{i+1}|t'_i}(.|\boldsymbol{\gamma})) + \mathrm{KL}(\bar{\mathbb{P}}_{t'_i}\|\widehat{\mathbb{P}}_{t'_i}). \tag{B.5}$$

Equivalently, we have that

$$\mathrm{KL}(\bar{\mathbb{P}}_{t'_{i+1}}||\widehat{\mathbb{P}}_{t'_{i+1}}) - \mathrm{KL}(\bar{\mathbb{P}}_{t'_i}||\widehat{\mathbb{P}}_{t'_i}) \leq \mathbb{E}_{\bar{\mathbb{P}}_{t'}}\mathrm{KL}(\bar{\mathbb{P}}_{t'_{i+1}|t'_i}(.|\boldsymbol{\gamma})||\widehat{\mathbb{P}}_{t'_{i+1}|t'_i}(.|\boldsymbol{\gamma})).$$

We can observe that if we do a telescope sum over $0 \leq i \leq M$, the left-hand side can cancel most of the term and left of $\mathrm{KL}(\bar{\mathbb{P}}_T||\widehat{\mathbb{P}}_T)$ and $\mathrm{KL}(\bar{\mathbb{P}}_0||\widehat{\mathbb{P}}_0)$.

Therefore, we can focus on the right hand side and deriving an expression for

$$\mathbb{E}_{\bar{\mathbb{P}}_{t'}}\mathrm{KL}(\bar{\mathbb{P}}_{t'_{i+1}|t'_i}(.|\boldsymbol{\gamma})||\widehat{\mathbb{P}}_{t'_{i+1}|t'_i}(.|\boldsymbol{\gamma})).$$

By Lemma B.3 we have that the two process satisfy the conditions: 1) unique solution and 2) twice continuously differentiable. Then, by Lemma B.1, we have the following time evolution relation for any $\boldsymbol{\gamma}$ and $t > t_i$

$$\frac{\mathrm{d}}{\mathrm{d}t}\mathbb{E}_{\bar{\mathbb{P}}_{t'}}\mathrm{KL}(\bar{\mathbb{P}}_{t'_{i+1}|t'_i}(.|\boldsymbol{\gamma})||\widehat{\mathbb{P}}_{t'_{i+1}|t'_i}(.|\boldsymbol{\gamma})) = -\frac{1}{2}\mathbb{E}_{\bar{\mathbb{P}}_{t'|t'_i}(\mathbf{X}_{t'}|\boldsymbol{\gamma})}\left\|\nabla\log\frac{\bar{\mathbb{P}}_{t'|t'_i}(\mathbf{X}_t|\boldsymbol{\gamma})}{\widehat{\mathbb{P}}_{t'|t'_i}(\mathbf{X}_{t'}|\boldsymbol{\gamma})}\right\|^2$$

$$+ \mathbb{E}_{\bar{\mathbb{P}}_{t'|t'_i}(\mathbf{X}_{t'}|\boldsymbol{\gamma})}\left[\left\langle\nabla\log\bar{\mathbb{P}}'_t(\mathcal{G}'_t) - s_{\boldsymbol{\theta}(\mathcal{G}_{t'_i},t'_i)} + \frac{1}{2}(\mathbf{X}_t - \mathbf{X}_{t'_i}), \nabla\log\frac{\bar{\mathbb{P}}_{t'|t'_i}(\mathbf{X}|\boldsymbol{\gamma})}{\widehat{\mathbb{P}}_{t'|t'_i}(\mathbf{X}|\boldsymbol{\gamma})}\right\rangle\right]$$

Using the fact that $\langle a, b\rangle \leq \frac{1}{2}\|a\|^2 + \frac{1}{2}\|b\|^2$, we get that

$$\leq -\frac{1}{2}\mathbb{E}_{\bar{\mathbb{P}}_{t'|t'_i}(\mathbf{X}_{t'}|\boldsymbol{\gamma})}\left\|\nabla\log\frac{\bar{\mathbb{P}}_{t'|t'_i}(\mathbf{X}_{t'}|\boldsymbol{\gamma})}{\widehat{\mathbb{P}}_{t'|t'_i}(\mathbf{X}_{t'}|\boldsymbol{\gamma})}\right\|^2$$

$$+ \frac{1}{2}\mathbb{E}_{\bar{\mathbb{P}}_{t'|t'_i}(\mathbf{X}_{t'}|\boldsymbol{\gamma})}\left\|\nabla\log\bar{\mathbb{P}}_{t'}(\mathbf{X}_{t'}) - s_{\boldsymbol{\theta}}(\mathcal{G}_{t'_i}, t'_i) + \frac{1}{2}(\mathbf{X}_{t'} - \mathbf{X}_{t'_i})\right\|^2$$

$$+ \frac{1}{2}\mathbb{E}_{\bar{\mathbb{P}}_{t'|t'_i}(\mathbf{X}_{t'}|\boldsymbol{\gamma})}\left\|\nabla\log\frac{\bar{\mathbb{P}}_{t'|t'_i}(\mathbf{X}_{t'}|\boldsymbol{\gamma})}{\widehat{\mathbb{P}}_{t'|t'_i}(\mathbf{X}_{t'}|\boldsymbol{\gamma})}\right\|^2$$

$$= \frac{1}{2}\mathbb{E}_{\bar{\mathbb{P}}_{t'|t'_i}(\mathbf{X}_{t'}|\boldsymbol{\gamma})}\left\|\nabla\log\bar{\mathbb{P}}_{t'}(\mathbf{X}_{t'}) - s_{\boldsymbol{\theta}}(\mathcal{G}_{t'_i}, t'_i) + \frac{1}{2}(\mathbf{X}_{t'} - \mathbf{X}_{t'_i})\right\|^2$$

This means that we have,

$$\frac{\mathrm{d}}{\mathrm{d}t}\mathrm{KL}(\bar{\mathbb{P}}_{t'|t'_i}||\widehat{\mathbb{P}}_{t'|t'_i}(.|\boldsymbol{\gamma})) \leq \frac{1}{2}\mathbb{E}_{\bar{\mathbb{P}}_{t'|t'_i}(\mathbf{X}_{t'}|\boldsymbol{\gamma})}\left\|\nabla\log\bar{\mathbb{P}}_{t'}(\mathbf{X}_{t'}) - s_{\boldsymbol{\theta}}(\mathcal{G}_{t'_i}, t'_i) + \frac{1}{2}(\mathbf{X}_{t'} - \mathbf{X}_{t'_i})\right\|^2$$

$$\leq \frac{1}{2}\mathbb{E}_{\bar{\mathbb{P}}_{t|t_i}(\mathbf{X}|\boldsymbol{\gamma})}\left(\left\|\nabla\log\bar{\mathbb{P}}_{t'}(\mathbf{X}_{t'}) - s_{\boldsymbol{\theta}}(\mathcal{G}_{t'_i}, t'_i)\right\|^2 + \left\|\frac{1}{2}(\mathbf{X}_{t'} - \mathbf{X}_{t'_i})\right\|^2\right),$$

$$\leq \frac{1}{2}\mathbb{E}_{\bar{\mathbb{P}}_{t'|t'_i}(\mathbf{X}_{t'}|\boldsymbol{\gamma})}\left\|\nabla\log\bar{\mathbb{P}}_{t'}(\mathbf{X}_{t'}) - s_{\boldsymbol{\theta}}(\mathcal{G}_{t'_i}, t'_i)\right\|^2 + \frac{1}{2}\mathbb{E}_{\bar{\mathbb{P}}_{t'|t'_i}}\left\|\frac{1}{2}(\mathbf{X}_{t'} - \mathbf{X}_{t'_i})\right\|^2$$

Then, by Lemma B.3, for a.e. $\boldsymbol{\gamma}$, we have

$$\lim_{t'\mapsto t'^+_i}\mathrm{KL}(\bar{\mathbb{P}}_{t'|t'_i}(.|\boldsymbol{\gamma})||\widehat{\mathbb{P}}_{t'|t'_i}(.|\boldsymbol{\gamma})) = 0.$$

This means that $\bar{\mathbb{P}}_{t'|t'_i}(.|\boldsymbol{\gamma})$ and $\widehat{\mathbb{P}}_{t'|t'_i}(.|\boldsymbol{\gamma})$ become increasingly similar as the approximation interval become small. Then, by the derivation above and taking the integral from $t'_i$ to $t'_{i+1}$, we get that,

$$\mathrm{KL}(\bar{\mathbb{P}}_{t'_{i+1}|t'_i}(.|\boldsymbol{\gamma})||\widehat{\mathbb{P}}_{t'_{i+1}|t'_i}(.|\boldsymbol{\gamma})) \leq$$

$$\frac{1}{2}\int_{t'_i}^{t'_{i+1}}\mathbb{E}_{\bar{\mathbb{P}}_{t'|t'_i}(\mathbf{X}_{t'}|\boldsymbol{\gamma})}\left\|\nabla\log\bar{\mathbb{P}}_{t'}(\mathbf{X}_{t'}) - s_{\boldsymbol{\theta}}(\mathcal{G}_{t'_i}, t'_i)\right\|^2 + \frac{1}{2}\mathbb{E}_{\bar{\mathbb{P}}_{t'|t'_i}}\left\|\frac{1}{2}(\mathbf{X}_{t'} - \mathbf{X}_{t'_i})\right\|^2 \mathrm{d}t.$$

Because of the fact that $\bar{\mathbb{P}}'_t(\mathbf{X}_{t'})$ is absolutely continuous with respect to the Lebesgue measure, integrating both sides above with respect to $\bar{\mathbb{P}}_{t'_i}$ we get,

$$\mathbb{E}_{\bar{\mathbb{P}}_{t'_i}} \mathrm{KL}(\bar{\mathbb{P}}_{t'_{i+1}|t'_i}(.|\boldsymbol{\gamma}) \| \widehat{\mathbb{P}}_{t'_{i+1}|t'_i}(.|\boldsymbol{\gamma})) \leq$$
$$\frac{1}{2} \int_{t'_i}^{t'_{i+1}} \mathbb{E}_{\bar{\mathbb{P}}_{t'}} \left\| \nabla \log \bar{\mathbb{P}}_{t'}(\mathbf{X}_{t'}) - s_{\boldsymbol{\theta}}(\mathcal{G}_{t'_i}, t'_i) \right\|^2 + \frac{1}{2} \mathbb{E}_{\bar{\mathbb{P}}_{t'}(\mathbf{X}_{t'})} \left\| \frac{1}{2}(\mathbf{X}_{t'} - \mathbf{X}_{t'_i}) \right\|^2 \mathrm{d}t$$

Substitute the above result into the progression relation given in Eq. B.5, we get the following progression relation,

$$\mathrm{KL}(\bar{\mathbb{P}}_{t'_{i+1}} | \widehat{\mathbb{P}}_{t'_i}) \leq \mathrm{KL}(\bar{\mathbb{P}}_{t'_i} | \widehat{\mathbb{P}}_{t'}) +$$
$$\frac{1}{2} \int_{t'_i}^{t'_{i+1}} \mathbb{E}_{\bar{\mathbb{P}}_{t'}(\mathbf{X}_{t'})} \left\| \nabla \log \bar{\mathbb{P}}_{t'}(\mathbf{X}_{t'}) - s_{\boldsymbol{\theta}}(\mathcal{G}_{t'_i}, t'_i) \right\|^2 + \frac{1}{2} \mathbb{E}_{\bar{\mathbb{P}}_{t'}(\mathbf{X}_{t'})} \left\| \frac{1}{2}(\mathbf{X}_{t'} - \mathbf{X}_{t'_i}) \right\|^2 \mathrm{d}t$$
$$= \mathrm{KL}(\bar{\mathbb{P}}_{t'_i} | \widehat{\mathbb{P}}_{t'}) +$$
$$\frac{1}{2} \int_{t'_i}^{t'_{i+1}} \mathbb{E}_{\bar{\mathbb{P}}_{t'}(\mathbf{X}_{t'})} \left\| \nabla \log \bar{\mathbb{P}}_{t'}(\mathbf{X}_{t'}) - s_{\boldsymbol{\theta}}(\mathcal{G}_{t'_i}, t'_i) \right\|^2 + \frac{1}{2} \mathbb{E}_{\bar{\mathbb{P}}_{t'}(\mathbf{X})} \left\| \frac{1}{2}(\mathbf{X}_{t'} - \mathbf{X}_{t'_i}) \right\|^2 \mathrm{d}t$$

Then, summing above iterative relation over $0 \leq i \leq M$ and replacing $\mathbb{P}_{t'} = \mathbb{P}_{T-t}$, we can obtain,

$$\mathrm{KL}(\mathbb{P}_0 \| \widehat{\mathbb{P}}_T) \leq \mathrm{KL}(\mathbb{P}_T \| \boldsymbol{\gamma}) + \frac{1}{2} \sum_{i=0}^{M} \int_{t_i}^{t_{i+1}} \mathbb{E}_{\bar{\mathbb{P}}_t(\mathbf{X})} \left\| \nabla \log \bar{\mathbb{P}}(\mathcal{G}_t) - s_{\boldsymbol{\theta}}(\mathcal{G}_{t_{i+1}}, t_{i+1}) \right\|^2 + \frac{1}{2} \mathbb{E}_{\bar{\mathbb{P}}_t(\mathbf{X})} \left\| \frac{1}{2}(\mathbf{X}_t - \mathbf{X}_{t_{i+1}}) \right\|^2 \mathrm{d}t$$

$$= \mathrm{KL}(\mathbb{P}_T \| \boldsymbol{\gamma}) + \frac{1}{2} \sum_{i=0}^{M} \int_{t_i}^{t_{i+1}} \mathbb{E}_{\bar{\mathbb{P}}_t(\mathbf{X})} \left\| \nabla_{\mathbf{X}} \log \bar{\mathbb{P}}(\mathcal{G}_t) - \nabla_{\mathbf{X}} \log \bar{\mathbb{P}}(\mathcal{G}_{t_{i+1}}) + \nabla_{\mathbf{X}} \log \bar{\mathbb{P}}(\mathcal{G}_{t_{i+1}}) - s_{\boldsymbol{\theta}}(\mathcal{G}_{t_{i+1}}, t_{i+1}) \right\|^2$$
$$+ \frac{1}{2} \mathbb{E}_{\bar{\mathbb{P}}_t(\mathbf{X})} \left\| \frac{1}{2}(\mathbf{X}_t - \mathbf{X}_{t_{i+1}}) \right\|^2 \mathrm{d}t$$

$$\leq \mathrm{KL}(\mathbb{P}_T \| \boldsymbol{\gamma}) + \frac{1}{2} \sum_{i=0}^{M} \int_{t_i}^{t_{i+1}} \mathbb{E}_{\bar{\mathbb{P}}_t(\mathbf{X})} \left\| \nabla_{\mathbf{X}} \log \bar{\mathbb{P}}(\mathcal{G}_t) - \nabla_{\mathbf{X}} \log \bar{\mathbb{P}}(\mathcal{G}_{t_i}) + \nabla_{\mathbf{X}} \log \bar{\mathbb{P}}(\mathcal{G}_{t_i}) - s_{\boldsymbol{\theta}}(\mathcal{G}_{t_i}, t_i) \right\|^2$$
$$+ \frac{1}{2} \mathbb{E}_{\bar{\mathbb{P}}_t(\mathbf{X})} \left\| \frac{1}{2}(\mathbf{X}_t - \mathbf{X}_{t_i}) \right\|^2 \mathrm{d}t$$

$$\leq \frac{1}{2} \sum_{i=0}^{M} \int_{t_i}^{t_{i+1}} \mathbb{E}_{\bar{\mathbb{P}}_t(\mathbf{X})} \left\| \nabla_{\mathbf{X}} \log \bar{\mathbb{P}}(\mathcal{G}_t) - \nabla_{\mathbf{X}} \log \bar{\mathbb{P}}(\mathcal{G}_{t_{i+1}}) \right\|^2 + \mathbb{E}_{\bar{\mathbb{P}}_t(\mathbf{X})} \left\| \nabla_{\mathbf{X}} \log \bar{\mathbb{P}}(\mathcal{G}_{t_{i+1}}) - s_{\boldsymbol{\theta}}(\mathcal{G}_{t_{i+1}}, t_{i+1}) \right\|^2$$
$$+ \frac{1}{2} \mathbb{E}_{\bar{\mathbb{P}}_t(\mathbf{X})} \left\| \frac{1}{2}(\mathbf{X}_t - \mathbf{X}_{t_{i+1}}) \right\|^2 \mathrm{d}t$$

This completes the derivation for the Euler-Marumaya scheme for the feature matrix. The derivation for the structure matrix is symmetric.

Furthermore, to obtain the derivation for the exponential integrator scheme, we only need to replace the time derivative with,

$$\frac{\mathrm{d}}{\mathrm{d}t} \mathrm{KL}(\bar{\mathbb{P}}_{t|t_i} \| \widehat{\mathbb{P}}_{t|t_i}(.|\boldsymbol{\gamma})) \leq \frac{1}{2} \mathbb{E}_{\bar{\mathbb{P}}_{t|t_i}(\mathbf{X}|\boldsymbol{\gamma})} \left\| \nabla \log \bar{\mathbb{P}}_t(\mathbf{X}) - s_{\boldsymbol{\theta}}(\mathcal{G}_{t_i}, t_i) \right\|^2.$$

Then, we can go through the derivation above in a similar manner to obtain the derivation for feature and structure with the exponential integrator scheme. $\square$

**Lemma B.5.** *Under Assumption 3.1, for $T > 1$, we have*

$$\mathrm{KL}(\mathbb{P}_T(\mathbf{X}) \| \boldsymbol{\Pi}_{\mathbf{X}}) \leq (NF + H_{\mathbf{X}})e^{-T},$$
$$\mathrm{KL}(\mathbb{P}_T(\mathbf{A}) \| \boldsymbol{\Pi}_{\mathbf{A}}) \leq (N^2 + H_{\mathbf{X}})e^{-T}.$$

*Proof.* First, we note that that $f(x) = x \log x$ is a convex function for $x > 0$. Then, for any $t > 0$, we can use Jensen's inequality to bound the entropy of $\mathbb{P}_t$:

$$\int \mathbb{P}_t(\mathbf{X}) \log \mathbb{P}_t(\mathbf{X}) \, d\mathbf{X} = \int \left( \int \mathbb{P}_{t|0}(\mathbf{X}|\mathbf{Y}) \mathbb{P}(\mathbf{Y}) \, d\mathbf{Y} \right) \log \left( \int \mathbb{P}_{t|0}(\mathbf{X}|\mathbf{Y}) \mathbb{P}(\mathbf{Y}) \, d\mathbf{Y} \right) \, d\mathbf{X},$$

$$\leq \int \int \mathbb{P}_{t|0}(\mathbf{X}|\mathbf{Y}) \log \mathbb{P}_{t|0}(\mathbf{X}|\mathbf{Y}) \, d\mathbb{P}(\mathbf{Y}) \, d\mathbf{X},$$

$$= \int \left( \int \mathbb{P}_{t|0}(\mathbf{X}|\mathbf{Y}) \log \mathbb{P}_{t|0}(\mathbf{X}|\mathbf{Y}) \, d\mathbf{X} \right) \, d\mathbb{P}(\mathbf{Y}).$$

Since $\mathbf{X}_t | \mathbf{X}_0 = \mathbf{Y} \sim \mathbb{N}(\alpha \mathbf{X}_0, \sigma_t^2 \mathbf{I})$, we have

$$\int \mathbb{P}_{t|0}(\mathbf{X}|\mathbf{Y}) \log \mathbb{P}_{t|0}(\mathbf{X}|\mathbf{Y}) \, d\mathbf{X} = -\frac{NF}{2} \log\left(2\pi\sigma_t^2\right) - \frac{NF}{2}.$$

Substitute this back into the derivation before, we have

$$\int \mathbb{P}_t(\mathbf{X}) \log \mathbb{P}_t(\mathbf{X}) \, d\mathbf{X} \leq -\frac{NF}{2} \log\left(2\pi\sigma_t^2\right) - \frac{NF}{2}. \tag{B.6}$$

Then, by the definition of KL divergence, we have that,

$$\mathrm{KL}(\mathbb{P}_t \| \mathbf{\Pi}) = \int \mathbb{P}_t(\mathbf{X}) \log \frac{\mathbb{P}_t(\mathbf{X})}{\mathbf{\Pi_X}} \, d\mathbf{X}$$

$$= \int \mathbb{P}_t(\mathbf{X}) \left[ \log \mathbb{P}_t(\mathbf{X}) - \log \mathbf{\Pi_X} \right] \, d\mathbf{X}$$

Substitute int the definition of standard Gaussian for $\mathbf{\Pi_X}$, we get

$$= \int \mathbb{P}_t(\mathbf{X}) \log \mathbb{P}_t(\mathbf{X}) \, d\mathbf{X} + \mathbb{E}_{\mathbb{P}_t} \left[ \frac{\|\mathbf{X}\|^2}{2} + \frac{NF}{2} \log(2\pi) \right]$$

Subsitute in the result of Eq. B.6 and rearrange the terms, we get

$$\leq \frac{NF}{2} \log \sigma_t^{-2} + \frac{1}{2}(H_\mathbf{X} - NF).$$

Since Langevin dynamics with strongly log-concave stationary distribution converge exponentially, we can obtain

$$\mathrm{KL}(\mathbb{P}_t \| \mathbf{\Pi}) \leq e^{-T+t} \frac{NF}{2} \log \sigma_t^{-2} + \frac{1}{2}(H_\mathbf{X} - NF),$$

Picking $t = \log 2$, $e^t \log\left(\frac{1}{\sigma_t^2}\right) \lesssim 1$, we obtain

$$\leq e^{-T}(NF + H_\mathbf{X}).$$

The proof for the structure is symmetric by replace the dimension of $NF$ with $N^2$. $\qquad\square$

**Lemma B.6.** *Suppose the step size $\Delta_i$ for the Euler-Maruyama scheme satisfies $\Delta_i \leq 1$, then for $1 \leq i \leq M$, we have*

$$\mathbb{E}\|\mathbf{X}_t - \mathbf{X}_{t_i}\|^2 \leq NF(t_i - t) + H_\mathbf{X}(t_i - t)^2, \quad t_{i-1} \leq t \leq t_i,$$
$$\mathbb{E}\|\mathbf{A}_t - \mathbf{A}_{t_i}\|^2 \leq N^2(t_i - t) + H_\mathbf{A}(t_i - t)^2, \quad t_{i-1} \leq t \leq t_i,$$

*Proof.* We start with the feature matrix. By the definition of the forward process, we get that

$$\mathbb{E}\|\mathbf{X}_t - \mathbf{X}_{t_i}\|^2 = \mathbb{E}\left\|\int_t^{t_i} \frac{1}{2}\mathbf{X}_s \mathrm{d}s - \int_t^{t_i} \frac{1}{2}\mathrm{d}\mathbf{W}_s\right\|^2$$

$$\leq \mathbb{E}\left\|\int_t^{t_i} \frac{1}{2}\mathbf{X}_s \mathrm{d}s\right\|^2 + \mathbb{E}\left\|\int_t^{t_i} \frac{1}{2}\mathrm{d}\mathbf{W}_s\right\|^2$$

By Cauchy-Schwartz inequality, we can get

$$\lesssim (t_i - t)\int_t^{t_i} \mathbb{E}\|\mathbf{X}_s\|^2 \mathrm{d}s + NF(t_i - t)$$

In addition, we have the explicit expression for the conditional density $\mathbf{X}_s|\mathbf{X}_0$ given by,

$$\mathbb{N}(e^{-1/2s}\mathbf{X}_0, (1 - e^{-s})\mathbf{I}).$$

Then, based on the expression and the Assumption 3.1 that the second moment is bounded, we can get that

$$\mathbb{E}\|\mathbf{X}_s\|^2 \leq H_{\mathbf{X}} + NF.$$

Substitute this back into the derivation before, we get

$$\mathbb{E}\|\mathbf{X}_t - \mathbf{X}_{t_i}\|^2 \lesssim NF(t_i - t) + (NF + H_{\mathbf{X}})(t_i - t)^2.$$

Similarly, for the structure matrix, we have that,

$$\mathbb{E}\|\mathbf{A}_t - \mathbf{A}_{t_i}\|^2 = \mathbb{E}\left\|\int_t^{t_i} \frac{1}{2}\mathbf{A}_s \mathrm{d}s - \int_t^{t_i} \frac{1}{2}\mathrm{d}\mathbf{W}_s\right\|^2$$

$$\leq \mathbb{E}\left\|\int_t^{t_i} \frac{1}{2}\mathbf{A}_s \mathrm{d}s\right\|^2 + \mathbb{E}\left\|\int_t^{t_i} \frac{1}{2}\mathrm{d}\mathbf{W}_s\right\|^2$$

By Cauchy-Schwartz inequality, we can get

$$\lesssim (t_i - t)\int_t^{t_i} \mathbb{E}\|\mathbf{A}_s\|^2 \mathrm{d}s + N^2(t_i - t)$$

In addition, we have the explicit expression for the conditional density $\mathbf{A}_s|\mathbf{A}_0$ given by,

$$\mathbb{N}(e^{-1/2s}\mathbf{A}_0, (1 - e^{-s})\mathbf{I}).$$

Then, based on the expression and the Assumption 3.1 that the second moment is bounded, we can get that

$$\mathbb{E}\|\mathbf{A}_s\|^2 \leq H_{\mathbf{A}} + N^2.$$

Substitute this back into the derivation before, we get

$$\mathbb{E}\|\mathbf{A}_t - \mathbf{A}_{t_i}\|^2 \lesssim N^2(t_i - t) + (N^2 + H_{\mathbf{X}})(t_i - t)^2.$$

$\square$

**Lemma B.7** ((Chewi et al., 2024)). *Let $\mathbb{P}$ be a continuously differentiable probability density. Suppose $\nabla \log \mathbb{P}$ is L-Lipschitz, we have*

$$\mathbb{E}\|\nabla_{\mathbf{X}} \log \mathbb{P}(\mathcal{G})\|^2 \leq NFL,$$

$$\mathbb{E}\|\nabla_{\mathbf{A}} \log \mathbb{P}(\mathcal{G})\|^2 \leq N^2 L.$$

*Proof.* Using integration by parts for feature $\mathbf{X}$, we have:

$$\mathbb{E}\|\nabla_{\mathbf{X}}\log\mathbb{P}(\mathcal{G})\|^2 = \int \mathbb{P}(\mathbf{X})\|\nabla_{\mathbf{X}}\log\mathbb{P}(\mathcal{G})\|^2 d\mathbf{X}$$

$$= \int \langle \nabla\mathbb{P}(\mathbf{X}), \nabla_{\mathbf{X}}\log\mathbb{P}(\mathcal{G})\rangle \, dX,$$

$$= \int \mathbb{P}(\mathbf{X})\Delta\log\mathbb{P}(\mathbf{X})\mathrm{d}\mathbf{X}$$

$$\leq NFL.$$

The derivation for the structure is similar.

$\square$

**Lemma B.8.** *For any $0 \leq t \leq s \leq T$, the forward process satisfies*

$$\mathbb{E}\left[\|\nabla_{\mathbf{X}}\log\mathbb{P}(\mathcal{G}_t) - \nabla_{\mathbf{X}}\log\mathbb{P}(\mathcal{G}_s)\|^2\right] \lesssim$$
$$\mathbb{E}\left[\|\nabla_{\mathbf{X}}\log\mathbb{P}(\mathcal{G}_t) - \nabla_{\mathbf{X}}\log\mathbb{P}(\alpha_{t,s}^{-1}\mathcal{G}_s)\|^2\right] + \mathbb{E}\left[\|\nabla_{\mathbf{X}}\log\mathbb{P}(\mathcal{G}_t)\|^2\right]\left(1 - \alpha_{t,s}^{-1}\right)^2.$$

$$\mathbb{E}\left[\|\nabla_{\mathbf{A}}\log\mathbb{P}(\mathcal{G}_t) - \nabla_{\mathbf{A}}\log\mathbb{P}(\mathcal{G}_s)\|^2\right] \lesssim$$
$$\mathbb{E}\left[\|\nabla_{\mathbf{A}}\log\mathbb{P}(\mathcal{G}_t) - \nabla_{\mathbf{A}}\log\mathbb{P}(\alpha_{t,s}^{-1}\mathcal{G}_s)\|^2\right] + \mathbb{E}\left[\|\nabla_{\mathbf{A}}\log\mathbb{P}(\mathcal{G}_t)\|^2\right]\left(1 - \alpha_{t,s}^{-1}\right)^2.$$

*Proof.* We start with proving for the feature $\mathbf{X}$. By our choice of hyper-parameter, the forward process is a OU process with condition density:

$$\mathbf{X}_s|\mathbf{X}_t \sim \mathbb{N}(\alpha_{t,s}\mathbf{X}_t, (1 - \alpha_{t,s}^2)\mathbf{I})$$
$$\mathbf{A}_s|\mathbf{A}_t \sim \mathbb{N}(\alpha_{t,s}\mathbf{A}_t, (1 - \alpha_{t,s}^2)\mathbf{I})$$

from Lemma B.2, we can rewrite $\nabla_{\mathbf{X}}\log\mathbb{P}(\mathcal{G}_s)$ as

$$\nabla_{\mathbf{X}}\log\mathbb{P}(\mathcal{G}_s) = \alpha_{t,s}^{-1}\mathbb{E}_{\mathbb{P}_{t|s}}\nabla_{\mathbf{X}}\log\mathbb{P}_t(\mathcal{G}),$$

where $\mathbb{P}_{t|s}$ is the conditional density of $\mathcal{G}_t$ given $\mathcal{G}_s$. Thus the discretization error can be bounded by

$$\mathbb{E}\left[\|\nabla_{\mathbf{X}}\log\mathbb{P}(\alpha_{t,s}^{-1}\mathcal{G}_s) - \nabla_{\mathbf{X}}\log\mathbb{P}(\mathcal{G}_s)\|^2\right] = \mathbb{E}_{\mathbb{P}_s}\left[\|\alpha_{t,s}^{-1}\mathbb{E}_{\mathbb{P}_{t|s}}\nabla_{\mathbf{X}}\log\mathbb{P}(\mathcal{G}_t) - \nabla_{\mathbf{X}}\log\mathbb{P}_t(\alpha_{t,s}^{-1}\mathcal{G}_s)\|^2\right]$$
$$\leq \mathbb{E}\left[\|\alpha_{t,s}^{-1}\nabla_{\mathbf{X}}\log\mathbb{P}(\mathcal{G}_t) - \nabla_{\mathbf{X}}\log\mathbb{P}(\alpha_{t,s}^{-1}\mathcal{G}_s)\|^2\right]$$
$$\leq 2(1 - \alpha_{t,s}^{-1})^2\mathbb{E}\left[\|\nabla_{\mathbf{X}}\log\mathbb{P}(\mathcal{G}_t)\|^2\right] +$$
$$2\mathbb{E}\left[\|\nabla_{\mathbf{X}}\log\mathbb{P}_t(\mathcal{G}_t) - \nabla_{\mathbf{X}}\log\mathbb{P}(\alpha_{t,s}^{-1}\mathcal{G}_s)\|^2\right].$$

Therefore, splitting the error into the space-discretization and the time-discretization error, we have

$$\mathbb{E}\left[\|\nabla_{\mathbf{X}}\log\mathbb{P}(\mathcal{G}_t) - \nabla_{\mathbf{X}}\log\mathbb{P}(\alpha_{t,s}^{-1}\mathcal{G}_s)\|^2\right]$$
$$\leq 2\mathbb{E}\left[\|\nabla_{\mathbf{X}}\log\mathbb{P}(\mathcal{G}_t) - \nabla_{\mathbf{X}}\log\mathbb{P}(\alpha_{t,s}^{-1}\mathcal{G}_s)\|^2\right] + 2\mathbb{E}\left[\|\nabla_{\mathbf{X}}\log\mathbb{P}(\alpha_{t,s}^{-1}\mathcal{G}_s) - \nabla_{\mathbf{X}}\log\mathbb{P}(\mathcal{G}_s)\|^2\right]$$
$$\leq 2(1 - \alpha_{t,s}^{-1})^2\mathbb{E}\left[\|\nabla_{\mathbf{X}}\log\mathbb{P}(\mathcal{G}_t)\|^2\right] + 4\mathbb{E}\left[\|\nabla_{\mathbf{X}}\log\mathbb{P}(\mathcal{G}_t) - \nabla\log\mathbb{P}(\alpha_{t,s}^{-1}\mathcal{G}_s)\|^2\right]$$
$$\lesssim \mathbb{E}\left[\|\nabla_{\mathbf{X}}\log\mathbb{P}(\mathcal{G}_t)\|^2\right]\left(1 - \alpha_{t,s}^{-1}\right)^2 + \mathbb{E}\left[\|\nabla_{\mathbf{X}}\log\mathbb{P}(\mathcal{G}_t) - \nabla_{\mathbf{X}}\log\mathbb{P}(\alpha_{t,s}^{-1}\mathcal{G}_s)\|^2\right].$$

The proof for the structure matrix $\mathbf{A}$ is symmetric and therefore obmitted here.

$\square$

**Lemma B.9.** *For $t_{k-1} \leq t \leq t_k$, if $L \geq 1$ and $\Delta_{t_k} \leq 1$, we have:*

$$\mathbb{E}\|\nabla_{\mathbf{X}}\log\mathbb{P}(\mathcal{G}_t) - \nabla_{\mathbf{X}}\log\mathbb{P}(\mathcal{G}_{t_k})\|^2 \lesssim NFL^2(t_k - t) + NFL(t_k - t)^2,$$
$$\mathbb{E}\|\nabla_{\mathbf{A}}\log\mathbb{P}(\mathcal{G}_t) - \nabla_{\mathbf{A}}\log\mathbb{P}(\mathcal{G}_{t_k})\|^2 \lesssim N^2L^2(t_k - t) + N^2L(t_k - t)^2.$$

*Proof.* We start with proving for the feature $\mathbf{X}$.

By Lemmas B.8, we have that

$$\mathbb{E}\|\nabla_{\mathbf{X}} \log \mathbb{P}(\mathcal{G}_t) - \nabla_{\mathbf{X}} \log \mathbb{P}(\mathcal{G}_{t_k})\|$$
$$\lesssim \mathbb{E}\left[\|\nabla_{\mathbf{X}} \log \mathbb{P}(\mathcal{G}_t) - \nabla_{\mathbf{X}} \log \mathbb{P}(\alpha_{t,t_k}^{-1}\mathcal{G}_t)\|^2\right] + \mathbb{E}\left[\|\nabla_{\mathbf{X}} \log \mathbb{P}(\mathcal{G}_t)\|^2\right]\left(1 - \alpha_{t,s}^{-1}\right)^2.$$

Next, we tackle each term of the equation above. By Assumption 3.1, we have that,

$$\mathbb{E}\|\nabla_{\mathbf{X}} \log \mathbb{P}(\mathcal{G}_t) - \nabla_{\mathbf{X}} \log \mathbb{P}(\mathcal{G}_{t_k})\|^2$$
$$= \mathbb{E}\|\nabla_{\mathbf{X}} \log \mathbb{P}(\mathcal{G}_t) - \nabla_{\mathbf{X}} \log \mathbb{P}(\alpha_{t,t_k}^{-1}\mathcal{G}_t)\|^2$$
$$\leq NFL^2 \mathbb{E}\|\mathbf{A}_t\mathbf{X}_t - \alpha_{t,t_k}^{-2}\mathbf{A}_t\mathbf{X}_t\|^2$$
$$= NFL^2(e^{2(t_k-t)} - 1)$$

Using the premise that $t_k - t \leq \Delta_k \leq 1$:

$$\mathbb{E}\|\nabla_{\mathbf{X}} \log \mathbb{P}(\mathcal{G}_t) - \nabla_{\mathbf{X}} \log \mathbb{P}(\mathcal{G}_{t_k})\| \lesssim NFL^2(t_k - t)$$

Then, by Lemma B.7, we have that

$$\mathbb{E}\|\nabla_{\mathbf{X}} \log \mathbb{P}(\mathcal{G}_t)\|^2(1 - \alpha_{t,k}^{-1})^2 \leq NFL(t_k - t)$$

Thus, we conclude that:

$$\mathbb{E}\|\nabla_{\mathbf{X}} \log \mathbb{P}(\mathcal{G}_t) - \nabla_{\mathbf{X}} \log \mathbb{P}(\mathcal{G}_{t_k})\|^2 \lesssim NFL^2(t_k - t) + NFL(t_k - t)^2$$

Similarly for the structure, we have that By Lemmas B.8, we have that

$$\mathbb{E}\|\nabla_{\mathbf{A}} \log \mathbb{P}(\mathcal{G}_t) - \nabla_{\mathbf{A}} \log \mathbb{P}(\mathcal{G}_{t_k})\|$$
$$\lesssim \mathbb{E}\left[\|\nabla_{\mathbf{A}} \log \mathbb{P}(\mathcal{G}_t) - \nabla_{\mathbf{A}} \log \mathbb{P}(\alpha_{t,t_k}^{-1}\mathcal{G}_t)\|^2\right] + \mathbb{E}\left[\|\nabla_{\mathbf{A}} \log \mathbb{P}(\mathcal{G}_t)\|^2\right]\left(1 - \alpha_{t,s}^{-1}\right)^2.$$

Next, we tackle each term of the equation above. By Assumption 3.1, we have that,

$$\mathbb{E}\|\nabla_{\mathbf{A}} \log \mathbb{P}(\mathcal{G}_t) - \nabla_{\mathbf{A}} \log \mathbb{P}(\mathcal{G}_{t_k})\|^2$$
$$= \mathbb{E}\|\nabla_{\mathbf{A}} \log \mathbb{P}(\mathcal{G}_t) - \nabla_{\mathbf{A}} \log \mathbb{P}(\alpha_{t,t_k}^{-1}\mathcal{G}_t)\|^2$$
$$\leq N^2L^2 \mathbb{E}\|\mathbf{A}_t\mathbf{X}_t - \alpha_{t,t_k}^{-2}\mathbf{A}_t\mathbf{X}_t\|^2$$
$$= N^2L^2(e^{2(t_k-t)} - 1)$$

Using the premise that $t_k - t \leq \Delta_k \leq 1$:

$$\mathbb{E}\|\nabla_{\mathbf{A}} \log \mathbb{P}(\mathcal{G}_t) - \nabla_{\mathbf{A}} \log \mathbb{P}(\mathcal{G}_{t_k})\| \lesssim N^2L^2(t_k - t)^2$$

Then, by Lemma B.7, we have that

$$\mathbb{E}\|\nabla_{\mathbf{A}} \log \mathbb{P}(\mathcal{G}_t)\|^2(1 - \alpha_{t,k}^{-1})^2 \leq N^2(t_k - t)$$

Thus, we conclude that:

$$\mathbb{E}\|\nabla_{\mathbf{X}} \log \mathbb{P}(\mathcal{G}_t) - \nabla_{\mathbf{X}} \log \mathbb{P}(\mathcal{G}_{t_k})\|^2 \lesssim N^2L^2(t_k - t) + N^2L(t_k - t)^2$$

$$\square$$

**Lemma B.10.** *For the graph paradigm given by Eq. 3.7, under the premise $\|\mathbf{A}^*\|^2 \leq \sigma_{\mathbf{A}}^2$, for $t_{k-1} \leq t \leq t_k$, if $L \geq 1$ and $\Delta_{t_k} \leq 1$, we have:*

$$\mathbb{E}\|\nabla_{\mathbf{X}} \log \mathbb{P}(\mathcal{G}_t) - \nabla_{\mathbf{X}} \log \mathbb{P}(\mathcal{G}_{t_k})\|^2 \lesssim NFL^2(t_k - t)\sigma_{\mathbf{A}}^2 + NFL(t_k - t)^2,$$

*Proof.* By Lemmas B.8, we have that

$$\mathbb{E}\|\nabla_{\mathbf{X}} \log \mathbb{P}(\mathcal{G}_t) - \nabla_{\mathbf{X}} \log \mathbb{P}(\mathcal{G}_{t_k})\|$$
$$\lesssim \mathbb{E}\|\nabla_{\mathbf{X}} \log \mathbb{P}(\mathcal{G}_t) - \nabla_{\mathbf{X}} \log \mathbb{P}(\mathcal{G}_{t_k})\|^2 + \mathbb{E}\|\nabla_{\mathbf{X}} \log \mathbb{P}(\mathcal{G}_t)\|^2(1 - \alpha_{t,k}^{-1})^2.$$

Next, we tackle each term of the equation above. By Assumption 3.1, we have that,

$$\mathbb{E}\|\nabla_{\mathbf{X}} \log \mathbb{P}(\mathcal{G}_t) - \nabla_{\mathbf{X}} \log \mathbb{P}(\mathcal{G}_{t_k})\|^2$$
$$= \mathbb{E}\|\nabla_{\mathbf{X}} \log \mathbb{P}(\mathcal{G}_t) - \nabla_{\mathbf{X}} \log \mathbb{P}(\alpha_{t,t_k}^{-1}\mathcal{G}_t)\|^2$$
$$\leq NFL^2 \mathbb{E}\|\mathbf{X}_t - \alpha_{t,t_k}^{-1}\mathbf{X}_t\|^2 \|\mathbf{A}^*\|^2$$
$$= NFL^2(e^{t_k - t} - 1)\sigma_{\mathbf{A}}^2$$

Using the premise that $t_k - t \leq \Delta_k \leq 1$:

$$\mathbb{E}\|\nabla_{\mathbf{X}} \log \mathbb{P}(\mathcal{G}_t) - \nabla_{\mathbf{X}} \log \mathbb{P}(\mathcal{G}_{t_k})\|^2 \lesssim NFL^2(t_k - t)\sigma_{\mathbf{A}}^2$$

Then, by Lemma B.7, we have that

$$\mathbb{E}\|\nabla_{\mathbf{X}} \log \mathbb{P}(\mathcal{G}_t)\|^2(1 - \alpha_{t,k}^{-1})^2 \leq NFL(t_k - t)^2$$

Thus, we conclude that:

$$\mathbb{E}\|\nabla_{\mathbf{X}} \log \mathbb{P}(\mathcal{G}_t) - \nabla_{\mathbf{X}} \log \mathbb{P}(\mathcal{G}_{t_k})\|^2 \lesssim NFL^2(t_k - t)\sigma_{\mathbf{A}}^2 + NFL(t_k - t)^2$$

We complete the proof. $\square$

**Lemma B.11.** *For graph generation paradigm given by Eq. 3.8, under the premise that $\|\mathbf{X}\|^2 \leq \sigma_{\mathbf{X}}^2$, for $t_{k-1} \leq t \leq t_k$, if $L \geq 1$ and $\Delta_{t_k} \leq 1$, we have:*

$$\mathbb{E}\|\nabla_{\mathbf{A}} \log \mathbb{P}(\mathcal{G}_t) - \nabla_{\mathbf{A}} \log \mathbb{P}(\mathcal{G}_{t_k})\|^2 \lesssim N^2L^2(t_k - t)\sigma_{\mathbf{A}}^2 + N^2L(t_k - t)^2,$$

*Proof.* By Lemmas B.8, we have that

$$\mathbb{E}\|\nabla_{\mathbf{A}} \log \mathbb{P}(\mathcal{G}_t) - \nabla_{\mathbf{A}} \log \mathbb{P}(\mathcal{G}_{t_k})\|$$
$$\lesssim \mathbb{E}\|\nabla_{\mathbf{A}} \log \mathbb{P}(\mathcal{G}_t) - \nabla_{\mathbf{A}} \log \mathbb{P}(\mathcal{G}_{t_k})\|^2 + \mathbb{E}\|\nabla_{\mathbf{A}} \log \mathbb{P}(\mathcal{G}_t)\|^2(1 - \alpha_{t,k}^{-1})^2.$$

Next, we tackle each term of the equation above. By Assumption 3.1, we have that,

$$\mathbb{E}\|\nabla_{\mathbf{A}} \log \mathbb{P}(\mathcal{G}_t) - \nabla_{\mathbf{A}} \log \mathbb{P}(\mathcal{G}_{t_k})\|^2$$
$$= \mathbb{E}\|\nabla_{\mathbf{A}} \log \mathbb{P}(\mathcal{G}_t) - \nabla_{\mathbf{X}} \log \mathbb{P}(\alpha_{t,t_k}^{-1}\mathcal{G}_t)\|^2$$
$$\leq N^2L^2 \mathbb{E}\|\mathbf{X}^*\|^2 \mathbb{E}\|\mathbf{A}_t - \alpha_{t,t_k}^{-1}\mathbf{A}_t\|^2$$
$$= N^2L^2(e^{t_k - t} - 1)\sigma_{\mathbf{X}}^2$$

Using the premise that $t_k - t \leq \Delta_k \leq 1$:

$$\mathbb{E}\|\nabla_{\mathbf{A}} \log \mathbb{P}(\mathcal{G}_t) - \nabla_{\mathbf{A}} \log \mathbb{P}(\mathcal{G}_{t_k})\|^2 \lesssim N^2L^2(t_k - t)\sigma_{\mathbf{X}}^2$$

Then, by Lemma B.7, we have that

$$\mathbb{E}\|\nabla_{\mathbf{A}}\log\mathbb{P}(\mathcal{G}_t)\|^2(1-\alpha_{t,k}^{-1})^2 \leq N^2(t_k-T)^2$$

Thus, we conclude that:

$$\mathbb{E}\|\nabla\log\mathbb{P}(\mathcal{G}_t)-\nabla\log\mathbb{P}(\mathcal{G}_{t_k})\|^2 \lesssim N^2L^2(t_k-t)\sigma_{\mathbf{A}}^2 + N^2L(t_k-t)^2$$

We complete the proof. □

**Lemma B.12.** *For $t_{k-1}\leq t\leq t_k$, if $L\geq 1$ and $\Delta_{t_k}\leq 1$, we have:*

$$\mathbb{E}\|\nabla_{\mathbf{X}}\log\mathbb{P}(\mathcal{G}_t)-\nabla_{\mathbf{X}}\log\mathbb{P}(\mathcal{G}_{t_k})\|^2 \lesssim NFL^2(t_k-t)^2,$$

*Proof.* By Lemmas B.8, we have that

$$\mathbb{E}\|\nabla_{\mathbf{X}}\log\mathbb{P}(\mathcal{G}_t)-\nabla_{\mathbf{X}}\log\mathbb{P}(\mathcal{G}_{t_k})\|$$
$$\lesssim \mathbb{E}\|\nabla_{\mathbf{X}}\log\mathbb{P}(\mathcal{G}_t)-\nabla_{\mathbf{X}}\log\mathbb{P}(\mathcal{G}_{t_k})\|^2 + \mathbb{E}\|\nabla_{\mathbf{X}}\log\mathbb{P}(\mathcal{G}_t)\|^2(1-\alpha_{t,k}^{-1})^2.$$

Next, we tackle each term of the equation above. By Assumption 3.1, we have that,

$$\mathbb{E}\|\nabla_{\mathbf{X}}\log\mathbb{P}(\mathcal{G}_t)-\nabla_{\mathbf{X}}\log\mathbb{P}(\mathcal{G}_{t_k})\|^2$$
$$= \mathbb{E}\|\nabla_{\mathbf{X}}\log\mathbb{P}(\mathcal{G}_t)-\nabla_{\mathbf{X}}\log\mathbb{P}(\alpha_{t,t_k}^{-1}\mathcal{G}_t)\|^2$$
$$\leq NFL^2\mathbb{E}\|\mathbf{X}_t-\alpha_{t,t_k}^{-1}\mathbf{X}_t\|^2\mathbb{E}\|\mathbf{A}_t-\alpha_{t,t_k}^{-1}\mathbf{A}_t\|^2$$
$$= NFL^2(e^{t_k-t}-1)^2$$

Using the premise that $t_k-t\leq\Delta_k\leq 1$:

$$\mathbb{E}\|\nabla_{\mathbf{X}}\log\mathbb{P}(\mathcal{G}_t)-\nabla_{\mathbf{X}}\log\mathbb{P}(\mathcal{G}_{t_k})\|^2 \lesssim NFL^2(t_k-t)^2$$

Then, by Lemma B.7, we have that

$$\mathbb{E}\|\nabla_{\mathbf{X}}\log\mathbb{P}(\mathcal{G}_t)\|^2(1-\alpha_{t,k}^{-1})^2 \leq NF(t_k)$$

Thus, we conclude that:

$$\lesssim dL^2(t_k-t) + dL(t_k-t)^2 \lesssim dL^2(t_k-t).$$

We complete the proof. □

## C. Proof of Main Results

In this section, we present the proof of the main results. The overall structure of the proof for each paradigm is similar. We first use Lemma B.4 to obtain the decomposition of the convergence bound and then apply the intermediate results we prove in the last section to get the final expression.

### C.1. Proof of Theorem 4.1

In this section, we present the proof for Theorem 4.1.

*Proof of Theorem 4.1.* Let $\mathbb{P}_0$ be the data distribution and $\widehat{\mathbb{P}}_T$ is the learned distribution through the sampling scheme. We start with considering the exponential integrator scheme. By Lemma B.4, we can decompose the KL divergence of $\mathbb{P}_0$ and $\widehat{\mathbb{P}}_T$ as follows,

$$\mathrm{KL}(\mathbb{P}(\mathbf{X}_0)\|\mathbb{P}(\widehat{\mathbf{X}}_T)) \lesssim \mathrm{KL}(\mathbb{P}(\mathbf{X}_T)\|\mathbf{\Pi_X}) + \sum_{i=1}^{M} \int_{t_{i-1}}^{t_i} \|s_{\boldsymbol{\theta}}(\mathcal{G}_{t_i}, t_i) - \nabla_{\mathbf{X}} \log \mathbb{P}(\mathcal{G}_{t_i})\|^2 \mathrm{d}t \tag{C.1}$$

$$+ \sum_{i=1}^{M} \int_{t_{i-1}}^{t_i} \mathbb{E}\|\nabla_{\mathbf{X}} \log \mathbb{P}(\mathcal{G}_t) - \nabla_{\mathbf{X}} \log \mathbb{P}(\mathcal{G}_t)\|^2 \mathrm{d}t.$$

Then, we tackle each term in the equation above.

By Lemma B.5, we can immediately bound the first term as follows,

$$\mathrm{KL}(\mathbb{P}_T(\mathbf{X})\|\mathbf{\Pi_X}) \lesssim (NF + H_{\mathbf{X}})e^{-T}.$$

Next, we tackle the second term. By Assumption 3.2, we have that,

$$\sum_{i=1}^{M} \int_{t_{i-1}}^{t_i} \|s_{\boldsymbol{\theta}}(\mathcal{G}_{t_i}, t_i) - \nabla_{\mathbf{X}} \log \mathbb{P}(\mathcal{G}_{t_i})\|^2 \mathrm{d}t$$

$$= \sum_{i=1}^{M} \Delta_{t_i} \|s_{\boldsymbol{\theta}}(\mathcal{G}_{t_i}, t_i) - \nabla_{\mathbf{X}} \log \mathbb{P}(\mathcal{G}_{t_i})\|^2$$

$$\leq T\epsilon_{\mathbf{X}}^2$$

Therefore, it remains to tackle the last term. By Lemma B.10, we immediately get that

$$\sum_{i=1}^{M} \int_{t_{i-1}}^{t_i} \mathbb{E}\|\nabla_{\mathbf{X}} \log \mathbb{P}(\mathcal{G}_{t_i}) - \nabla_{\mathbf{X}} \log \mathbb{P}(\mathcal{G}_t)\|^2 \mathrm{d}t$$

$$\lesssim \sum_{i=1}^{M} NFL^2 \sigma_{\mathbf{A}}^2 \int_{t_{i-1}}^{t_i} (t_i - t)\mathrm{d}t + NFL \int_{t_{i-1}}^{t_i} (t_i - t)^2$$

$$\lesssim \sum_{i=1}^{M} NFL^2 \sigma_{\mathbf{A}}^2 \Delta_{t_i}^2 + NFL\Delta_{t_i}^3$$

Combining all the results above, we get that,

$$\mathrm{KL}(\mathbb{P}(\mathbf{X}_0)\|\widehat{\mathbb{P}}(\mathbf{X}_T)) \lesssim \mathrm{KL}(\mathbb{P}(\mathbf{X}_T)\|\mathbf{\Pi_X}) + \sum_{i=1}^{M} \int_{t_{i-1}}^{t_i} \|s_{\boldsymbol{\theta}}(\mathcal{G}_{t_i}, t_i) - \nabla_{\mathbf{X}} \log \mathbb{P}(\mathcal{G}_{t_i})\|^2 \mathrm{d}t$$

$$+ \sum_{i=1}^{M} \int_{t_{i-1}}^{t_i} \mathbb{E}\|\nabla_{\mathbf{X}} \log \mathbb{P}(\mathcal{G}_{t_i}) - \nabla_{\mathbf{X}} \log \mathbb{P}(\mathcal{G}_t)\|^2 \mathrm{d}t$$

$$\lesssim (NF + H_{\mathbf{X}})e^{-T} + T\epsilon_{\mathbf{X}}^2 + \sum_{i=1}^{M} NFL^2 \sigma_{\mathbf{A}}^2 \Delta_{t_i}^2 + NFL\Delta_{t_i}^3$$

This completes the derivation for the exponential integrator scheme.

Again, from Lemma B.4, we have that the decomposition of the convergence bound of the Euler-Marumaya scheme is given by,

$$\mathrm{KL}(\mathbb{P}(\mathbf{X})\|\mathbb{P}_T(\widehat{\mathbf{X}})) \lesssim \mathrm{KL}(\mathbb{P}(\mathbf{X}_T)\|\mathbf{\Pi_X}) + \sum_{i=1}^{M} \int_{t_{i-1}}^{t_i} \|s_{\boldsymbol{\theta}}(\mathcal{G}_{t_i}, t_i) - \nabla_{\mathbf{X}} \log \mathbb{P}(\mathcal{G}_{t_i})\|^2 \mathrm{d}t \tag{C.2}$$

$$+ \sum_{i=1}^{M} \int_{t_{i-1}}^{t_i} \left( \mathbb{E}\|\nabla_{\mathbf{X}} \log \mathbb{P}_t(\mathcal{G}_t) - \nabla_{\mathbf{X}} \log \mathbb{P}(\mathcal{G}_{t_i})\|^2 + \mathbb{E}\|\mathbf{X}_t - \mathbf{X}_{t_i}\|^2 \right) \mathrm{d}t,$$

From comparing the decomposition of Eulery-Marumaya and the exponential integrator scheme, we can easily see that the difference between the Euler-Marumaya scheme and the exponential integrator scheme is the extra term on

$$\sum_{i=1}^{M} \int_{t_{i-1}}^{t_i} \mathbb{E}\|\mathbf{X}_t - \mathbf{X}_{t_i}\|^2 dt$$

For this, we can use Lemma B.6 and immediately get that

$$\mathbb{E}\|\mathbf{X}_t - \mathbf{X}_{t_i}\|^2 \le NF(t_i - t) + H_{\mathbf{X}}(t_i - t)^2, \quad t_{i-1} \le t \le t_i$$

Substituting this result into the extra term, we have that,

$$\sum_{i=1}^{M} \int_{t_{i-1}}^{t_i} \mathbb{E}\|\mathbf{X}_t - \mathbf{X}_{t_i}\|^2 dt$$

$$\le \sum_{i=1}^{M} \int_{t_{i-1}}^{t_i} NF(t_i - t) + H_{\mathbf{X}}(t_i - t)^2 dt$$

$$\le \sum_{i=1}^{M} NF(t_i - t)^2 + H_{\mathbf{X}}(t_i - t)^3$$

$$\le \sum_{i=1}^{M} NF\Delta_{t_i}^2 + H_{\mathbf{X}}\Delta_{t_i}^3$$

Therefore, combining this result and the result of the exponential integrator scheme, we have,

$$\mathrm{KL}(\mathbb{P}(\mathbf{X}_0)\|\widehat{\mathbb{P}}(\mathbf{X}_T)) \lesssim \mathrm{KL}(\mathbb{P}(\mathbf{X}_T)\|\mathbf{\Pi}_{\mathbf{X}}) + \sum_{i=1}^{M} \int_{t_{i-1}}^{t_i} \|s_{\boldsymbol{\theta}}(\mathcal{G}_{t_{i-1}}, t_{i-1}) - \nabla_{\mathbf{X}} \log \mathbb{P}(\mathcal{G}_{t_{i-1}})\|^2 dt$$

$$+ \sum_{i=1}^{M} \int_{t_{i-1}}^{t_i} \mathbb{E}\|\nabla_{\mathbf{X}} \log \mathbb{P}(\mathcal{G}_t) - \nabla_{\mathbf{X}} \log \mathbb{P}(\mathcal{G}_t)\|^2 dt$$

$$\lesssim (NF + H_{\mathbf{X}})e^{-T} + T\epsilon_{\mathbf{X}}^2 + \sum_{i=1}^{M} NFL^2\sigma_{\mathbf{A}}^2\Delta_{t_i}^2 + NFL\Delta_{t_i}^3 + \sum_{i=1}^{M} NF\Delta_{t_i}^2 + H_{\mathbf{X}}\Delta_{t_i}^3$$

This completes the proof. $\qquad \square$

## C.2. Proof of Theorem 4.2

In this section, we present the proof for Theorem 4.2.

*Proof of Theorem 4.2.* Let $\mathbb{P}_0$ be the data distribution and $\widehat{\mathbb{P}}_T$ is the learned distribution through the sampling scheme. We start with considering the exponential integrator scheme. By Lemma B.4, we can decompose the KL divergence of $\mathbb{P}_0$ and $\widehat{\mathbb{P}}_T$ as follows,

$$\mathrm{KL}(\mathbb{P}(\mathbf{A}_0)\|\mathbb{P}(\widehat{\mathbf{A}}_T)) \lesssim \mathrm{KL}(\mathbb{P}(\mathbf{A}_T)\|\mathbf{\Pi}_{\mathbf{A}}) + \sum_{i=1}^{M} \int_{t_{i-1}}^{t_i} \|s_{\boldsymbol{\theta}}(\mathcal{G}_{t_i}, t_i) - \nabla_{\mathbf{A}} \log \mathbb{P}(\mathcal{G}_{t_i})\|^2 dt \qquad (\text{C.3})$$

$$+ \sum_{i=1}^{M} \int_{t_{i-1}}^{t_i} \mathbb{E}\|\nabla_{\mathbf{A}} \log \mathbb{P}(\mathcal{G}_{t_i}) - \nabla_{\mathbf{A}} \log \mathbb{P}(\mathcal{G}_t)\|^2 dt.$$

Then, we tackle each term in the equation above.
By Lemma B.5, we can immediately bound the first term as follows,

$$\mathrm{KL}(\mathbb{P}_T(\mathbf{A})\|\mathbf{\Pi}_{\mathbf{A}}) \lesssim (N^2 + H_{\mathbf{A}})e^{-T}.$$

Next, we tackle the second term. By Assumption 3.2, we have that,

$$\sum_{i=1}^{M} \int_{t_{i-1}}^{t_i} \|s_\phi(\mathcal{G}_{t_i}, t_i) - \nabla_{\mathbf{A}} \log \mathbb{P}(\mathcal{G}_{t_i})\|^2 \mathrm{d}t$$

$$= \sum_{i=1}^{M} \Delta_{t_i} \|s_\phi(\mathcal{G}_{t_i}, t_i) - \nabla_{\mathbf{A}} \log \mathbb{P}(\mathcal{G}_{t_i})\|^2$$

$$\leq T\epsilon_{\mathbf{A}}^2$$

Therefore, it remains to tackle the last term. By Lemma B.11, we immediately get that

$$\sum_{i=1}^{M} \int_{t_{i-1}}^{t_i} \mathbb{E}\|\nabla_{\mathbf{A}} \log \mathbb{P}(\mathcal{G}_t) - \nabla_{\mathbf{A}} \log \mathbb{P}(\mathcal{G}_{t_i})\|^2 \mathrm{d}t$$

$$\lesssim \sum_{i=1}^{M} \int_{t_{i-1}}^{t_i} N^2 L^2 (t_i - t)\sigma_{\mathbf{A}}^2 + N^2 L(t_i - t)^2 \mathrm{d}t$$

$$\lesssim N^2 L^2 \sigma_{\mathbf{A}}^2 \sum_{i=1}^{M} \Delta_{t_i}^2 + N^2 L \sum_{i=1}^{M} \Delta_{t_i}^3,$$

Combining all the results above, we get that,

$$\mathrm{KL}(\mathbb{P}(\mathbf{A}_0)\|\mathbb{P}(\widehat{\mathbf{A}}_T)) \lesssim \mathrm{KL}(\mathbb{P}(\mathbf{A}_T)\|\mathbf{\Pi}_{\mathbf{A}}) + \sum_{i=1}^{M} \int_{t_{i-1}}^{t_i} \|s_\phi(\mathcal{G}_{t_i}, t_i) - \nabla_{\mathbf{A}} \log \mathbb{P}(\mathcal{G}_{t_i})\|^2 \mathrm{d}t$$

$$+ \sum_{i=1}^{M} \int_{t_{i-1}}^{t_i} \mathbb{E}\|\nabla_{\mathbf{A}} \log \mathbb{P}(\mathcal{G}_{t_i}) - \nabla_{\mathbf{A}} \log \mathbb{P}(\mathcal{G}_t)\|^2 \mathrm{d}t$$

$$\lesssim (N^2 + H_{\mathbf{X}})e^{-T} + T\epsilon_{\mathbf{A}}^2 + N^2 L^2 \sigma_{\mathbf{X}}^2 \sum_{i=1}^{M} \Delta_{t_i}^2 + N^2 L \sum_{i=1}^{M} \Delta_{t_i}^3$$

$$= (N^2 + H_{\mathbf{X}})e^{-T} + T\epsilon_{\mathbf{A}}^2 + N^2 L \left( L\sigma_{\mathbf{X}}^2 \sum_{i=1}^{M} \Delta_{t_i}^2 + \sum_{i=1}^{M} \Delta_{t_i}^3 \right)$$

This complete the derivation for the exponential integrator scheme.

Again, from Lemma B.4, we have that the decomposition of convergence bound of the Euler-Marumaya scheme is given by,

$$\mathrm{KL}(\mathbb{P}(\mathbf{A})\|\mathbb{P}_T(\widehat{\mathbf{A}})) \lesssim \mathrm{KL}(\mathbb{P}(\mathbf{A}_T)\|\mathbf{\Pi}_{\mathbf{A}}) + \sum_{i=1}^{M} \int_{t_{i-1}}^{t_i} \|s_\phi(\mathcal{G}_{t_i}, t_i) - \nabla_{\mathbf{A}} \log \mathbb{P}(\mathcal{G}_{t_i})\|^2 \mathrm{d}t \qquad \text{(C.4)}$$

$$+ \sum_{i=1}^{M} \int_{t_{i-1}}^{t_i} \left( \mathbb{E}\|\nabla_{\mathbf{A}} \log \mathbb{P}_t(\mathcal{G}_t) - \nabla_{\mathbf{A}} \log \mathbb{P}(\mathcal{G}_{t_i})\|^2 + \mathbb{E}\|\mathbf{A}_t - \mathbf{A}_{t_i}\|^2 \right) \mathrm{d}t,$$

From comparing the decomposition of Eulery-Marumaya and the exponential integrator scheme, we can easily see that the difference between Euler-Marumaya scheme and the exponential integrator scheme is the extra term on

$$\sum_{i=1}^{M} \int_{t_{i-1}}^{t_i} \mathbb{E}\|\mathbf{A}_t - \mathbf{A}_{t_i}\|^2 \mathrm{d}t$$

For this, we can use Lemma B.6 and immediately get that

$$\mathbb{E}\|\mathbf{A}_t - \mathbf{A}_{t_i}\|^2 \leq N^2(t_i - t) + H_{\mathbf{A}}(t_i - t)^2, \quad t_{i-1} \leq t \leq t_i$$

Substituting this result into the extra term, we have that,

$$\sum_{i=1}^{M} \int_{t_{i-1}}^{t_i} \mathbb{E}\|\mathbf{A}_t - \mathbf{A}_{t_k}\|^2 \mathrm{d}t$$

$$\leq \sum_{i=1}^{M} \int_{t_{i-1}}^{t_i} N^2(t_i - t) + H_{\mathbf{A}}(t_i - t)^2$$

$$\leq \sum_{i=1}^{M} N^2(t_i - t)^2 + H_{\mathbf{A}}(t_i - t)^3$$

$$\leq \sum_{i=1}^{M} N^2 \Delta_{t_i}^2 + H_{\mathbf{A}} \Delta_{t_i}^3$$

Therefore, combining this result and the result of the exponential integrator scheme, we have,

$$\mathrm{KL}(\mathbb{P}(\mathbf{A}_0)\|\widehat{\mathbb{P}}(\mathbf{A}_T)) \lesssim \mathrm{KL}(\mathbb{P}(\mathbf{A}_T)\|\mathbf{\Pi_A}) + \sum_{i=1}^{M} \int_{t_{i-1}}^{t_i} \|s_{\boldsymbol{\phi}}(\mathcal{G}_{t_i}, t_i) - \nabla_{\mathbf{A}} \log \mathbb{P}(\mathcal{G}_{t_i})\|^2 \mathrm{d}t$$

$$+ \sum_{i=1}^{M} \int_{t_{i-1}}^{t_i} \mathbb{E}\|\nabla_{\mathbf{A}} \log \mathbb{P}(\mathcal{G}_t) - \nabla_{\mathbf{A}} \log \mathbb{P}(\mathcal{G}_{t_i})\|^2 \mathrm{d}t$$

$$\lesssim (N^2 + H_{\mathbf{X}})e^{-T} + T\epsilon_{\mathbf{A}}^2 + N^2 L^2 \sigma_{\mathbf{X}}^2 \sum_{i=1}^{M} \Delta_{t_i}^2 + N^2 L \sum_{i=1}^{M} \Delta_{t_i}^3 + \sum_{i=1}^{M} N^2 \Delta_{t_i}^2 + H_{\mathbf{A}} \Delta_{t_i}^3$$

$$= (N^2 + H_{\mathbf{X}})e^{-T} + T\epsilon_{\mathbf{A}}^2 + (N^2 L^2 \sigma_{\mathbf{X}}^2 + N^2) \sum_{i=1}^{M} \Delta_{t_i}^2 + (N^2 L + H_{\mathbf{A}}) \sum_{i=1}^{M} \Delta_{t_i}^3$$

This completes the proof. □

## C.3. Proof of Theorem 4.3

In this section, we present the proof for Theorem 4.3.

*Proof of Theorem 4.3.* We start with proving the result for the exponential integrator scheme.

Let $\mathbb{P}_0$ be the data distribution and $\widehat{\mathbb{P}}_T$ is the learned distribution through the sampling scheme. We start with considering the exponetial integrator scheme. By Lemma B.4, we can decompose the KL divergence of $\mathbb{P}_0$ and $\widehat{\mathbb{P}}_T$ as follows,

$$\mathrm{KL}(\mathbb{P}(\mathbf{X}_0)\|\mathbb{P}(\widehat{\mathbf{X}}_T)) \lesssim \mathrm{KL}(\mathbb{P}(\mathbf{X}_T)\|\mathbf{\Pi_X}) + \sum_{i=1}^{M} \int_{t_{i-1}}^{t_i} \|s_{\boldsymbol{\theta}}(\mathcal{G}_{t_i}, t_i) - \nabla_{\mathbf{X}} \log \mathbb{P}(\mathcal{G}_{t_i})\|^2 \mathrm{d}t \tag{C.5}$$

$$+ \sum_{i=1}^{M} \int_{t_{i-1}}^{t_i} \mathbb{E}\|\nabla_{\mathbf{X}} \log \mathbb{P}(\mathcal{G}_{t_i}) - \nabla_{\mathbf{X}} \log \mathbb{P}(\mathcal{G}_t)\|^2 \mathrm{d}t.$$

Then, we tackle each term in the equation above.
By Lemma B.5, we can immediately bound the first term as follows,

$$\mathrm{KL}(\mathbb{P}_T(\mathbf{X})\|\mathbf{\Pi_X}) \lesssim (NF + H_{\mathbf{X}})e^{-T}.$$

Next, we tackle the second term. By Assumption 3.2, we have that,

$$\sum_{i=1}^{M} \int_{t_{i-1}}^{t_i} \|s_{\boldsymbol{\theta}}(\mathcal{G}_{t_i}, t_i) - \nabla_{\mathbf{X}} \log \mathbb{P}(\mathcal{G}_{t_i})\|^2 \mathrm{d}t$$

$$= \sum_{i=1}^{M} \Delta_{t_i} \|s_{\boldsymbol{\theta}}(\mathcal{G}_{t_i}, t_i) - \nabla_{\mathbf{X}} \log \mathbb{P}(\mathcal{G}_{t_i})\|^2$$

$$\leq T\epsilon_{\mathbf{X}}^2$$

Therefore, it remains to tackle the last term. By Lemma B.9, we immediately get that

$$\sum_{i=1}^{M} \int_{t_{i-1}}^{t_i} \mathbb{E}\|\nabla_{\mathbf{X}} \log \mathbb{P}(\mathcal{G}_t) - \nabla_{\mathbf{X}} \log \mathbb{P}(\mathcal{G}_t)\|^2 \mathrm{d}t$$

$$\lesssim \sum_{i=1}^{M} \int_{t_{i-1}}^{t_i} NFL^2(t_k - t) + NFL(t_k - t)^2 \mathrm{d}t$$

$$\lesssim NFL^2 \sum_{i=1}^{M} \Delta_{t_i}^2 + NFL \sum_{i=1}^{M} \Delta_{t_i}^3$$

Combining all the results above, we get that,

$$\mathrm{KL}(\mathbb{P}(\mathbf{X}_0)\|\mathbb{P}(\widehat{\mathbf{X}}_T)) \lesssim \mathrm{KL}(\mathbb{P}(\mathbf{X}_T)\|\mathbf{\Pi_X}) + \sum_{i=1}^{M} \int_{t_{i-1}}^{t_i} \|s_{\boldsymbol{\theta}}(\mathcal{G}_{t_i}, t_i) - \nabla_{\mathbf{X}} \log \mathbb{P}(\mathcal{G}_{t_i})\|^2 \mathrm{d}t$$

$$+ \sum_{i=1}^{M} \int_{t_{i-1}}^{t_i} \mathbb{E}\|\nabla_{\mathbf{X}} \log \mathbb{P}(\mathcal{G}_t) - \nabla_{\mathbf{X}} \log \mathbb{P}(\mathcal{G}_{t_i})\|^2 \mathrm{d}t$$

$$\lesssim (NF + H_{\mathbf{X}})e^{-T} + T\epsilon_{\mathbf{X}}^2 + NFL^2 \sum_{i=1}^{M} \Delta_{t_i}^2 + NFL \sum_{i=1}^{M} \Delta_{t_i}^3$$

$$= (NF + H_{\mathbf{X}})e^{-T} + T\epsilon_{\mathbf{X}}^2 + NFL \left( L \sum_{i=1}^{M} \Delta_{t_i}^2 + \sum_{i=1}^{M} \Delta_{t_i}^3 \right)$$

This complete the derivation for the exponential integrator scheme.

Again, from Lemma B.4, we have that the decomposition of convergence bound of the Euler-Marumaya scheme is given by,

$$\mathrm{KL}(\mathbb{P}(\mathbf{X})\|\mathbb{P}_T(\widehat{\mathbf{X}})) \lesssim \mathrm{KL}(\mathbb{P}(\mathbf{X}_T)\|\mathbf{\Pi_X}) + \sum_{i=1}^{M} \int_{t_{i-1}}^{t_i} \|s_{\boldsymbol{\theta}}(\mathcal{G}_{t_i}, t_i) - \nabla_{\mathbf{X}} \log \mathbb{P}(\mathcal{G}_{t_i})\|^2 \mathrm{d}t \qquad \text{(C.6)}$$

$$+ \sum_{i=1}^{M} \int_{t_{i-1}}^{t_i} \left( \mathbb{E}\|\nabla_{\mathbf{X}} \log \mathbb{P}_t(\mathcal{G}_t) - \nabla_{\mathbf{X}} \log \mathbb{P}(\mathcal{G}_{t_i})\|^2 + \mathbb{E}\|\mathbf{X}_t - \mathbf{X}_{t_i}\|^2 \right) \mathrm{d}t,$$

From comparing the decomposition of Eulery-Marumaya and the exponential integrator scheme, we can easily see that the difference between Euler-Marumaya scheme and the exponential integrator scheme is the extra term on

$$\sum_{i=1}^{M} \int_{t_{i-1}}^{t_i} \mathbb{E}\|\mathbf{X}_t - \mathbf{X}_{t_k}\|^2 \mathrm{d}t$$

For this, we can use Lemma B.6 and immediately get that

$$\mathbb{E}\|\mathbf{X}_t - \mathbf{X}_{t_i}\|^2 \leq NF(t_i - t) + H_{\mathbf{X}}(t_i - t)^2, \quad t_{i-1} \leq t \leq t_i$$

Substituting this result into the extra term, we have that,

$$
\begin{aligned}
\sum_{i=1}^{M} & \int_{t_{i-1}}^{t_i} \mathbb{E}\|\mathbf{X}_t - \mathbf{X}_{t_k}\|^2 \mathrm{d}t \\
&\leq \sum_{i=1}^{M} \int_{t_{i-1}}^{t_i} NF(t_i - t) + H_{\mathbf{X}}(t_i - t)^2 \\
&\leq \sum_{i=1}^{M} NF(t_i - t)^2 + H_{\mathbf{X}}(t_i - t)^3 \\
&\leq \sum_{i=1}^{M} NF\Delta_{t_i}^2 + H_{\mathbf{X}}\Delta_{t_i}^3
\end{aligned}
$$

Therefore, combining this result and the result of the exponential integrator scheme, we have,

$$
\begin{aligned}
\mathrm{KL}(\mathbb{P}(\mathbf{X}_0)\|\widehat{\mathbb{P}}(\mathbf{X}_T)) &\lesssim \mathrm{KL}(\mathbb{P}(\mathbf{X}_T)\|\mathbf{\Pi}_{\mathbf{X}}) + \sum_{i=1}^{M} \int_{t_{i-1}}^{t_i} \|s_{\boldsymbol{\theta}}(\mathcal{G}_{t_{i-1}}, t_{i-1}) - \nabla_{\mathbf{X}} \log \mathbb{P}(\mathcal{G}_{t_{i-1}})\|^2 \mathrm{d}t \\
&\quad + \sum_{i=1}^{M} \int_{t_{i-1}}^{t_i} \mathbb{E}\|\nabla_{\mathbf{X}} \log \mathbb{P}(\mathcal{G}_t) - \nabla_{\mathbf{X}} \log \mathbb{P}(\mathcal{G}_t)\|^2 \mathrm{d}t \\
&\lesssim (NF + H_{\mathbf{X}})e^{-T} + T\epsilon_{\mathbf{X}}^2 + NFL^2 \sum_{i=1}^{M} \Delta_{t_i}^2 + NFL \sum_{i=1}^{M} \Delta_{t_i}^3 + \sum_{i=1}^{M} NF\Delta_{t_i}^2 + H_{\mathbf{X}}\Delta_{t_i}^3
\end{aligned}
$$

This completes the proof.

The derivation for the structure matrix $\mathbf{A}$ is similar and therefore is omitted here $\qquad\square$

## D. Additional Results

In this appendix, we present additional results on the selection of hyper-parameters.

**Corollary D.1.** *Suppose the discretization step is uniform $\Delta t_i = T/M \leq 1, \forall i \in 1, ..., M$. Then the convergence results of SGGMs under the exponential integrator scheme are given by*

$$
\begin{aligned}
\mathrm{KL}(\mathbb{P}(\mathbf{X}_0)\|\mathbb{P}(\widehat{\mathbf{X}}_T)) &\lesssim (H_{\mathbf{X}} + NF)e^{-T} + T\epsilon_{\mathbf{X}}^2 + \frac{NFL^2T^2}{M}, \\
\mathrm{KL}(\mathbb{P}(\mathbf{A}_0)\|\mathbb{P}(\widehat{\mathbf{A}}_T)) &\lesssim (H_{\mathbf{A}} + N^2)e^{-T} + T\epsilon_{\mathbf{A}}^2 + \frac{N^2LT^2}{M}.
\end{aligned}
$$

*Furthermore, taking*

$$
T = \max\left\{\log\left(\frac{H_{\mathbf{X}} + NF}{\epsilon_{\mathbf{X}}^2}\right), \log\left(\frac{H_{\mathbf{X}} + N^2}{\epsilon_{\mathbf{A}}^2}\right)\right\},
$$

$$
M = \max\left\{\frac{NFL^2T^2}{\epsilon_{\mathbf{X}}^2}, \frac{N^2L^2T^2}{\epsilon_{\mathbf{A}}^2}\right\},
$$

*we have that the overall generative error of SGGMs is bounded by the score estimation errors, i.e.,*

$$
\mathrm{KL}(\mathbb{P}(\mathbf{X}_0)\|\mathbb{P}(\widehat{\mathbf{X}}_T)) \lesssim \epsilon_{\mathbf{X}}^2, \quad \mathrm{KL}(\mathbb{P}(\mathbf{A}_0)\|\mathbb{P}(\widehat{\mathbf{A}}_T)) \lesssim \epsilon_{\mathbf{A}}^2.
$$

*Proof.* By Theorem 4.3, we have that,

$$\mathrm{KL}(\mathbb{P}(\mathbf{X}_0)\|\mathbb{P}(\widehat{\mathbf{X}}_T)) \lesssim (H_{\mathbf{X}} + NF)e^{-T} + T\epsilon_{\mathbf{X}}^2$$
$$+ NFL\left(L\sum_i^M \Delta t_i^2 + \sum_i^M \Delta t_i^3\right),$$
$$\mathrm{KL}(\mathbb{P}(\mathbf{A}_0)\|\widehat{\mathbb{P}}_T(\mathbf{A})) \lesssim (H_{\mathbf{A}} + N^2)e^{-T} + T\epsilon_{\mathbf{A}}^2$$
$$+ N^2 L\left(L\sum_i^M \Delta t_i^2 + \sum_i^M \Delta t_i^3\right) \tag{D.1}$$

By premise, we have that $\Delta t_i^3 \leq 1$. This means that

$$\Delta t_i^2 \geq \Delta t_i^3$$

This means that we can absorb the higher-order term with a larger constant. This leads to:

$$\mathrm{KL}(\mathbb{P}(\mathbf{X}_0)\|\mathbb{P}(\widehat{\mathbf{X}}_T)) \lesssim (H_{\mathbf{X}} + NF)e^{-T} + T\epsilon_{\mathbf{X}}^2$$
$$+ NFL\left(L\sum_i^M \Delta t_i^2\right),$$
$$\mathrm{KL}(\mathbb{P}(\mathbf{A}_0)\|\widehat{\mathbb{P}}_T(\mathbf{A})) \lesssim (H_{\mathbf{A}} + N^2)e^{-T} + T\epsilon_{\mathbf{A}}^2$$
$$+ N^2 L\left(L\sum_i^M \Delta t_i^2\right) \tag{D.2}$$

Then, replacing $\Delta_{t_i}$ with $T/M$ we get,

$$\mathrm{KL}(\mathbb{P}(\mathbf{X}_0)\|\mathbb{P}(\widehat{\mathbf{X}}_T)) \lesssim (H_{\mathbf{X}} + NF)e^{-T} + T\epsilon_{\mathbf{X}}^2$$
$$+ NFL\left(LT^2/M\right),$$
$$\mathrm{KL}(\mathbb{P}(\mathbf{A}_0)\|\widehat{\mathbb{P}}(\widehat{\mathbf{A}})) \lesssim (H_{\mathbf{A}} + N^2)e^{-T} + T\epsilon_{\mathbf{A}}^2$$
$$+ N^2 L\left(LT^2/M\right) \tag{D.3}$$

With some algebra, we arrive

$$\mathrm{KL}(\mathbb{P}(\mathbf{X}_0)\|\mathbb{P}(\widehat{\mathbf{X}}_T)) \lesssim (H_{\mathbf{X}} + NF)e^{-T} + T\epsilon_{\mathbf{X}}^2 + \frac{NFL^2T^2}{M},$$
$$\mathrm{KL}(\mathbb{P}(\mathbf{A}_0)\|\mathbb{P}(\widehat{\mathbf{A}}_T)) \lesssim (H_{\mathbf{A}} + N^2)e^{-T} + T\epsilon_{\mathbf{A}}^2 + \frac{N^2 LT^2}{M}.$$

Next, substituting in our choice of $T$ and $M$

$$T = \max\left\{\log\left(\frac{H_{\mathbf{X}} + NF}{\epsilon_{\mathbf{X}}^2}\right), \log\left(\frac{H_{\mathbf{X}} + N^2}{\epsilon_{\mathbf{A}}^2}\right)\right\},$$

$$M = \max\left\{\frac{NFL^2T^2}{\epsilon_{\mathbf{X}}^2}, \frac{N^2 L^2 T^2}{\epsilon_{\mathbf{A}}^2}\right\},$$

It is easy to check that the overall generative error of SGGMs is bounded by the score estimation errors, i.e.,

$$\mathrm{KL}(\mathbb{P}(\mathbf{X}_0)\|\mathbb{P}(\widehat{\mathbf{X}}_T)) \lesssim \epsilon_{\mathbf{X}}^2,$$
$$\mathrm{KL}(\mathbb{P}(\mathbf{A}_0)\|\mathbb{P}(\widehat{\mathbf{A}}_T)) \lesssim \epsilon_{\mathbf{A}}^2.$$

$\square$

Similarly, we have the following result for the Euler-Marumaya scheme and omit the proof due to similarity.

**Corollary D.2.** *Suppose the discretization step is uniform* $\Delta t_i = T/M \leq 1, \forall i \in 1, ..., M$. *Then the convergence results of SGGMs under Euler-Marumaya scheme are given by*

$$\mathrm{KL}(\mathbb{P}(\mathbf{X}_0)\|\mathbb{P}(\widehat{\mathbf{X}}_T)) \lesssim (H_{\mathbf{X}} + NF)e^{-T} + T\epsilon_{\mathbf{X}}^2 + \frac{NFL^2T^2}{M},$$

$$\mathrm{KL}(\mathbb{P}(\mathbf{A}_0)\|\mathbb{P}(\widehat{\mathbf{A}}_T)) \lesssim (H_{\mathbf{A}} + N^2)e^{-T} + T\epsilon_{\mathbf{A}}^2 + \frac{N^2LT^2}{M}.$$

*Furthermore, taking*

$$T = \max\left\{\log\left(\frac{H_{\mathbf{X}} + NF}{\epsilon_{\mathbf{X}}^2}\right), \log\left(\frac{H_{\mathbf{X}} + N^2}{\epsilon_{\mathbf{A}}^2}\right)\right\},$$

$$M = \max\left\{\frac{NFL^2T^2}{\epsilon_{\mathbf{X}}^2}, \frac{N^2L^2T^2}{\epsilon_{\mathbf{A}}^2}\right\},$$

*we have that the overall generative error of SGGMs is bounded by the score estimation errors, i.e.,*

$$\mathrm{KL}(\mathbb{P}(\mathbf{X}_0)\|\mathbb{P}(\widehat{\mathbf{X}}_0)) \lesssim \epsilon_{\mathbf{X}}^2, \quad \mathrm{KL}(\mathbb{P}(\mathbf{A}_0)\|\mathbb{P}(\widehat{\mathbf{A}}_0)) \lesssim \epsilon_{\mathbf{A}}^2.$$

## D.1. Bound with Other Measures

In the context of generative models and their convergence properties, it is often useful to bound the discrepancy between two probability distributions, $P$ and $Q$, using different metrics. Among the most commonly used divergence measures are the Kullback-Leibler (KL) divergence, Wasserstein distance, and total variation (TV) distance. In this paper, we adopt the KL divergence as the measure for the analysis of its direct relation with the training objectives. In this section, we present a discussion of how our result with KL divergence also immediately implies a bound on the other two bounds. We start by providing a brief introduction to each of the metrics.

The Kullback-Leibler (KL) divergence, is defined as:

$$\mathrm{KL}(\mathbb{P}\|\mathbb{Q}) = \mathbb{E}_{\mathbb{P}}\left[\log\frac{d\mathbb{P}}{d\mathbb{Q}}\right],$$

measures the expected logarithmic difference between the probability densities $\mathbb{P}$ and $\mathbb{Q}$ corresponding to the distributions $\mathbb{P}$ and $\mathbb{Q}$. This divergence is widely used in variational inference and optimization because of its convenient properties, such as non-negativity and the fact that it is always zero when $\mathbb{P} = \mathbb{Q}$. However, KL divergence is not symmetric and does not satisfy the triangle inequality, making it less suitable as a distance metric in certain contexts.

In contrast, the Wasserstein distance, is defined as:

$$\mathrm{W}_p(\mathbb{P}, \mathbb{Q}) = \inf_{\gamma \in \Gamma(\mathbb{P}, \mathbb{Q})} \left(\mathbb{E}_{(\mathbf{X}, \mathbf{Y}) \sim \gamma} \|\mathbf{X} - \mathbf{Y}\|^p\right)^{1/p},$$

It measures the "cost" of transforming one distribution into another by considering the optimal transport plan $\gamma$ that minimizes this cost. It is a more intuitive and geometrically meaningful metric compared to KL divergence, especially in high-dimensional spaces. Wasserstein distance has the advantage of being symmetric and satisfying the triangle inequality, which makes it a true metric. When $p = 1$, the Wasserstein distance is often referred to as the Earth Mover's Distance (EMD).

Total variation (TV) distance is another useful metric defined as:

$$\mathrm{TV}(\mathbb{P}, \mathbb{Q}) = \frac{1}{2}\int |\mathbb{P}(x) - \mathbb{Q}(x)|dx,$$

which quantifies the maximum difference between the probabilities assigned to the same event by two distributions. TV distance is symmetric and satisfies the triangle inequality, making it a true distance metric. It is tightly connected to other divergences like KL divergence and can be bounded in terms of them.

It is known that the KL divergence can be bounded in terms of both the Wasserstein distance and the total variation distance. Specifically, the following inequalities provide useful bounds:

- KL Divergence and Total Variation Distance:

$$\text{KL}(\mathbb{P}\|\mathbb{Q}) \geq 2\text{TV}(\mathbb{P}, \mathbb{Q})^2 \quad \text{for distributions with bounded support.}$$

  The result above is known to be Pinsker's inequality. It shows that while KL divergence is at least as large as the square of the TV distance, up to a constant factor.

- Wasserstein Distance AND Total Variation Distance:

$$\text{W}_1(\mathbb{P}, \mathbb{Q}) \lesssim \text{TV}(\mathbb{P}, \mathbb{Q}),$$

  where $\text{W}_1(\mathbb{P}, \mathbb{Q})$ is the Wasserstein distance with $p = 1$. The relation above holds for bounded metric space. Then, by transitivity property, we get that $\text{W}_1(\mathbb{P}, \mathbb{Q}) \lesssim \text{KL}(\mathbb{P}\|\mathbb{Q})$.

The results above show that our results in this paper immediately imply a (growth) bound with the other two commonly used measures (at least in some cases) as well.

# E. Experiment Details

In this appendix, we provide additional details on our experimental study. This section elaborates on the testbed setup, graph generation model, the implementation of score-based graph generative models (SGGMs), and the hyperparameter search.

## E.1. Testbed

Our experiments were conducted on a Dell PowerEdge C4140 server. The key specifications of this server, relevant to our research, are as follows:

**CPU:** Dual Intel Xeon Gold 6230 processors, each offering 20 cores and 40 threads.
**GPU:** Four NVIDIA Tesla V100 SXM2 units, each equipped with 32GB of memory, tailored for NV Link.
**Memory:** A total of 256GB RAM, distributed across eight 32GB RDIMM modules.
**Storage:** Dual 1.92TB SSDs with a 6Gbps SATA interface.
**Networking:** Dual 1Gbps NICs and a Mellanox ConnectX-5 EX Dual Port 40/100GbE QSFP28 Adapter with GPUDirect support.
**Operating System:** Ubuntu 18.04 LTS.

## E.2. Graph Generation Model

For our experiments, we primarily utilize the NetworkX Python library (Hagberg et al., 2008). Specifically, we use the default implementations of regular graph generation and the Barabási-Albert model (Pósfai & Barabási, 2016) provided by NetworkX. The Barabási-Albert (BA) model is widely used to generate scale-free networks characterized by a power-law degree distribution. The core idea behind the BA model is preferential attachment, where new nodes are more likely to connect to existing nodes with a higher degree of connections. This mechanism mirrors real-world networks, such as social networks, where popular individuals (nodes) tend to attract more connections.

**Procedure of the Barabási-Albert Model.** The Barabási-Albert model generates a network through the following steps:

1. **Initialization:** Start with a small connected network of $m_0$ nodes.

2. **Growth:** Add one new node at a time. Each new node forms $m$ edges connecting it to $m$ existing nodes.

3. **Preferential Attachment:** The probability that a new node connects to an existing node $i$ is proportional to the degree of node $i$. Formally, the probability $P(i)$ that the new node connects to node $i$ is given by:

$$P(i) = \frac{k_i}{\sum_j k_j}$$

where $k_i$ is the degree of node $i$, and the sum is taken over all existing nodes.

**Hyperparameters.**    The BA model has two key hyperparameters:

- $\mathbf{m_0}$ : The initial number of nodes in the network.

- $\mathbf{m}$ : The number of edges each new node adds when it is introduced to the network. This parameter influences the density and structure of the resulting network.

### E.3. Score Networks and Diffusion

In the experiments, we use a simple one-layer Graph Convolutional Network (GCN) as the score network. However, to ensure sufficient capacity for learning, we use two-layer MLPs (multi-layer perceptrons) preceding the GCN layer. The hidden layer size is set to 500 for all experiments. For the diffusion process, we set the diffusion length $T = 50$ to ensure sufficient diffusion, and we use a uniform step size of 0.1 for the sampling scheme. For simplicity, our experiments focus on the Euler-Maruyama scheme, and early stopping is employed to prevent variance explosion near the end.

### E.4. Hyperparameter Search for Learning Algorithm

To train the SGGMs, we use the widely adopted Adam optimizer (Kingma & Ba, 2014). A simple parameter search is performed for the learning rate, testing values from the set $[0.1, 0.01, 0.001, 0.0001]$. The implementation of the Adam algorithm is provided by the PyTorch library.

