# OpenReview forum: "A Non-Asymptotic Convergent Analysis for Scored-Based Graph Generative Model via a System of Stochastic Differential Equations"
_ICML.cc/2025/Conference — ICML 2025 poster_

### Official Review · Reviewer_LiK1 · 2025-02-25

**Overall Recommendation:** 4

**Summary:**

This paper provides a non-asymptotic convergence analysis for score-based graph generative models (SGGMs), which involve coupled stochastic differential equations for graph structure and node features. The authors explore convergence bounds across three graph generation paradigms and identify factors like graph topology and feature normalization that influence convergence. Theoretical insights are paired with experimental validation using synthetic graphs, offering practical guidance on hyperparameter selection. The work deepens understanding of SGGMs' theoretical behavior and their application in fields such as drug discovery.

**Claims And Evidence:**

The claims made in the submission are well supported by clear and convincing evidence.

**Essential References Not Discussed:**

N/A

**Experimental Designs Or Analyses:**

Experiment design in Section 5 checked, no issue identified.

**Methods And Evaluation Criteria:**

The proposed methods and evaluation criteria make sense.

**Other Comments Or Suggestions:**

N/A

**Other Strengths And Weaknesses:**

Strengths:

1. The paper presents a novel non-asymptotic convergence analysis for score-based graph generative models (SGGMs), extending traditional SGM theories to accommodate the complexities of coupled stochastic differential equations (SDEs), which enhances its theoretical contributions in graph generation.

2. By addressing SGGMs, the paper makes a meaningful contribution to critical real-world applications like drug discovery and protein design, where graph-based generative models have significant practical value.

3. The authors provide empirical evidence to support their theoretical findings, using synthetic graph models. This strengthens the paper's practical relevance and enhances the reliability of the proposed convergence analysis.

Weaknesses:
See questions for authors

**Questions For Authors:**

Although the paper is overall sound and convincing, I have some confusion regarding the equation: $P(g_t|g_0)=p(X_t|X_0)P(A_t|A_0)$ on page 4. If I understand correctly, does this imply that, at time t, the influence of the node feature $X$ and and the adjacency matrix $A$ on the graph are independent of each other?

While this can be explained by the assumption that the generation process is based on a standard Gaussian distribution, an intuitive explanation for why this assumption holds—or at least does not significantly impact the model's performance—would strengthen the paper further.

**Relation To Broader Scientific Literature:**

This paper builds on prior work in score-based generative models (SGMs), which typically use a single SDE. It extends these analyses to score-based graph generative models (SGGMs) involving coupled SDEs for graph structure and node features. The paper provides novel non-asymptotic convergence bounds and insights for hyperparameter tuning.

**Theoretical Claims:**

Theorem 4.1 to Theorem 4.3 in page 6 checked, no issue found.

---

> ### Author Rebuttal · Authors · 2025-03-31
>
> Dear Reviewer, thank you for positive comments and insightful questions. These greatly help us in making our paper better, and we appreciate for the opportunity to address your  questions here.
>
> # Independency in the forward equation
>
> * We would like to clarify that the independence assumption in the forward equation $\mathbb{P}(\mathbf{G}_t|\mathbf{G}_0)=\mathbb{P}(\mathbf{X}_t|\mathbf{X}_0)\mathbb{P}(\mathbf{A}_t|\mathbf{A}_0)$  is conditional on the initial graph $\mathbf{G}_0 = (\mathbf{X}_0, \mathbf{A}_0)$. In other words, while the forward noising processes for $\mathbf{X}$ and $\mathbf{A}$ are independently modeled given their initial states, the resulting variables $\mathbf{X}_t$ and $\mathbf{A}_t$ remain implicitly interdependent due to their dependence on $\mathbf{X}_0$ and  $\mathbf{A}_0$ (that are dependent of each other). This conditional independence assumption aligns with standard practices in score-based graph generation literature, such as those discussed in [1, 2].
>
> * Your question also highlights an exciting future research direction for the analysis of this paper: exploring the impact of replacing the independent noising processes with a jointly dependent one, potentially by comparing their convergence behaviors. We hypothesize that if the diffusion process could better reflect the dependencies in the original data (i.e., between $\mathbf{X}_0$ and $\mathbf{A}_0$), it may lead to improved learning. Specifically, aligning the noise structure with the data dependencies could help the score networks more effectively capture the interactions between features and topology—by providing samples with more consistent relationships throughout the diffusion process.
>
> Thank you again for this valuable point. We will incorporate this discussion into the revised version of our manuscript, and hope our response has satisfactorily addressed your questions.
>
> [1] "Score-based generative modeling of graphs via the system of stochastic differential equations." *International conference on machine learning*.
>
> [2] "DiGress: Discrete Denoising diffusion for graph generation." *The Eleventh International Conference on Learning Representations*.

---

### Official Review · Reviewer_cnQ4 · 2025-03-14

**Overall Recommendation:** 4

**Summary:**

The paper presents an analysis of graph diffusion processes building heavily on results from https://arxiv.org/abs/2211.01916 (at least based on my reading?) . They find that graph size is much more determinative for convergence rate than feature complexity (mirroring results from https://proceedings.neurips.cc/paper/2020/hash/99503bdd3c5a4c4671ada72d6fd81433-Abstract.html ) derive testable implications from their analysis (normalization helps, regular graphs are easier than power law graphs, graph size hurts more than feature size) which are verified in a toy example.

**Claims And Evidence:**

The analysis looks sensible, although I did not have time to critically check each step.
The empirical analysis is a bit weak, but passes a toy example sniff test in what is primarily a theory paper

**Essential References Not Discussed:**

none except the above suggestion.

**Experimental Designs Or Analyses:**

see the methods and evaluation section.

**Methods And Evaluation Criteria:**

As above, the numerics are a toy example, I think it would be beneficiel to check literature results of SOTA models that fit the papers framework and check the graph properties for the expected correlation (maybe kendal tau) with performance to beef up the papers applicability with realtively little work. alterantively, picking tractable datasets or  standard models (SBM, AB is already present, a few more), and confirm with numerics not via the kernel generation method but via somehting more interpretable (the images look the same to me). failing that, at least using suitable statistical tests (see e.g. https://arxiv.org/abs/1811.12808 ) should be performed.

**Other Comments Or Suggestions:**

nothing I can highlight

**Other Strengths And Weaknesses:**

nothing I can highlight

**Questions For Authors:**

nothing I can highlight

**Relation To Broader Scientific Literature:**

I think https://proceedings.neurips.cc/paper/2020/hash/99503bdd3c5a4c4671ada72d6fd81433-Abstract.html is an important paper to cite, since it offers some justification for why feature complexity might not matter. open question of _why_ graph size hurts so much though (intuitively it makes sense)

**Theoretical Claims:**

As stated above, the claims seem supported, but no detailed anaysis has been performed.

---

> ### Author Rebuttal · Authors · 2025-03-31
>
> Dear Reviewer, thank you for the positive comments and thoughtful suggestions. These greatly help us in making our paper better. Below, we outline the efforts we have undertaken and plan to take in response to your suggestions.
>
> ## Suggested Reference
>
> Thank you for pointing out the relevant reference! It will help us further strengthen our results. We will make sure to include it in the revised version of the paper.
>
> ## Empirical study improvement
>
> Thank you very much for your thoughtful suggestions regarding the empirical study — we greatly appreciate the detailed and constructive feedback.
>
> We have attempted to collect data from existing literature involving SOTA models, but so far have not had much success due to the variability in experimental setups. Real-world studies often differ across multiple factors simultaneously, making it challenging to isolate the effects of individual variables such as graph size or feature complexity. That said, we agree that this is a promising direction and will continue investigating it for future updates and revisions.
>
> Regarding the evaluation metric: we chose KL divergence as our primary measure because our theoretical bounds are expressed in terms of KL divergence, allowing for a principled comparison between theory and experiment. While we acknowledge that more interpretable metrics could enhance accessibility, most evaluation methods for graph generation—especially when comparing distributions—ultimately rely on kernel-based similarity metrics (e.g., MMD, Gaussian kernel estimates, etc.). Developing more interpretable or task-specific evaluation metrics remains a meaningful open question in the field of generative modeling.
>
> In response to your suggestion, we conducted an additional experiment using the Erdős–Rényi (ER) random graph model of connection probability $0.1$ [1]. While ER does not allow us to control structural features like degree heterogeneity (e.g., power-law vs. regular), it provides a clean testbed for examining the relative influence of graph size versus feature size, as well as the effect of normalization, under a controlled setting.  The setup mirrors our experiments in the paper.
>
> To quantify the effects, we further computed best-fit line coefficients to estimate the sensitivity of KL-divergence to changes in each variable:
>
> |  | 50 | 100 | 200 | 500 | Best-Fit Line Coefficient |
> | --- | --- | --- | --- | --- | --- |
> | Changing Graph Size (with fixed feature size= 50, with normalization) | 0.56 | 2.00 | 4.86 | 8.67 | 0.017 |
> | Changing Feature size with normalization (with fixed graph size = 50) | 0.56 | 1.23 | 2.23 | 3.88 | 0.007 |
> | Changing Feature size without normalization (with fixed graph size = 50) | 0.60 | 1.31 | 2.62 | 4.46 | 0.008 |
>
> As shown above, the results support our theoretical predictions: graph size has a more pronounced effect on convergence than feature size, and feature normalization improves performance. These trends are consistent with those observed in our original experiments.
>
> Thank you again for your valuable comments. We will incorporate these new results and the accompanying analysis into the revised version of the paper.
>
> [1] Newman, Mark, Networks: An Introduction

---

### Official Review · Reviewer_cgKb · 2025-03-31

**Overall Recommendation:** 3

**Summary:**

The authors present convergence analysis for score-based graph diffusion generative models where both the generation of the feature vectors at each node of the graph as well as the graph structure is generated based on a diffusion model.

**Claims And Evidence:**

The claims made are clear

**Essential References Not Discussed:**

No.

**Experimental Designs Or Analyses:**

I did not check the soundness of the experimental designs.

**Methods And Evaluation Criteria:**

Evaluation and method make sense

**Other Comments Or Suggestions:**

No other comments.

**Other Strengths And Weaknesses:**

I found the main claim of the essence of the technical contribution not clear.

It is true that the two reverse SDEs are interdependent, but what was not at all clear to me was why theoretically can't one just view this as a single SDE on the pair G_t = (Xt, At).

What breaks if one views the generative process as a single SDE on Gt and applies prior results on the convergence of score-based generative models. In any single generative model, the process that generates any sub-component of the vector, is interdependent with the process that generates the remaining sub-components. So this type of difficulty exists in prior proofs too, if one views it as a single diffusion process.

The authors do not explain clearly, why such a route to a theoretical result is not viable, even if in practice one would view them qualitatively as two processes.

The proof seems to essentially be invoking prior theoretical results on convergence for single diffusion processes, but simply twice, for each diffusion. A better explanation as to which technical lemmas are novel would be very elucidating.

**Questions For Authors:**

No other questions.

**Relation To Broader Scientific Literature:**

The joint generation creates extra difficulties in the convergence analysis as compared to prior work that focused on convergence analysis of generative models, but not graphical models. In particular, the two reverse diffusion processes are interdependent.

**Theoretical Claims:**

I did not check the correctness of the proofs. I skimmed the proof structure and key lemmas.

---

> ### Author Rebuttal · Authors · 2025-04-01
>
> Dear Reviewer,  Thank you for your insightful comments! Your questions have helped us sharpen the presentation of our technical contributions, and we sincerely appreciate the opportunity to clarify them here.
>
> ## Why Not Formulate as a Single SDE on $\mathbf{G}_t$
>
> You are absolutely right that, in principle, the coupled reverse SDEs for $\mathbf{X}_t$ (node features) and $\mathbf{A}_t$ (graph structure) could be rewritten as a single SDE over the joint variable $\mathbf{G}_t=(\mathbf{X}_t,\mathbf{A}_t)$. However, the two-process formulation is not just a modeling preference, but a technical necessity for capturing the rich structure and asymmetry in graph data. Specifically, graphs involve:
>
> - Intra-structure dependency: among entries of $\mathbf{A}_t$.
> - Intra-feature dependency: among entries of $\mathbf{X}_t$.
> - Cross-dependency: between $\mathbf{A}_t$ and $\mathbf{X}_t$, which evolves through the shared time parameter and coupled score functions.
>
> The evolution of these dependencies involves coupled score functions that dynamically influence each other. Representing these coupled processes as a single SDE would obscure their distinct dependency structures, hindering our ability to isolate and analyze the individual and cross-dependent behaviors clearly. Crucially, such an aggregation would mask critical insights into the non-isotropic effects of increasing graph size versus feature dimensionality and would obscure how topological properties (such as regular versus power-law structures) impact convergence.
>
> Therefore, the two-process formulation not only matches more closely with the practical implementation (providing more practical insight) but also enables a finer-grained convergence analysis that reveals how these factors differentially affect generative performance—something not accessible in standard single SDE theory.
>
> ## What Is Technically Novel and Important?
>
> We would like to first emphasize that our contribution is the first rigorous theoretical extension of these techniques to score-based graph generative models (SGGMs). Given the critical applications of SGGMs in domains such as drug discovery and protein synthesis, understanding their convergence behavior is essential for ensuring reliability and efficacy. While our overall proof framework builds on classical convergence theory for score-based generative models (SGMs), as we acknowledged in the paper, our work introduces several significant technical innovations:
>
> - Due to the distinct dependency structure in graphs and the two-process formulation (discussed above), the derivations in Lemmas (such as B.5–B.12) are novel extension of existing SDE results. These lemmas require careful handling of the cross-dependencies between $\mathbf{A}_t$ and $\mathbf{X}_t$, which are absent in standard (single-process) SGM analyses. This leads to new forms of error decomposition and convergence bounds tailored to graph-structured data (as demonstrated explicitly in Theorems 4.1–4.3).
> - Another core technical contribution lies in linking convergence behavior to graph-specific properties. Our analysis uniquely leverages spectral graph theory tools to link convergence behaviors directly to graph characteristics (e.g., graph size/feature dimensionality and topology). These tools lead to novel insights, such as identifying a preference for certain graph structures (regular vs. power-law graph), which are absent from the traditional SGM convergence literature.
>
> We sincerely appreciate your valuable feedback, which has allowed us to refine and strengthen our paper. We will incorporate these discussions into the revision and hope our response has satisfactorily addressed your concerns and questions.

---

### Decision · Program_Chairs · 2025-05-01

**Decision:**

Accept (poster)

**Comment:**

The authors present a strong theoretical contribution on the consistency of graph-based diffusion generative models. Their analysis is heavily buidling on prior work on consistency of score based diffusion models, albeit by explicitly analyzing the convergence behavior of the graph component and the node value component they shed light on practical aspects of graph generative modeling.